# ON THE CERTIFIED ROBUSTNESS FOR ENSEMBLE MODELS AND BEYOND

## ABSTRACT

Recent studies show that deep neural networks (DNN) are vulnerable to adversarial examples, which aim to mislead DNNs to make arbitrarily incorrect predictions. To defend against such attacks, both empirical and theoretical defense approaches have been proposed for a single ML model. In this work, we aim to explore and characterize the robustness conditions for ensemble ML models. We prove that the *diversified gradient* and *large confidence margin* are sufficient and necessary conditions for certifiably robust ensemble models under the model-smoothness assumption. We also show that an ensemble model can achieve higher certified robustness than a single base model based on these conditions. To our best knowledge, this is the first work providing tight conditions for the ensemble robustness. Inspired by our analysis, we propose the lightweight Diversity Regularized Training (DRT) for ensemble models. We derive the certified robustness of DRT based ensembles such as standard Weighted Ensemble and Max-Margin Ensemble following the sufficient and necessary conditions. Besides, to efficiently calculate the model-smoothness, we leverage adapted randomized model smoothing to obtain the certified robustness for different ensembles in practice. We show that the certified robustness of ensembles, on the other hand, verifies the necessity of DRT. To compare different ensembles, we prove that when the adversarial transferability among base models is high, Max-Margin Ensemble can achieve higher certified robustness than Weighted Ensemble; vice versa. Extensive experiments show that ensemble models trained with DRT can achieve the state-of-the-art certified robustness under various settings. Our work will shed light on future analysis for robust ensemble models.

## 1 INTRODUCTION

Deep neural networks (DNN) have been widely applied in various applications, such as image classification (Krizhevsky, 2012; He et al., 2016), face recognition (Taigman et al., 2014; Sun et al., 2014), and natural language processing (Vaswani et al., 2017; Devlin et al., 2019). However, it is well-known that DNNs are vulnerable to adversarial examples (Szegedy et al., 2013; Carlini & Wagner, 2017; Xiao et al., 2018), and it has raised great concerns especially when they are deployed in the safety-critical applications such as autonomous driving and facial recognition.

To defend against such attacks, several empirical defenses have been proposed (Papernot et al., 2016b; Buckman et al., 2018; Madry et al., 2018); however, many of them have been attacked again by strong adaptive attackers (Athalye et al., 2018; Tramer et al., 2020). On the other hand, the certified defenses (Wong & Kolter, 2018; Cohen et al., 2019) have been proposed to provide certified robustness guarantees for given ML models, so that no additional attack can break the model under certain conditions. For instance, randomized smoothing has been proposed as an effective defense providing certified robustness (Lecuyer et al., 2019; Li et al., 2019; Cohen et al., 2019; Yang et al., 2020). Compared with other certified robustness approaches such as linear bound propagation (Weng et al., 2018; Mirman et al., 2018) and interval bound propagation (Gowal et al., 2019), randomized smoothing provides a way to smooth a given DNN efficiently and does not depend on the neural network architecture.

However, existing defenses mainly focus on the robustness of a single ML model, and it is unclear whether an ensemble ML model could provide additional robustness. In this work, we aim to char-

acterize the conditions for a robust ensemble and answer the question from both theoretical and empirical perspectives. In particular, we analyze the standard Weighted Ensemble (WE) and Max-Margin Ensemble (MME) protocols, and prove the necessary and sufficient conditions for robust ensemble models under mild model-smoothness assumptions. Under these conditions, we can see that an ensemble model would be more robust than each single base model. The intuitive illustration of their certified robust radius is in Fig 1. Our analysis shows that *diversified gradient* and *large confidence margins* of base models would lead to higher certified robustness for ensemble models.

Inspired by our analysis, we propose **D**iversity-**R**egularized **T**raining, a lightweight regularization-based ensemble training approach. We derive certified robustness for both WE and MME trained with DRT, and realize model-smoothness assumption via randomized smoothing. We analyze different smoothing protocols and prove that Ensemble Before Smoothing provides higher certified robustness. We further prove that when the adversarial transferability among base models is high, MME is more robust than WE.

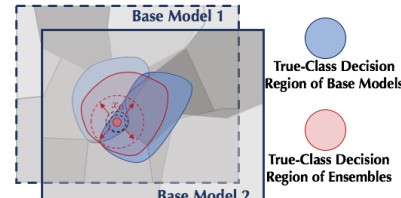

Figure 1: Illustration of a robust ensemble.

We evaluate DRT on a wide range of datasets including MNIST, CIFAR-10, and ImageNet. Extensive experiments show that DRT can achieve higher certified robustness compared with the state-of-the-art baselines with similar training cost as training a single model. Furthermore, when we combine DRT with existing robust models as the base models, DRT can achieve the highest certified robustness to our best knowledge.

We summarize our main contributions as follows: 1) We provide the necessary and sufficient conditions for robust ensemble models including Weighted Ensemble (WE) and Max-Margin Ensemble (MME) under the model-smoothness assumptions. We prove that an ensemble model is more robust than a single base model under the model-smoothness assumption. Our analysis shows that diversified gradients and large confidence margins of base models are the keys to robust ensembles. 2) Based on our analysis, we propose DRT, a lightweight regularization-based training approach, containing both Gradient Diversity Loss and Confidence Margin Loss. 3) We derive certified robustness for ensemble models trained with DRT. The analysis of certified robustness further reveals the importance of DRT. Under mild conditions, we further prove that when the adversarial transferability among base models is high MME is more robust than WE. 4) We conduct extensive experiments to evaluate the effectiveness of DRT on various datasets, which show that DRT can achieve the best certified robustness with similar training time as a single ML model.

**Related work.** DNNs are known vulnerable to adversarial examples (Szegedy et al., 2013). To defend against such attacks, several empirical defenses have been proposed (Papernot et al., 2016b; Madry et al., 2018). For ensemble models, existing work mainly focuses on empirical robustness (Pang et al., 2019; Li et al., 2020; Srisakaokul et al., 2018) where the robustness is measured by accuracy under existing attacks and no certifiable robustness guarantee could be provided or enhanced; or certify the robustness for a vanilla weighted ensemble (Zhang et al., 2019; Liu et al., 2020) using either LP-based (Zhang et al., 2018) verification or randomized smoothing but without diversity enforcement. In this paper, we aim to prove that the gradient diversity and base model margin are two key factors for certified ensemble robustness and based on these key factors, we propose a training approach to enhance the certified robustness of model ensemble.

Randomized smoothing (Cohen et al., 2019) has been proposed to provide certified robustness for a single ML model. It achieved the state-of-the-art certified robustness on large ImageNet and CIFAR-10 dataset under $L_2$ norm. Several approaches have further improved it by: (1) choosing different smoothing distributions for different $L_p$ norms (Dvijotham et al., 2019; Zhang et al., 2020; Yang et al., 2020), and (2) training more robust smoothed classifiers, using data augmentation (Cohen et al., 2019), unlabeled data (Carmon et al., 2019), adversarial training (Salman et al., 2019), regularization (Li et al., 2019; Zhai et al., 2019), and denoising (Salman et al., 2020). However, within our knowledge, there is no work studying how to customize randomized smoothing for ensemble models. In this paper, we compare and select a good randomized smoothing strategy to improve the certified robustness of the ensemble.

In this paper, we mainly focus on the certified robustness under $L_2$ norm. Though randomized smoothing suffers from difficulties when it comes to $L_\infty$ norm (Yang et al., 2020; Kumar et al.,

2020), the analysis of certified robustness and the training approach DRT can be further extended to other $L_p$ norms.

## 2 DIVERSITY-REGULARIZED TRAINING

In this section, we will first provide the robustness conditions for the standard Weighted Ensemble and Max-Margin Ensemble. Using the robustness condition, we can compare the certified robustness of the ensemble models and a single base model. The comparison shows that under model-smoothness assumptions, the ensemble models are more robust in terms of their certified robustness. Motivated by the key factors in the robustness conditions, we then propose **D**iversity-**R**egularized **T**raining.

**Notations.** Throughout the paper, we consider the classification task with $C$ classes. We first define the classification scoring function $f : \mathbb{R}^d \mapsto \mathbf{\Delta}^C$, which maps the input to a *confidence vector*, and $f(x)_i$ represents the confidence for the $i$th class. We mainly focus on the confidence after normalization, i.e., $f(\boldsymbol{x}) \in \mathbf{\Delta}^C = \{\boldsymbol{p} \in \mathbb{R}_{\geq 0}^C : \|\boldsymbol{p}\|_1 = 1\}$ is in the probability simplex. To characterize the *confidence margin* between two classes we define $f^{y_1/y_2}(\boldsymbol{x}) := f(\boldsymbol{x})_{y_1} - f(\boldsymbol{x})_{y_2}$. The corresponding *prediction* $F : \mathbb{R}^d \mapsto [C]$ is defined by $F(\boldsymbol{x}) := \arg\max_{i \in [C]} f(\boldsymbol{x})_i$. We are also interested in the *runner-up prediction* $F^{(2)}(\boldsymbol{x}) := \arg\max_{i \in [C]:i \neq F(\boldsymbol{x})} f(\boldsymbol{x})_i$.

In this paper, we mainly consider the model robustness against the $L_2$-bounded perturbations.

**Definition 1** (*r*-Robustness). For a prediction function $F : \mathbb{R}^d \mapsto [C]$ and input $\boldsymbol{x}_0$, if any instance $\boldsymbol{x} \in \{\boldsymbol{x}_0 + \boldsymbol{\delta} : \|\boldsymbol{\delta}\|_2 \leq r\}$ satisfies $F(\boldsymbol{x}) = F(\boldsymbol{x}_0)$, we say model $F$ is *r-robust* (at point $\boldsymbol{x}_0$).

We map existing certified robustness (Cohen et al., 2019) to $r$-Robustness in Appendix B.1.

### 2.1 ROBUSTNESS CONDITIONS FOR ENSEMBLE MODELS

An ensemble model contains $N$ *base models* $\{f_i\}_{i=1}^N$, where $F_i$ and $F_i^{(2)}$ are their top and runner-up predictions respectively. The ensemble prediction is denoted by $\mathcal{M} : \mathbb{R}^d \mapsto [C]$, which is computed based on outputs of base models following certain ensemble protocols. In this paper, we consider both Weighted Ensemble (WE) and Maximum Margin Ensemble (MME).

**Definition 2** (Weighted Ensemble (WE)). Given $N$ base models $\{f_i\}_{i=1}^N$, and the weight vector $\{w_i\}_{i=1}^N \in \mathbb{R}_+^N$, the Weighted Ensemble is constructed as $\mathcal{M}_{\mathrm{WE}} : \mathbb{R}^d \mapsto [C]$ such that for any input $\boldsymbol{x}_0$:

$$\mathcal{M}_{\mathrm{WE}}(\boldsymbol{x}_0) := \arg\max_{i \in [C]} \sum_{j=1}^N w_j f_j(\boldsymbol{x}_0)_i. \tag{1}$$

**Definition 3** (Max-Margin Ensemble (MME)). Given $N$ base models $\{f_i\}_{i=1}^N$, for input $\boldsymbol{x}_0$, the Max-Margin Ensemble model $\mathcal{M}_{\mathrm{MME}} : \mathbb{R}^d \mapsto [C]$ is defined by

$$\mathcal{M}_{\mathrm{MME}}(\boldsymbol{x}_0) := F_c(\boldsymbol{x}_0) \quad \text{where} \quad c = \arg\max_{i \in [N]} \left( f_i(\boldsymbol{x}_0)_{F_i(\boldsymbol{x}_0)} - f_i(\boldsymbol{x}_0)_{F_i^{(2)}(\boldsymbol{x}_0)} \right). \tag{2}$$

WE sums up the weighted confidence scores of base models $\{f_i\}_{i=1}^N$ with weight vector $\{w_i\}_{i=1}^N$, and predicts the class with the highest value. WE is commonly-used (Zhang et al., 2019; Liu et al., 2020). Max-Margin Ensemble chooses the base model with the largest confidence margin between the top and the runner-up classes, which is a direct extension from max-margin training (Huang et al., 2008).

#### 2.1.1 GENERAL ROBUSTNESS CONDITIONS

For WE, since it predicts the class with the highest aggregated confidence, we can easily observe its *sufficient and necessary conditions* for the certified robustness.

**Proposition 1** (Robustness Condition for WE). *Consider an input $\boldsymbol{x}_0 \in \mathbb{R}^d$ with ground-truth label $y_0 \in [C]$, and a Weighted Ensemble model $\mathcal{M}_{\mathrm{WE}}$ constructed by base models $\{f_i\}_{i=1}^N$ with weights $\{w_i\}_{i=1}^N$. Suppose $\mathcal{M}_{\mathrm{WE}}(\boldsymbol{x}_0) = y_0$. Then, the ensemble $\mathcal{M}_{\mathrm{WE}}$ is r-robust at point $\boldsymbol{x}_0$ if and only if for any $\boldsymbol{x} \in \{\boldsymbol{x}_0 + \boldsymbol{\delta} : \|\boldsymbol{\delta}\|_2 \leq r\}$, $\min_{y_i \in [C]} \sum_{j=1}^N w_j f_j^{y_0/y_i}(\boldsymbol{x}) \geq 0$.*

For MME, however, the model prediction is decided by the base model with the largest confidence margin. This "maximum" is a discrete operator and poses challenges especially in the multi-class setting. We cannot assert that the model predicts the true label by simply looking at the margins between only the true label and other labels unless carefully filtering out possible violated cases. In the following theorem, through careful analysis of the layout of confidence scores (e.g. enumerating the cases where $y_0$ is the top class, runner-up class, or one of other classes, see details in Lemmas B.1 and B.2), we present a succinct but sufficient and necessary condition for MME robustness.

**Theorem 1** (Robustness Condition for MME). *Consider an input $\boldsymbol{x}_0 \in \mathbb{R}^d$ with ground-truth label $y_0 \in [C]$. Let $\mathcal{M}_{\mathrm{MME}}$ be an MME defined over base models $\{f_i\}_{i=1}^N$. Suppose: (1) $\mathcal{M}_{\mathrm{MME}}(\boldsymbol{x}_0) = y_0$; (2) for any $\boldsymbol{x} \in \{\boldsymbol{x}_0 + \boldsymbol{\delta} : \|\boldsymbol{\delta}\|_2 \leq r\}$, given any base model $i \in [N]$, either $F_i(\boldsymbol{x}) = y_0$ or $F_i^{(2)}(\boldsymbol{x}) = y_0$. Then, the ensemble $\mathcal{M}_{\mathrm{MME}}$ is r-robust at point $\boldsymbol{x}_0$ if and only if for any $\boldsymbol{x} \in \{\boldsymbol{x}_0 + \boldsymbol{\delta} : \|\boldsymbol{\delta}\|_2 \leq r\}$,*

$$\max_{i \in [N]} \min_{y_i \in [C] : y_i \neq y_0} f_i^{y_0/y_i}(\boldsymbol{x}) \geq \max_{i \in [N]} \min_{y_i' \in [C] : y_i' \neq y_0} f_i^{y_i'/y_0}(\boldsymbol{x}). \tag{3}$$

In above theorem, the $y_i$'s and $y_i'$'s are associated with each base model $f_i$ and are respectively minimized among all the $C$ classes except class $y_0$. We defer the proof to Appendix B.2. This theorem along with the intermediate lemmas in the proof serves as the foundation for our subsequent analysis.

### 2.1.2 DIVERSIFIED GRADIENTS AND LARGE CONFIDENCE MARGIN CONDITIONS

The conditions in Proposition 1 and Theorem 1 are rather general and involve $\boldsymbol{x} \in \{\boldsymbol{x}_0 + \boldsymbol{\delta} : \|\boldsymbol{\delta}\|_2 \leq r\}$, which is challenging to verify for neural networks due to its non-convexity. In this section, we adapt the above conditions for DNNs based on *confidence scores* and *gradient* of base models at input $\boldsymbol{x}_0$, showing the diversified gradients and large confidence margin are the sufficient and necessary conditions for ensemble robustness.

**Definition 4** ($\beta$-Smoothness). *A differentiable function $f : \mathbb{R}^d \mapsto \mathbb{R}^C$ is $\beta$-smooth, if for any $\boldsymbol{x}, \boldsymbol{y} \in \mathbb{R}^d$ and any output dimension $j \in [C]$, $\frac{\|\nabla_{\boldsymbol{x}} f(\boldsymbol{x})_j - \nabla_{\boldsymbol{y}} f(\boldsymbol{y})_j\|_2}{\|\boldsymbol{x} - \boldsymbol{y}\|_2} \leq \beta$.*

The definition of $\beta$-smoothness is inherited from optimization theory literature, and is equivalent to the curvature bound in certified robustness literature (Singla & Feizi, 2020). Note that smaller $\beta$ indicates smoother models, and when $\beta = 0$ the model is linear.

For **Weighted Ensemble**, we have the following robustness conditions.

**Theorem 2** (Gradient and Confidence Margin Conditions for WE Robustness). *Given input $\boldsymbol{x}_0 \in \mathbb{R}^d$ with ground-truth label $y_0 \in [C]$, and $\mathcal{M}_{\mathrm{WE}}$ as a WE defined over base models $\{f_i\}_{i=1}^N$ with weights $\{w_i\}_{i=1}^N$. $\mathcal{M}_{\mathrm{WE}}(\boldsymbol{x}_0) = y_0$. All base models $f_i$'s are $\beta$-smooth.*

- *(Sufficient Condition) The $\mathcal{M}_{\mathrm{WE}}$ is r-robust at point $\boldsymbol{x}_0$ if for any $y_i \neq y_0$,*

$$\Big\| \sum_{j=1}^N w_j \nabla_{\boldsymbol{x}} f_j^{y_0/y_i}(\boldsymbol{x}_0) \Big\|_2 \leq \frac{1}{r} \sum_{j=1}^N w_j f_j^{y_0/y_i}(\boldsymbol{x}_0) - \beta r \sum_{j=1}^N w_j, \tag{4}$$

- *(Necessary Condition) If $\mathcal{M}_{\mathrm{WE}}$ is r-robust at point $\boldsymbol{x}_0$, for any $y_i \neq y_0$,*

$$\Big\| \sum_{j=1}^N w_j \nabla_{\boldsymbol{x}} f_j^{y_0/y_i}(\boldsymbol{x}_0) \Big\|_2 \leq \frac{1}{r} \sum_{j=1}^N w_j f_j^{y_0/y_i}(\boldsymbol{x}_0) + \beta r \sum_{j=1}^N w_j. \tag{5}$$

The proof directly follows from our general robustness conditions and Taylor expansion at $\boldsymbol{x}_0$.

For **Max-Margin Ensemble** with two base models, we derive the following robustness conditions.

**Theorem 3** (Gradient and Confidence Margin Conditions for MME Robustness). *Given input $\boldsymbol{x}_0 \in \mathbb{R}^d$ with ground-truth label $y_0 \in [C]$, and $\mathcal{M}_{\mathrm{MME}}$ as an MME defined over base models $\{f_1, f_2\}$. $\mathcal{M}_{\mathrm{MME}}(\boldsymbol{x}_0) = y_0$. Both $f_1$ and $f_2$ are $\beta$-smooth.*

- *(Sufficient Condition) The $\mathcal{M}_{\mathrm{MME}}$ is r-robust at point $\boldsymbol{x}_0$ if for any $y_1, y_2 \in [C]$ such that $y_1 \neq y_0$ and $y_2 \neq y_0$,*

$$\|\nabla_{\boldsymbol{x}} f_1^{y_0/y_1}(\boldsymbol{x}_0) + \nabla_{\boldsymbol{x}} f_2^{y_0/y_2}(\boldsymbol{x}_0)\|_2 \leq \frac{1}{r}(f_1^{y_0/y_1}(\boldsymbol{x}_0) + f_2^{y_0/y_2}(\boldsymbol{x}_0)) - 2\beta r. \tag{6}$$

- *(Necessary Condition) Suppose $\boldsymbol{x} \in \{\boldsymbol{x}_0 + \boldsymbol{\delta} : \|\boldsymbol{\delta}\|_2 \leq r\}$, and for $i \in \{1, 2\}$ either $F_i(\boldsymbol{x}) = y_0$ or $F_i^{(2)}(\boldsymbol{x}) = y_0$. If $\mathcal{M}_{\mathrm{MME}}$ is $r$-robust at point $\boldsymbol{x}_0$, for any $y_1, y_2 \in [C]$ such that $y_1 \neq y_0$ and $y_2 \neq y_0$,*

$$\|\nabla_{\boldsymbol{x}} f_1^{y_0/y_1}(\boldsymbol{x}_0) + \nabla_{\boldsymbol{x}} f_2^{y_0/y_2}(\boldsymbol{x}_0)\|_2 \leq \frac{1}{r}(f_1^{y_0/y_1}(\boldsymbol{x}_0) + f_2^{y_0/y_2}(\boldsymbol{x}_0)) + 2\beta r. \tag{7}$$

The proof combines the proof procedure of Theorem 1 with Taylor expansion at $\boldsymbol{x}_0$. We remark that for MME, it is challenging to extend the theorem to the case with $n > 2$ base models. The reason is that the "maximum" operator in MME poses difficulties for expressing the robust condition in succinct form of continuous function. Therefore, Taylor expansion is unable to apply. However, the general tendency should be the same.

We also derive the robustness conditions for a single model for comparison in Appendix B.3.

**Comparison of ensemble and single-model robustness.** The preceding theorems enable the analysis on comparing the certified robustness for an ensemble and a single ML model.

**Corollary 1** (Comparison of Ensemble and Single-Model Robustness). *Given an input $\boldsymbol{x}_0 \in \mathbb{R}^d$ with ground-truth label $y_0 \in [C]$. Suppose we have two $\beta$-smooth base models $\{f_1, f_2\}$, which are both $r$-robust at point $\boldsymbol{x}_0$. For any $\Delta \in [0, 1)$:*

- *(Weighted Ensemble) Define Weighted Ensemble $\mathcal{M}_{\mathrm{WE}}$ by base models $\{f_1, f_2\}$ with weights $\{w_1, w_2\}$. Suppose $\mathcal{M}_{\mathrm{WE}}(\boldsymbol{x}_0) = y_0$. If for any label $y_i \neq y_0$, the base models' smoothness $\beta \leq \Delta \cdot \min\{f_1^{y_0/y_i}(\boldsymbol{x}_0), f_2^{y_0/y_i}(\boldsymbol{x}_0)\}/(c^2 r^2)$, and the gradient cosine similarity $\cos\langle \nabla_{\boldsymbol{x}} f_1^{y_0/y_i}(\boldsymbol{x}_0), \nabla_{\boldsymbol{x}} f_2^{y_0/y_i}(\boldsymbol{x}_0)\rangle \leq \cos\theta$, then the $\mathcal{M}_{\mathrm{WE}}$ is at least $R$-robust at $\boldsymbol{x}_0$ with*

$$R = r \cdot \frac{1 - \Delta}{1 + \Delta}\left(1 - C_{\mathrm{WE}}(1 - \cos\theta)\right)^{-1/2}, \text{where} \tag{8}$$

$$C_{\mathrm{WE}} = \min_{y_i : y_i \neq y_0} \frac{2 w_1 w_2 f_1^{y_0/y_i}(\boldsymbol{x}_0) f_2^{y_0/y_i}(\boldsymbol{x}_0)}{(w_1 f_1^{y_0/y_i}(\boldsymbol{x}_0) + w_2 f_2^{y_0/y_i}(\boldsymbol{x}_0))^2}, c = \max\{\tfrac{1-\Delta}{1+\Delta}\left(1 - C_{\mathrm{WE}}(1 - \cos\theta)\right)^{-1/2}, 1\}.$$

- *(Max-Margin Ensemble) Define Max-Margin Ensemble $\mathcal{M}_{\mathrm{MME}}$ by base models $\{f_1, f_2\}$. Suppose $\mathcal{M}_{\mathrm{MME}}(\boldsymbol{x}_0) = y_0$. If for any label $y_1 \neq y_0$ and $y_2 \neq y_0$, the base models' smoothness $\beta \leq \Delta \cdot \min\{f_1^{y_0/y_1}(\boldsymbol{x}_0), f_2^{y_0/y_2}(\boldsymbol{x}_0)\}/(c^2 r^2)$, and the gradient cosine similarity $\cos\langle \nabla_{\boldsymbol{x}} f_1^{y_0/y_1}(\boldsymbol{x}_0), \nabla_{\boldsymbol{x}} f_2^{y_0/y_2}(\boldsymbol{x}_0)\rangle \leq \cos\theta$, then the $\mathcal{M}_{\mathrm{MME}}$ is at least $R$-robust at $\boldsymbol{x}_0$ with*

$$R = r \cdot \frac{1 - \Delta}{1 + \Delta}\left(1 - C_{\mathrm{MME}}(1 - \cos\theta)\right)^{-1/2}, \text{where} \tag{9}$$

$$C_{\mathrm{MME}} = \min_{y_1, y_2 : y_1, y_2 \neq y_0} \frac{2 f_1^{y_0/y_1}(\boldsymbol{x}_0) f_2^{y_0/y_2}(\boldsymbol{x}_0)}{(f_1^{y_0/y_1}(\boldsymbol{x}_0) + f_2^{y_0/y_2}(\boldsymbol{x}_0))^2}, c = \max\{\tfrac{1-\Delta}{1+\Delta}\left(1 - C_{\mathrm{MME}}(1 - \cos\theta)\right)^{-1/2}, 1\}.$$

The above corollary reveals the connection between an ensemble and single-model robustness. As we can see, when $\cos\theta < 1 - \frac{4\Delta}{C(1+\Delta)^2}$ ($C$ is either $C_{\mathrm{WE}}$ or $C_{\mathrm{MME}}$), $R > r$. When the base models are smooth enough ($\beta \to 0^+$ so $\Delta \to 0^+$), in Equations (8) and (9), the $RHS \to 1^-$. As long as the cosine similarity of base models' gradients are not close to 1, this condition can be easily satisfied, i.e., the ensemble models achieve higher certified robustness than base models. Furthermore, the diversity of gradients measured by cosine similarity is important for improving ensemble robustness. We defer the proof to Appendix B.4. Next, we discuss the implications of our theoretical analysis.

**Key factors for the certified robustness of an ensemble.** We observe that smaller magnitude of joint gradients (e.g. $\|\sum_{j=1}^N w_j \nabla_{\boldsymbol{x}} f_j^{y_0/y_i}(\boldsymbol{x}_0)\|_2$ (Theorem 2) or $\|\nabla_{\boldsymbol{x}} f_1^{y_0/y_1}(\boldsymbol{x}_0) + \nabla_{\boldsymbol{x}} f_2^{y_0/y_2}(\boldsymbol{x}_0)\|_2$ (Theorem 3)) indicates smaller LHS in Equations (4) to (7). Since these robustness condition has the form $LHS \leq RHS$, it implies that the robustness condition is easier to be satisfied for current radius $r$, i.e., the certified robust radius $r$ could be improved. Therefore, smaller magnitude of joint gradients leads to higher certified ensemble robustness.

Inspired from low of cosines: for any two vectors $\boldsymbol{a}, \boldsymbol{b}$, $\|\boldsymbol{a} + \boldsymbol{b}\|_2 = \left(\|\boldsymbol{a}\|_2^2 + \|\boldsymbol{b}\|_2^2 + 2\|\boldsymbol{a}\|_2\|\boldsymbol{b}\|_2 \cos\langle \boldsymbol{a}, \boldsymbol{b}\rangle\right)^{1/2}$, the smaller magnitude of joint gradients can be achieved by smaller gradient magnitude of base models or larger diversity (in terms of smaller

cosine similarity) between the gradient of base models. Therefore, *constraining the magnitude of joint gradient is equivalent to improving gradient diversity and reducing base models' gradient magnitude, and they both contribute to improved ensemble robustness.*

We can also observe that large confidence margins, such as $\sum_{j=1}^{N} w_j f_j^{y_0/y_i}(\boldsymbol{x}_0)$ (Theorem 2) and $f_1^{y_0/y_1}(\boldsymbol{x}_0) + f_2^{y_0/y_2}(\boldsymbol{x}_0)$ (Theorem 3), directly lead to larger RHS in Equations (4) to (7). It again implies that the robustness condition becomes easier to be satisfied and larger robust radius $r$ can be achieved. Thus, *increasing confidence margins can lead to higher ensemble robustness.*

**Comparison between ensemble and single-model robustness.** As the discussion following Corollary 1 reveals, when the base models are smooth enough, the ensemble model is more robust than the base models. Moreover, we prove that certified robustness of ensembles is positively correlated with the base model (gradient) diversity, which is aligned with existing empirical observations (Tramèr et al., 2017; Pang et al., 2019).

## 2.2 DIVERSITY-REGULARIZED TRAINING

Inspired by the above key factors for the certified ensemble robustness, we propose the **D**iversity-**R**egularized **T**raining. In particular, let $\boldsymbol{x}_0$ be a training sample, DRT contains the following two regularization terms in the objective function to minimize:

- Gradient Diversity Loss (GD Loss):

$$\mathcal{L}_{\mathrm{GD}}(\boldsymbol{x}_0)_{ij} = \left\| \nabla_{\boldsymbol{x}} f_i^{y_0/y_i^{(2)}}(\boldsymbol{x}_0) + \nabla_{\boldsymbol{x}} f_j^{y_0/y_j^{(2)}}(\boldsymbol{x}_0) \right\|_2. \tag{10}$$

- Confidence Margin Loss (CM Loss):

$$\mathcal{L}_{\mathrm{CM}}(\boldsymbol{x}_0)_{ij} = f_i^{y_i^{(2)}/y_0}(\boldsymbol{x}_0) + f_j^{y_j^{(2)}/y_0}(\boldsymbol{x}_0). \tag{11}$$

In Equations (10) and (11), $y_0$ is the ground-truth label of $\boldsymbol{x}_0$, and $y_i^{(2)}$ (or $y_j^{(2)}$) is the runner-up class of base model $F_i$ (or $F_j$). Intuitively, for each model pair $(i, j)$ where $i, j \in [N]$ and $i \neq j$, the GD Loss encourages the joint gradient, i.e., gradient vector sum between model $i$ and $j$, to be small. Note that the gradient computed here is actually the gradient difference between different labels. As our theorems reveal, it is the gradient difference between different labels instead of pure gradient itself that matters, which improves previous understanding of gradient diversity (Pang et al., 2019; Demontis et al., 2019). The GD Loss encourages both large gradient diversity and small base models' gradient magnitude in a naturally balanced way, and encodes the interplay between gradient magnitude and direction diversity. Compared with GD Loss, solely regularizing the base models' gradient would hurt the model's benign accuracy, and solely regularizing gradient diversity is hard to realize due to the boundedness of cosine similarity.

The CM Loss encourages the large margin between the true and runner-up classes for base models. Both regularization terms are directly motivated by our analysis, and the detailed implementation process can be found in Section 4.

## 3 ROBUSTNESS FOR SMOOTHED ML ENSEMBLES

To compute the certified robustness for different ensemble models based on Theorems 2 and 3, we need to ensure the model smoothness which is challenging. Thus, in this section, we apply an adapted randomized model smoothing to compute certified robustness for general ensembles based on our conditions. We focus on Ensemble Before Smoothing (EBS) strategy: first construct the ensemble $\mathcal{M}$ from base models, then smooth $\mathcal{M}$'s prediction. $\mathcal{M}$ could be either $\mathcal{M}_{\mathrm{WE}}$ or $\mathcal{M}_{\mathrm{MME}}$. We also consider another strategy as smoothing base models first then ensemble. The analysis of these two strategies which proves EBS is more robust is deferred to Appendix C.

## 3.1 CERTIFIED ROBUSTNESS OF ENSEMBLES VIA RANDOMIZED SMOOTHING

To derive the certified robustness of both MME and WE, we first define the statistical robustness and confidence for the single model and ensemble models.

**Definition 5** (($\boldsymbol{\epsilon}$, $p$)-Statistical Robust). Given a random variable $\boldsymbol{\epsilon}$ and model $F : \mathbb{R}^d \mapsto [C]$, at point $\boldsymbol{x}_0$ with ground truth label $y_0$, we call $F$ is ($\boldsymbol{\epsilon}$, $p$)-statistical robust if $\mathrm{Pr}_{\boldsymbol{\epsilon}}(F(\boldsymbol{x}_0 + \boldsymbol{\epsilon}) = y_0) \geq p$.

Note that based on Theorem B.1, when $\boldsymbol{\epsilon} \sim \mathcal{N}(0, \sigma^2 \boldsymbol{I}_d)$, if $F$ is $(\boldsymbol{\epsilon}, p)$-statistical robust at point $\boldsymbol{x}_0$, the smoothed model $G_F^{\boldsymbol{\epsilon}}$ over $F$ is $(\sigma \Phi^{-1}(p))$-robust at point $\boldsymbol{x}_0$.

**Definition 6** (($\boldsymbol{\epsilon}, \lambda, p$)-WE Confident). Let $\mathcal{M}_{\mathrm{WE}}$ be Weighted Ensemble defined over base models $\{f_i\}_{i=1}^N$ with weights $\{w_i\}_{i=1}^N$. If at point $\boldsymbol{x}_0$ with ground-truth $y_0$ and random variable $\boldsymbol{\epsilon}$, we have

$$\Pr_{\boldsymbol{\epsilon}} \left( \max_{y_j \in [C] : y_j \neq y_0} \left( \sum_{i=1}^N w_i f_i(\boldsymbol{x}_0 + \boldsymbol{\epsilon})_{y_j} \right) \leq \lambda \sum_{i=1}^N w_i \left(1 - f_i(\boldsymbol{x}_0 + \boldsymbol{\epsilon})_{y_0}\right) \right) = 1 - p, \qquad (12)$$

we call Weighted Ensemble $\mathcal{M}_{\mathrm{WE}}$ ($\boldsymbol{\epsilon}, \lambda, p$)-WE confident at point $\boldsymbol{x}_0$.

**Definition 7** (($\boldsymbol{\epsilon}, \lambda, p$)-MME Confident). Let $\mathcal{M}_{\mathrm{MME}}$ be a Max-Margin Ensemble over $\{f_i\}_{i=1}^N$. If at point $\boldsymbol{x}_0$ with ground-truth $y_0$ and random variable $\boldsymbol{\epsilon}$, we have

$$\Pr_{\boldsymbol{\epsilon}} \left( \bigwedge_{i \in [N]} \left( \max_{y_j \in [C] : y_j \neq y_0} f_i(\boldsymbol{x}_0 + \boldsymbol{\epsilon})_{y_j} \leq \lambda (1 - f_i(\boldsymbol{x}_0 + \boldsymbol{\epsilon})_{y_0}) \right) \right) = 1 - p, \qquad (13)$$

we call Max-Margin Ensemble $\mathcal{M}_{\mathrm{MME}}$ ($\boldsymbol{\epsilon}, \lambda, p$)-MME confident at point $\boldsymbol{x}_0$.

Note that the confidence of every single model lies in the probability simplex, and $\lambda$ reflects the confidence portion that a wrong prediction class can take beyond the true class $(1 - f_i(\boldsymbol{x}_0 + \boldsymbol{\epsilon}))$. Now we are ready to present the certified robustness for different ensemble models.

**Theorem 4** (Certified Robustness for WE). *Let $\boldsymbol{\epsilon}$ be a random variable supported on $\mathbb{R}^d$. Let $\mathcal{M}_{\mathrm{WE}}$ be a Weighted Ensemble defined over $\{f_i\}_{i=1}^N$ with weights $\{w_i\}_{i=1}^N$. The $\mathcal{M}_{\mathrm{WE}}$ is $(\boldsymbol{\epsilon}, \lambda_1, p)$-WE confident. Let $\boldsymbol{x}_0 \in \mathbb{R}^d$ be the input with ground-truth label $y_0 \in [C]$. Assume $\{f_i(\boldsymbol{x}_0 + \boldsymbol{\epsilon})_{y_0}\}_{i=1}^N$, the confidence scores across base models for label $y_0$, are i.i.d. and follow symmetric distribution with mean $\mu$ and variance $s^2$, where $\mu > (1 + \lambda_1^{-1})^{-1}$. We have*

$$\Pr_{\boldsymbol{\epsilon}}(\mathcal{M}_{\mathrm{WE}}(\boldsymbol{x}_0 + \boldsymbol{\epsilon}) = y_0) \geq 1 - p - \frac{\|\boldsymbol{w}\|_2^2}{\|\boldsymbol{w}\|_1^2} \cdot \frac{s^2}{2 \left( \mu - \left(1 + \lambda_1^{-1}\right)^{-1} \right)^2}. \qquad (14)$$

**Theorem 5** (Certified Robustness for MME). *Let $\boldsymbol{\epsilon}$ be a random variable supported on $\mathbb{R}^d$. Let $\mathcal{M}_{\mathrm{MME}}$ be a Max-Margin Ensemble defined over $\{f_i\}_{i=1}^N$. The $\mathcal{M}_{\mathrm{MME}}$ is $(\boldsymbol{\epsilon}, \lambda_2, p)$-MME confident. Let $\boldsymbol{x}_0 \in \mathbb{R}^d$ be the input with ground-truth label $y_0 \in [C]$. Assume $\{f_i(\boldsymbol{x}_0 + \boldsymbol{\epsilon})_{y_0}\}_{i=1}^N$, the confidence scores across base models for label $y_0$, are i.i.d. and follow symmetric distribution with mean $\mu$ where $\mu > (1 + \lambda_2^{-1})^{-1}$. Define $s_f^2 = \mathrm{Var}(\min_{i \in [N]} f_i(\boldsymbol{x}_0 + \boldsymbol{\epsilon})_{y_0})$. We have*

$$\Pr_{\boldsymbol{\epsilon}}(\mathcal{M}_{\mathrm{MME}}(\boldsymbol{x}_0 + \boldsymbol{\epsilon}) = y_0) \geq 1 - p - \frac{s_f^2}{2 \left( \mu - \left(1 + \lambda_2^{-1}\right)^{-1} \right)^2}. \qquad (15)$$

The condition $\mu > 1/(1 + \lambda^{-1})$ guarantees normal performance of a model, which is the sufficient condition for the standard setup $p_A > p_B$ as in (Cohen et al., 2019)). For comparison, we also derive certified robustness for a single model in Proposition D.1. We defer the proofs to Appendix D.1. Based on our theoretical analysis above, we draw additional implications on the connections between the certified robustness and different losses. For **Confidence Margin Loss**, which aims at increasing the confidence margin of ensembles by enlarging that of base models, from Theorems 4 and 5, we can see that small $\lambda$ ($\lambda_1$ in WE and $\lambda_2$ in MME) results in large $\Pr(\mathcal{M}(\boldsymbol{x}_0 + \boldsymbol{\epsilon}) = y_0)$, i.e., large certified robustness. For **Standard Training Loss**, which increases base models' confidence of true class, we can view it as increasing the average confidence score, $\mu$, and its effectiveness is revealed from Theorems 4 and 5.

### 3.2 COMPARISON FOR THE CERTIFIED ROBUSTNESS OF ENSEMBLES

The unified form of certified robustness above allows us to compare it for different ensembles.

**Corollary 2** (Comparison of Certified Robustness). *Let $\boldsymbol{\epsilon}$ be a random variable supported on $\mathbb{R}^d$. Over base models $\{f_i\}_{i=1}^N$, let $\mathcal{M}_{\mathrm{MME}}$ be Max-Margin Ensemble, and $\mathcal{M}_{\mathrm{WE}}$ the Weighted Ensemble with weights $\{w_i\}_{i=1}^N$. Let $\boldsymbol{x}_0 \in \mathbb{R}^d$ be the input with ground-truth label $y_0 \in [C]$. Assume $\{f_i(\boldsymbol{x}_0 + \boldsymbol{\epsilon})_{y_0}\}_{i=1}^N$, the confidence scores across base models for label $y_0$, are i.i.d. and follow symmetric distribution with mean $\mu$ and variance $s^2$, where $\mu > \max\{(1 + \lambda_1^{-1})^{-1}, (1 + \lambda_2^{-1})^{-1}\}$. Define $s_f^2 = \mathrm{Var}(\min_{i \in [N]} f_i(\boldsymbol{x}_0 + \boldsymbol{\epsilon})_{y_0})$ and assume $s_f < s$.*

- *When*

$$\frac{\lambda_1}{\lambda_2} < \lambda_2^{-1} \left( \left( \frac{s}{s_f} \left( \mu - \left( 1 + \lambda_2^{-1} \right)^{-1} \right) + 1 - \mu \right)^{-1} - 1 \right),$$ (16)

*for any weights $\{w_i\}_{i=1}^N$, $\mathcal{M}_{\mathrm{WE}}$ has higher certified robustness than $\mathcal{M}_{\mathrm{MME}}$.*

- *When*

$$\frac{\lambda_1}{\lambda_2} > \lambda_2^{-1} \left( \left( \frac{s}{\sqrt{N} s_f} \left( \mu - \left( 1 + \lambda_2^{-1} \right)^{-1} \right) + 1 - \mu \right)^{-1} - 1 \right),$$ (17)

*for any weights $\{w_i\}_{i=1}^N$, $\mathcal{M}_{\mathrm{MME}}$ has higher certified robustness than $\mathcal{M}_{\mathrm{WE}}$.*

*Here, the certified robustness is given by Theorems 4 and 5.*

Appendix D.2 entails the detailed proofs. Note that given $\lambda_1$ is the weighted average and $\lambda_2$ the maximum over $\lambda$'s of all base models, $\lambda_1/\lambda_2$ reflects the adversarial transferability (Papernot et al., 2016a) among base models under the same $p$: If the transferability is high the confidence scores of base models are similar ($\lambda$'s are similar), and thus $\lambda_1$ is large resulting in large $\lambda_1/\lambda_2$. On the other hand, when the transferability is low, the confidence scores are diverse ($\lambda$'s are diverse), and thus $\lambda_1$ is small resulting in small $\lambda_1/\lambda_2$. Based on our theoretical analysis we can see that MME is more robust when the transferability is high; WE is more robust when the transferability is low. In Appendix D.3, we also prove that under certain distribution of $f_i(\boldsymbol{x}_0 + \boldsymbol{\epsilon})_{y_0}$, when $N$ is sufficiently large, the MME always more robust. Appendix D.4 entails the numerical evaluations.

## 4 EXPERIMENTAL EVALUATION

In order to make a fair comparison with existing work (Cohen et al., 2019; Salman et al., 2019), we evaluate our approach on different datasets: MNIST (LeCun et al., 2010), CIFAR-10 (Krizhevsky, 2012), and ImageNet (Deng et al., 2009). We show that by training MME/WE with DRT, our model can achieve the state-of-the-art certified robustness.

### 4.1 EXPERIMENTAL SETUP

**Baselines**: We mainly consider two state-of-the-art baselines for certified robustness: 1) Gaussian smoothing (Cohen et al., 2019), which trains a smoothed classifier by applying Gaussian augmentation. 2) SmoothAdv (Salman et al., 2019), which integrates adversarial training on the soft approximation. The comparison with more baselines can be found in Appendix E.4.

**Model structures**: For each base model in our ensemble, we follow the same configuration of the baselines: LeNet (LeCun et al., 1998) for MNIST, ResNet-110 and ResNet-50 (He et al., 2016) for CIFAR-10 and ImageNet datasets.

**Training:** We smooth the $N$ base models of an ensemble following the baselines (Cohen et al., 2019; Salman et al., 2019). For each input $\boldsymbol{x}_0$ with ground truth $y_0$, we use $\boldsymbol{x}_0 + \boldsymbol{\epsilon}$ with $\boldsymbol{\epsilon} \sim \mathcal{N}(0, \sigma^2 \boldsymbol{I}_d)$ as training input for each base model. We call two base models $(f_i, f_j)$ valid model pair at $(\boldsymbol{x}_0, y_0)$ if both $F_i(\boldsymbol{x}_0 + \boldsymbol{\epsilon})$ and $F_j(\boldsymbol{x}_0 + \boldsymbol{\epsilon})$ predict $y_0$. For every valid model pair, we apply GD Loss and CM Loss with $\rho_1$ and $\rho_2$ as the weight parameters. The final training loss of an ensemble is as below:

$$L = \sum_{i \in [N]} L_{\mathrm{std}}(\boldsymbol{x}_0 + \boldsymbol{\epsilon}, y_0)_i + \rho_1 \sum_{\substack{i,j \in [N], i \neq j \\ F_i(\boldsymbol{x}_0+\boldsymbol{\epsilon})=y_0 \\ F_j(\boldsymbol{x}_0+\boldsymbol{\epsilon})=y_0}} \mathcal{L}_{\mathrm{GD}}(\boldsymbol{x}_0 + \boldsymbol{\epsilon})_{ij} + \rho_2 \sum_{\substack{i,j \in [N], i \neq j \\ F_i(\boldsymbol{x}_0+\boldsymbol{\epsilon})=y_0 \\ F_j(\boldsymbol{x}_0+\boldsymbol{\epsilon})=y_0}} \mathcal{L}_{\mathrm{CM}}(\boldsymbol{x}_0 + \boldsymbol{\epsilon})_{ij}.$$ (18)

The standard training loss $L_{\mathrm{std}}(\boldsymbol{x}_0 + \boldsymbol{\epsilon}, y_0)_i$ of each base model $f_i$ is either cross-entropy loss in (Cohen et al., 2019; Yang et al., 2020), or adversarial training loss in (Salman et al., 2019). We leave more training details in Appendix E.

**Robustness Certification:** During certification, we apply the MME or WE ensemble protocols over the trained base models $\{f_i\}_{i=1}^N$ to obtain ensemble $\mathcal{M}$, then smooth $\mathcal{M}$ with noise $\boldsymbol{\epsilon} \sim \mathcal{N}(0, \sigma^2 \boldsymbol{I}_d)$. We report the standard *certified accuracy* under different $\mathcal{L}_2$ radius $r$ as our evaluation metric (Cohen et al., 2019) (more implementation details in Appendix E).

Table 1: The certified accuracy under different radius $r$ for MNIST dataset.

| Radius $r$ | 0.00 | 0.25 | 0.50 | 0.75 | 1.00 | 1.25 | 1.50 | 1.75 | 2.00 | 2.25 | 2.50 |
|---|---|---|---|---|---|---|---|---|---|---|---|
| Gaussian (Cohen et al., 2019) | 99.1 | 97.9 | 96.6 | 94.7 | 90.0 | 83.0 | 68.2 | 46.6 | 33.0 | 20.5 | 11.5 |
| SmoothAdv (Salman et al., 2019) | 99.1 | 98.4 | 97.0 | 96.3 | 93.0 | 87.7 | 80.2 | 66.3 | 43.2 | 34.3 | 24.0 |
| MME (Gaussian) | 99.2 | 98.4 | 96.8 | 94.9 | 90.5 | 84.3 | 69.8 | 48.8 | 34.7 | 23.4 | 12.7 |
| DRT + MME (Gaussian) | **99.5** | **98.6** | 97.5 | 95.5 | 92.6 | 86.8 | 75.2 | 55.8 | 44.8 | 38.0 | 27.0 |
| MME (SmoothAdv) | 99.2 | 98.2 | 97.3 | 96.4 | 93.1 | 88.1 | 80.6 | 67.9 | 44.8 | 35.0 | 25.2 |
| DRT + MME (SmoothAdv) | 99.2 | 98.4 | **97.6** | **96.7** | 93.1 | **88.5** | 83.2 | 68.9 | 48.2 | 39.2 | **33.5** |
| WE (Gaussian) | 99.2 | 98.4 | 96.9 | 94.9 | 90.6 | 84.5 | 70.4 | 49.0 | 35.2 | 23.7 | 12.9 |
| DRT + WE (Gaussian) | **99.5** | **98.6** | 97.4 | 95.6 | 92.6 | 86.9 | 75.4 | 55.8 | 44.6 | 38.2 | 27.2 |
| WE (SmoothAdv) | 99.1 | 98.2 | 97.4 | 96.4 | **93.4** | 88.2 | 81.1 | 67.9 | 44.7 | 35.2 | 24.9 |
| DRT + WE (SmoothAdv) | 99.1 | 98.4 | **97.6** | **96.7** | **93.4** | **88.5** | 83.3 | 69.6 | 48.3 | 39.4 | 33.5 |

Table 2: The certified accuracy under different radius $r$ for CIFAR-10 dataset.

| Radius $r$ | 0.00 | 0.25 | 0.50 | 0.75 | 1.00 | 1.25 | 1.50 | 1.75 | 2.00 |
|---|---|---|---|---|---|---|---|---|---|
| Gaussian (Cohen et al., 2019) | 78.9 | 64.4 | 47.4 | 33.7 | 23.1 | 18.3 | 13.6 | 10.5 | 7.3 |
| SmoothAdv (Salman et al., 2019) | 68.9 | 61.0 | 54.4 | 45.7 | 34.8 | 28.5 | 21.9 | 18.2 | 15.7 |
| MME (Gaussian) | 80.8 | 68.2 | 53.4 | 38.4 | 29.0 | 19.6 | 15.6 | 11.6 | 8.8 |
| DRT + MME (Gaussian) | 81.4 | **70.4** | 57.8 | 43.8 | 31.6 | 26.2 | 22.4 | 18.8 | 16.6 |
| MME (SmoothAdv) | 71.4 | 64.5 | 57.6 | 48.4 | 36.2 | 29.8 | 23.9 | 19.5 | 16.2 |
| DRT + MME (SmoothAdv) | 72.6 | 67.2 | **60.2** | 50.4 | 39.4 | 35.8 | **30.4** | **24.0** | 20.1 |
| WE (Gaussian) | 80.8 | 68.4 | 53.6 | 38.4 | 29.2 | 19.7 | 15.9 | 11.8 | 8.9 |
| DRT + WE (Gaussian) | **81.5** | **70.4** | 57.9 | 44.0 | 31.7 | 26.2 | 22.5 | 19.1 | 16.8 |
| WE (SmoothAdv) | 71.8 | 64.6 | 57.8 | 48.5 | 36.2 | 29.6 | 24.2 | 19.6 | 16.0 |
| DRT + WE (SmoothAdv) | 72.6 | 67.0 | **60.2** | **50.5** | **39.5** | **36.0** | 30.3 | 24.1 | **20.3** |

## 4.2 EXPERIMENTAL RESULTS

In our experiments, we consider ensemble models consisting of three base models on MNIST, CIFAR-10, and ImageNet datasets. We observe that when training MME or WE (using different base models) with DRT, they can achieve the state-of-the-art certified robustness.

The evaluation results on MNIST are shown in Table 1. We observe that the certified accuracy can be improved slightly by applying the MME or WE compared with a single base model (aligned with Corollary 1). After training with DRT, the improvements become significant and the DRT-trained ensemble models can achieve the highest certified accuracy under every radius $r$. In particular, DRT ensemble model can surpass the base model's certified accuracy around 7% under large radius $r = 2.50$. We further compare the certified robustness of WE and MME in Appendix D.4.

On CIFAR-10 the evaluation results are shown in Table 2. Similarly, we can see that the DRT-based ensemble model can achieve the best certified robustness under different radius $r$ (more experimental details are in Appendix E.2). Note that the DRT-based ensemble with Gaussian smoothed base models can achieve comparable results to SmoothAdv with less training time (detailed efficiency analysis is in Appendix E.2). We defer the results on ImageNet to Appendix E.3 and put all discussions about hyper-parameters under different settings in Appendix E.

## 5 CONCLUSION

In this paper, we explore and characterize the robustness conditions for ensemble ML models theoretically, and propose DRT for training a robust ensemble in practice. Our analysis provides the justification of the regularization-based training approach DRT, as well as why an ensemble model could have higher robustness compared with a single model. Especially, we show that smaller magnitude of joint gradients, and the large confidence margins are the key factors that contribute to high certified robustness of an ensemble. We further compare the certified robustness of two types of ensembles: Weighted Ensemble and Max-Margin Ensemble under the randomized smoothing regime. Extensive experiments show that the ensemble models trained with DRT can achieve higher certified robustness than existing approaches.

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

Table 3: Main Theoretical Results.

|  |  | Weighted Ensemble | Max-Margin Ensemble | Single Model |
|---|---|---|---|---|
| General Robustness (Section 2) | General Condition | Proposition 1 | Theorem 1 | Fact B.1 |
|  | Gradient and Confidence Margin Condition | Theorem 2 | Theorem 3 | Proposition B.1 |
|  | Comparison | Corollary 1 | | |
| Robustness with Rand. Smooth. (Section 3) | Certified Robustness | Theorem 4 | Theorem 5 | Proposition D.1 |
|  | Comparison | Corollary 2 | | Appendix D.4 |

## A  TABLE OF THEORETICAL RESULTS

For a quick index for the theoretical results, we refer the readers to Table 3.

## B  FORMAL DEFINITIONS AND PROOFS OF ROBUSTNESS CONDITIONS

In this appendix, we discuss the connection between $r$-robustness and certified robustness given by randomized smoothing, and the detailed proofs of the robustness conditions in Section 2.

### B.1  $r$-ROBUSTNESS AND RANDOMIZED SMOOTHING

In this subsection, we discuss the connection between $r$-robustness and the certified robustness given by randomized smoothing (Cohen et al., 2019).

In randomized smoothing, each input's prediction is given by the most probable prediction after adding noise. Formally, let $\epsilon \sim \mathcal{N}(0, \sigma^2 \boldsymbol{I}_d)$ be a Gaussian random variable, from model prediction $F$, we can define the *smoothed classifier* $G_F^{\epsilon} : \mathbb{R}^d \mapsto [C]$ where $G_F^{\epsilon}(\boldsymbol{x}) = \arg\max_{j \in [C]} g_F^{\epsilon}(\boldsymbol{x})_j$:

$$g_F^{\epsilon}(\boldsymbol{x})_j := \mathop{\mathbb{E}}_{\epsilon \sim \mathcal{N}(0, \sigma^2 \boldsymbol{I}_d)} \mathbf{1}[F(\boldsymbol{x} + \epsilon) = j] = \Pr_{\epsilon \sim \mathcal{N}(0, \sigma^2 \boldsymbol{I}_d)} (F(\boldsymbol{x} + \epsilon) = j).$$

Intuitively, the confidence score for each class is given by the probability of predicting that class under noised inputs.

**Theorem B.1** (Simplified; Certified Robustness via Randomized Smoothing; Cohen et al. (2019))**.** *At point $\boldsymbol{x}_0$, let $\epsilon \sim \mathcal{N}(0, \sigma^2 \boldsymbol{I}_d)$, a smoothed model $G_F^{\epsilon}$ is $r$-robust, where*

$$r := \sigma \Phi^{-1}(g_F^{\epsilon}(\boldsymbol{x}_0)_{G_F^{\epsilon}(\boldsymbol{x}_0)}), \tag{19}$$

*and $\Phi^{-1}$ is the inverse function of Gaussian CDF.*

We remark that a tighter certified radius is $r' := \frac{\sigma}{2} \left( \Phi^{-1}(g_F^{\epsilon}(\boldsymbol{x}_0)_{G_F^{\epsilon}(\boldsymbol{x}_0)}) - \Phi^{-1}(g_F^{\epsilon}(\boldsymbol{x}_0))_{G_F^{\epsilon(2)}(\boldsymbol{x}_0)} \right) \geq r$, while for the ease of sampling, the Equation (19) is used more often in the literature. In Sections 3 and 4, we use Equation (19) for analysis and empirical evaluation, and these results can be generalized to the tighter radius $r'$ easily.

### B.2  GENERAL ROBUSTNESS CONDITIONS

**Proposition 1** (Robustness Condition for WE)**.** *Consider an input $\boldsymbol{x}_0 \in \mathbb{R}^d$ with ground-truth label $y_0 \in [C]$, and an ensemble model $\mathcal{M}_{\mathrm{WE}}$ constructed by base models $\{f_i\}_{i=1}^N$ with weights $\{w_i\}_{i=1}^N$. Suppose $\mathcal{M}_{\mathrm{WE}}(\boldsymbol{x}_0) = y_0$. Then, the ensemble $\mathcal{M}_{\mathrm{WE}}$ is $r$-robust at point $\boldsymbol{x}_0$ if and only if for any $\boldsymbol{x} \in \{\boldsymbol{x}_0 + \boldsymbol{\delta} : \|\boldsymbol{\delta}\|_2 \leq r\}$,*

$$\min_{y_i \in [C]: y_i \neq y_0} \sum_{j=1}^N w_j f_j^{y_0/y_i}(\boldsymbol{x}) \geq 0. \tag{20}$$

*Proof of Proposition 1.* According the the definition of $r$-robust, we know $\mathcal{M}_{\mathrm{WE}}$ is $r$-robust if and only if for any point $\boldsymbol{x} := \boldsymbol{x}_0 + \boldsymbol{\delta}$ where $\|\boldsymbol{\delta}\|_2 \leq r$, $\mathcal{M}_{\mathrm{WE}}(\boldsymbol{x}_0 + \boldsymbol{\delta}) = y_0$, which means that for any

other label $y_i \neq y_0$, the confidence score for label $y_0$ is larger or equal than the confidence score for label $y_i$. It means that

$$\sum_{j=1}^{N} w_j f_j(\boldsymbol{x})_{y_0} \geq \sum_{j=1}^{N} w_j f_j(\boldsymbol{x})_{y_i}$$

for any $\boldsymbol{x} \in \{\boldsymbol{x}_0 + \boldsymbol{\delta} : \|\boldsymbol{\delta}\|_2 \leq r\}$. Since this should hold for any $y_i \neq y_0$, we have the necessary and sufficient condition

$$\min_{y_i \in [C]: y_i \neq y_0} \sum_{j=1}^{N} w_j f_j^{y_0/y_i}(\boldsymbol{x}) \geq 0. \tag{20}$$

$\square$

**Theorem 1** (Robustness Condition for MME). *Consider an input $\boldsymbol{x}_0 \in \mathbb{R}^d$ with ground-truth label $y_0 \in [C]$. Let $\mathcal{M}_{\mathrm{MME}}$ be an MME defined over base models $\{f_i\}_{i=1}^N$. Suppose: (1) $\mathcal{M}_{\mathrm{MME}}(\boldsymbol{x}_0) = y_0$; (2) for any $\boldsymbol{x} \in \{\boldsymbol{x}_0 + \boldsymbol{\delta} : \|\boldsymbol{\delta}\|_2 \leq r\}$, given any base model $i \in [N]$, either $F_i(\boldsymbol{x}) = y_0$ or $F_i^{(2)}(\boldsymbol{x}) = y_0$. Then, the ensemble $\mathcal{M}_{\mathrm{MME}}$ is r-robust at point $\boldsymbol{x}_0$ if and only if for any $\boldsymbol{x} \in \{\boldsymbol{x}_0 + \boldsymbol{\delta} : \|\boldsymbol{\delta}\|_2 \leq r\}$,*

$$\max_{i \in [N]} \min_{y_i \in [C]: y_i \neq y_0} f_i^{y_0/y_i}(\boldsymbol{x}) \geq \max_{i \in [N]} \min_{y_i' \in [C]: y_i' \neq y_0} f_i^{y_i'/y_0}(\boldsymbol{x}). \tag{3}$$

The theorem states the sufficient and necessary robustness condition for MME. We divide the two directions into the following two lemmas and prove them separately. We mainly use the alternative form of Equation (3) as such in the following lemmas and their proofs:

$$\max_{i \in [N]} \min_{y_i \in [C]: y_i \neq y_0} f_i^{y_0/y_i}(\boldsymbol{x}) + \min_{i \in [N]} \min_{y_i' \in [C]: y_i' \neq y_0} f_i^{y_0/y_i'}(\boldsymbol{x}) \geq 0. \tag{3}$$

**Lemma B.1** (Sufficient Condition for MME). *Let $\mathcal{M}_{\mathrm{MME}}$ be an MME defined over base models $\{f_i\}_{i=1}^N$. For any input $\boldsymbol{x}_0 \in \mathbb{R}^d$, the Max-Margin Ensemble $\mathcal{M}_{\mathrm{MME}}$ predicts $\mathcal{M}_{\mathrm{MME}}(\boldsymbol{x}_0) = y_0$ if*

$$\max_{i \in [N]} \min_{y_i \in [C]: y_i \neq y_0} f_i^{y_0/y_i}(\boldsymbol{x}_0) + \min_{i \in [N]} \min_{y_i' \in [C]: y_i' \neq y_0} f_i^{y_0/y_i'}(\boldsymbol{x}_0) \geq 0. \tag{3}$$

*Proof of Lemma B.1.* For brevity, for $i \in [N]$, we denote $y_i := F_i(\boldsymbol{x}_0), y_i' := F_i^{(2)}(\boldsymbol{x}_0)$ for each base model's top class and runner-up class at point $\boldsymbol{x}_0$.

Suppose $\mathcal{M}_{\mathrm{MME}}(\boldsymbol{x}_0) \neq y_0$, then according to ensemble definition (see Definition 3), there exists $c \in [N]$, such that $\mathcal{M}_{\mathrm{MME}}(\boldsymbol{x}_0) = F_c(\boldsymbol{x}_0) = y_c$, and

$$\forall i \in [N], i \neq c, f_c(\boldsymbol{x}_0)^{y_c/y_c'} > f_i(\boldsymbol{x}_0)^{y_i/y_i'}. \tag{21}$$

Because $y_c \neq y_0$, we have $f_c(\boldsymbol{x}_0)_{y_0} \leq f_c(\boldsymbol{x}_0)_{y_c'}$, so that $f_c(\boldsymbol{x}_0)^{y_c/y_0} \geq f_c(\boldsymbol{x}_0)^{y_c/y_c'}$. Now consider any model $f_i$ where $i \in [N]$, we would like to show that there exists $y^* \neq y_0$, such that $f_i(\boldsymbol{x}_0)^{y_i/y_i'} \geq f_i(\boldsymbol{x}_0)^{y_0/y^*}$:

- If $y_i = y_0$, let $y^* := y_i'$, trivially $f_i(\boldsymbol{x}_0)^{y_i/y_i'} = f_i(\boldsymbol{x}_0)^{y_0/y^*}$;

- If $y_i \neq y_0$, and $y_i' \neq y_0$, we let $y^* := y_i'$, then $f_i(\boldsymbol{x}_0)^{y_i/y_i'} = f_i(\boldsymbol{x}_0)^{y_i/y^*} \geq f_i(\boldsymbol{x}_0)^{y_0/y^*}$;

- If $y_i \neq y_0$, but $y_i' = y_0$, we let $y^* := y_i$, then $f_i(\boldsymbol{x}_0)^{y_i/y_i'} = f_i(\boldsymbol{x}_0)^{y_i/y_0} \geq f_i(\boldsymbol{x}_0)^{y_0/y_i} = f_i(\boldsymbol{x}_0)^{y_0/y^*}$.

Combine the above findings with Equation 21, we have:

$$\forall i \in [N], i \neq c, \exists y_c^* \in [C] \text{ and } y_c^* \neq y_0, \exists y_i^* \in [C] \text{ and } y_i^* \neq y_0, f_c(\boldsymbol{x}_0)^{y_c^*/y_0} > f_i(\boldsymbol{x}_0)^{y_0/y_i^*}.$$

Therefore, its negation

$$\exists i \in [N], i \neq c, \forall y_c^* \in [C] \text{ and } y_c^* \neq y_0, \forall y_i^* \in [C] \text{ and } y_i^* \neq y_0, f_c(\boldsymbol{x}_0)^{y_0/y_c^*} + f_i(\boldsymbol{x}_0)^{y_0/y_i^*} \geq 0 \tag{22}$$

implies $\mathcal{M}(\boldsymbol{x}_0) = y_0$. Since Equation (22) holds for any $y_c^*$ and $y_i^*$, the equation is equivalent to

$$\exists i \in [N],\, i \neq c,\, \min_{y_c \in [C]:y_c \neq y_0} f_c(\boldsymbol{x}_0)^{y_0/y_c}(\boldsymbol{x}_0) + \min_{y_i' \in [C]:y_i' \neq y_0} f_i(\boldsymbol{x}_0)^{y_0/y_i'}(\boldsymbol{x}_0) \geq 0.$$

The existence qualifier over $i$ can be replaced by maximum:

$$\min_{y_c \in [C]:y_c \neq y_0} f_c(\boldsymbol{x}_0)^{y_0/y_c}(\boldsymbol{x}_0) + \max_{i \in [N]} \min_{y_i' \in [C]:y_i' \neq y_0} f_i(\boldsymbol{x}_0)^{y_0/y_i'}(\boldsymbol{x}_0) \geq 0.$$

It is implied by

$$\max_{i \in [N]} \min_{y_i \in [C]:y_i \neq y_0} f_i^{y_0/y_i}(\boldsymbol{x}_0) + \min_{i \in [N]} \min_{y_i' \in [C]:y_i' \neq y_0} f_i^{y_0/y_i'}(\boldsymbol{x}_0) \geq 0. \tag{3}$$

Thus, Equation (3) is a sufficient condition for $\mathcal{M}_{\mathrm{MME}}(\boldsymbol{x}_0) = y_0$. $\qquad\square$

**Lemma B.2** (Necessary Condition for MME). *For any input $\boldsymbol{x}_0 \in \mathbb{R}^d$, if for any base model $i \in [N]$, either $F_i(\boldsymbol{x}_0) = y_0$ or $F_i^{(2)}(\boldsymbol{x}_0) = y_0$, then Max-Margin Ensemble $\mathcal{M}_{\mathrm{MME}}$ predicting $\mathcal{M}_{\mathrm{MME}}(\boldsymbol{x}_0) = y_0$ implies*

$$\max_{i \in [N]} \min_{y_i \in [C]:y_i \neq y_0} f_i^{y_0/y_i}(\boldsymbol{x}_0) + \min_{i \in [N]} \min_{y_i' \in [C]:y_i' \neq y_0} f_i^{y_0/y_i'}(\boldsymbol{x}_0) \geq 0. \tag{3}$$

*Proof of Lemma B.2.* Similar as before, for brevity, for $i \in [N]$, we denote $y_i := F_i(\boldsymbol{x}_0), y_i' := F_i^{(2)}(\boldsymbol{x}_0)$ for each base model's top class and runner-up class at point $\boldsymbol{x}_0$.

Suppose Equation (3) is not satisfied, it means that

$$\exists c \in [N],\, \exists y_c^* \in [C] \text{ and } y_c^* \neq y_0,\, \forall i \in [N],\, \exists y_i^* \in [C] \text{ and } y_i^* \neq y_0,\, f_c^{y_c^*/y_0}(\boldsymbol{x}_0) > f_i^{y_0/y_i^*}(\boldsymbol{x}_0).$$

- If $y_c = y_0$, then $f_c^{y_c^*/y_0}(\boldsymbol{x}_0) \leq 0$, which implies that $f_i^{y_0/y_i^*}(\boldsymbol{x}_0) < 0$, and hence $F_i(\boldsymbol{x}_0) \neq y_0$. Moreover, we know that $f_i^{y_i/y_i'}(\boldsymbol{x}_0) = f_i^{y_i/y_0}(\boldsymbol{x}_0) \geq f_i^{y_i^*/y_0}(\boldsymbol{x}_0) > f_c^{y_0/y_c^*}(\boldsymbol{x}_0) \geq f_c^{y_0/y_c'}(\boldsymbol{x}_0) = f_c^{y_c/y_c'}(\boldsymbol{x}_0)$ so $\mathcal{M}(\boldsymbol{x}_0) \neq F_c(\boldsymbol{x}_0) = y_0$.

- If $y_c \neq y_0$, i.e., $y_c' = y_0$, then $f_c^{y_c/y_0}(\boldsymbol{x}_0) \geq f_c^{y_c^*/y_0}(\boldsymbol{x}_0) > f_i^{y_0/y_i^*}(\boldsymbol{x}_0)$. If $F_i(\boldsymbol{x}_0) = y_0$, then $f_i^{y_0/y_i^*}(\boldsymbol{x}_0) \geq f_i^{y_0/y_i'}(\boldsymbol{x}_0) = f_i^{y_i/y_i'}(\boldsymbol{x}_0)$. Thus, $f_c^{y_c/y_c'}(\boldsymbol{x}_0) = f_c^{y_c/y_0}(\boldsymbol{x}_0) > f_i^{y_i/y_i'}(\boldsymbol{x}_0)$. As the result, $\mathcal{M}(\boldsymbol{x}_0) = F_c(\boldsymbol{x}_0) \neq y_0$.

For both cases, we show that $\mathcal{M}_{\mathrm{MME}}(\boldsymbol{x}_0) \neq y_0$, i.e., Equation (3) is a necessary condition for $\mathcal{M}(\boldsymbol{x}_0) = y_0$. $\qquad\square$

*Proof of Theorem 1.* Lemmas B.1 and B.2 are exactly the two directions (necessary and sufficient condition) of $\mathcal{M}_{\mathrm{MME}}$ predicting label $y_0$ at point $\boldsymbol{x}$. Therefore, if the condition (Equation (3)) holds for any $\boldsymbol{x} \in \{\boldsymbol{x}_0 + \boldsymbol{\delta} : \|\boldsymbol{\delta}\|_2 \leq r\}$, the ensemble $\mathcal{M}_{\mathrm{MME}}$ is $r$-robust at point $\boldsymbol{x}_0$; vice versa. $\qquad\square$

For comparison, here we list the trivial robustness condition for single model.

**Fact B.1** (Robustness Condition for Single Model). *Consider an input $\boldsymbol{x}_0 \in \mathbb{R}^d$ with ground-truth label $y_0 \in [C]$. Suppose a model $F$ satisfies $F(\boldsymbol{x}_0) = y_0$. Then, the model $F$ is $r$-robust at point $\boldsymbol{x}_0$ if and only if for any $\boldsymbol{x} \in \{\boldsymbol{x}_0 + \boldsymbol{\delta} : \|\boldsymbol{\delta}\|_2 \leq r\}$,*

$$\min_{y_i \in [C]:y_i \neq y_0} f^{y_0/y_i}(\boldsymbol{x}) \geq 0.$$

The fact is apparent given that the model predicts the class with the highest confidence.

### B.3 GRADIENT AND CONFIDENCE MARGIN-BASED CONDITION

We can concertize the preceding general robustness condition by gradients and confidence margins of base models leveraging Taylor expansion.

**Theorem 2** (Gradient and Confidence Margin Condition for WE Robustness). *Given input $\boldsymbol{x}_0 \in \mathbb{R}^d$ with ground-truth label $y_0 \in [C]$, and $\mathcal{M}_{\mathrm{WE}}$ as a WE defined over base models $\{f_i\}_{i=1}^N$ with weights $\{w_i\}_{i=1}^N$. $\mathcal{M}_{\mathrm{WE}}(\boldsymbol{x}_0) = y_0$. All base model $f_i$'s are $\beta$-smooth.*

- *(Sufficient Condition) $\mathcal{M}_{\mathrm{WE}}$ is $r$-robust at point $\boldsymbol{x}_0$ if for any $y_i \neq y_0$,*

$$\Big\| \sum_{j=1}^N w_j \nabla_{\boldsymbol{x}} f_j^{y_0/y_i}(\boldsymbol{x}_0) \Big\|_2 \leq \frac{1}{r} \sum_{j=1}^N w_j f_j^{y_0/y_i}(\boldsymbol{x}_0) - \beta r \sum_{j=1}^N w_j. \tag{4}$$

- *(Necessary Condition) If $\mathcal{M}_{\mathrm{WE}}$ is $r$-robust at point $\boldsymbol{x}_0$, then for any $y_i \neq y_0$,*

$$\Big\| \sum_{j=1}^N w_j \nabla_{\boldsymbol{x}} f_j^{y_0/y_i}(\boldsymbol{x}_0) \Big\|_2 \leq \frac{1}{r} \sum_{j=1}^N w_j f_j^{y_0/y_i}(\boldsymbol{x}_0) + \beta r \sum_{j=1}^N w_j. \tag{5}$$

*Proof of Theorem 2.* From Taylor expansion with Lagrange remainder and the $\beta$-smoothness assumption on the base models, we have

$$\sum_{j=1}^N w_j f_j^{y_0/y_i}(\boldsymbol{x}_0) - r \Big\| \sum_{j=1}^N w_j \nabla_{\boldsymbol{x}} f_j^{y_0/y_i}(\boldsymbol{x}_0) \Big\|_2 - \frac{1}{2} r^2 \sum_{j=1}^N (2\beta w_j) \leq \min_{\boldsymbol{x}: \|\boldsymbol{x}-\boldsymbol{x}_0\|_2 \leq r} \sum_{j=1}^N w_j f_j^{y_0/y_i}(\boldsymbol{x})$$

$$\leq \sum_{j=1}^N w_j f_j^{y_0/y_i}(\boldsymbol{x}_0) - r \Big\| \sum_{j=1}^N w_j \nabla_{\boldsymbol{x}} f_j^{y_0/y_i}(\boldsymbol{x}_0) \Big\|_2 + \frac{1}{2} r^2 \sum_{j=1}^N (2\beta w_j), \tag{23}$$

where the term $-\frac{1}{2} r^2 \sum_{j=1}^N (2\beta w_j)$ and $\frac{1}{2} r^2 \sum_{j=1}^N (2\beta w_j)$ are bounded from Lagrange remainder. From Proposition 1, the sufficient and necessary condition of WE's $r$-robustness is $\sum_{j=1}^N w_j f_j^{y_0/y_i}(\boldsymbol{x}) \geq 0$ for any $y_i \in [C]$ such that $y_i \neq y_0$, and any $\boldsymbol{x} = \boldsymbol{x}_0 + \boldsymbol{\delta}$ where $\|\boldsymbol{\delta}\|_2 \leq r$. Plugging this term into Equation (23) we get the theorem. $\square$

**Theorem 3** (Gradient and Confidence Margin Condition for MME Robustness). *Given input $\boldsymbol{x}_0 \in \mathbb{R}^d$ with ground-truth label $y_0 \in [C]$, and $\mathcal{M}_{\mathrm{MME}}$ as an MME defined over base models $\{f_1, f_2\}$. $\mathcal{M}_{\mathrm{MME}}(\boldsymbol{x}_0) = y_0$. Both $f_1$ and $f_2$ are $\beta$-smooth.*

- *(Sufficient Condition) If for any $y_1, y_2 \in [C]$ such that $y_1 \neq y_0$ and $y_2 \neq y_0$,*

$$\|\nabla_{\boldsymbol{x}} f_1^{y_0/y_1}(\boldsymbol{x}_0) + \nabla_{\boldsymbol{x}} f_2^{y_0/y_2}(\boldsymbol{x}_0)\|_2 \leq \frac{1}{r} (f_1^{y_0/y_1}(\boldsymbol{x}_0) + f_2^{y_0/y_2}(\boldsymbol{x}_0)) - 2\beta r, \tag{6}$$

  *then $\mathcal{M}_{\mathrm{MME}}$ is $r$-robust at point $\boldsymbol{x}_0$.*

- *(Necessary Condition) Suppose for any $\boldsymbol{x} \in \{\boldsymbol{x}_0 + \boldsymbol{\delta} : \|\boldsymbol{\delta}\|_2 \leq r\}$, for any $i \in \{1, 2\}$, either $F_i(\boldsymbol{x}) = y_0$ or $F_i^{(2)}(\boldsymbol{x}) = y_0$. If $\mathcal{M}_{\mathrm{MME}}$ is $r$-robust at point $\boldsymbol{x}_0$, then for any $y_1, y_2 \in [C]$ such that $y_1 \neq y_0$ and $y_2 \neq y_0$,*

$$\|\nabla_{\boldsymbol{x}} f_1^{y_0/y_1}(\boldsymbol{x}_0) + \nabla_{\boldsymbol{x}} f_2^{y_0/y_2}(\boldsymbol{x}_0)\|_2 \leq \frac{1}{r} (f_1^{y_0/y_1}(\boldsymbol{x}_0) + f_2^{y_0/y_2}(\boldsymbol{x}_0)) + 2\beta r. \tag{7}$$

*Proof of Theorem 3.* We prove the sufficient condition and necessary condition separately.

- (Sufficient Condition)
  From Lemma B.1, since there are only two base models, we can simplify the sufficient condition for $\mathcal{M}_{\mathrm{MME}}(\boldsymbol{x}) = y_0$ as

$$\min_{y_i \in [C]: y_i \neq y_0} f_1^{y_0/y_i}(\boldsymbol{x}) + \min_{y_i' \in [C]: y_i' \neq y_0} f_2^{y_0/y_i'}(\boldsymbol{x}) \geq 0.$$

In other words, for any $y_1 \neq y_0$ and $y_2 \neq y_0$,

$$f_1^{y_0/y_1}(\boldsymbol{x}) + f_2^{y_0/y_2}(\boldsymbol{x}) \geq 0. \tag{24}$$

With Taylor expansion and smoothness assumption, we have

$$\min_{\boldsymbol{x}:\|\boldsymbol{x}-\boldsymbol{x}_0\|_2 \leq r} f_1^{y_0/y_1}(\boldsymbol{x}) + f_2^{y_0/y_2}(\boldsymbol{x})$$

$$\geq f_1^{y_0/y_1}(\boldsymbol{x}_0) + f_2^{y_0/y_2}(\boldsymbol{x}_0) - r\|\nabla_{\boldsymbol{x}} f_1^{y_0/y_1}(\boldsymbol{x}_0) + \nabla_{\boldsymbol{x}} f_2^{y_0/y_2}(\boldsymbol{x}_0)\|_2 - \frac{1}{2} \cdot 4\beta r^2.$$

Plugging this into Equation (24) yields the sufficient condition. In the above equation, the term $-\frac{1}{2} \cdot 4\beta r^2$ is bounded from Lagrange remainder. Here, the $4\beta$ term comes from the fact that $f_1^{y_0/y_1}(\boldsymbol{x}) + f_2^{y_0/y_2}(\boldsymbol{x})$ is $(4\beta)$-smooth since it is the sum of difference of $\beta$-smooth function.

- (Necessary Condition)
  From Lemma B.2, similarly, the necessary condition for $\mathcal{M}_{\mathrm{MME}}(\boldsymbol{x}) = y_0$ is simplified to: for any $y_1 \neq y_0$ and $y_2 \neq y_0$,

$$f_1^{y_0/y_1}(\boldsymbol{x}) + f_2^{y_0/y_2}(\boldsymbol{x}) \geq 0. \tag{24}$$

Again, from Taylor expansion, we have

$$\min_{\boldsymbol{x}:\|\boldsymbol{x}-\boldsymbol{x}_0\|_2 \leq r} f_1^{y_0/y_1}(\boldsymbol{x}) + f_2^{y_0/y_2}(\boldsymbol{x})$$

$$\leq f_1^{y_0/y_1}(\boldsymbol{x}_0) + f_2^{y_0/y_2}(\boldsymbol{x}_0) - r\|\nabla_{\boldsymbol{x}} f_1^{y_0/y_1}(\boldsymbol{x}_0) + \nabla_{\boldsymbol{x}} f_2^{y_0/y_2}(\boldsymbol{x}_0)\|_2 + \frac{1}{2} \cdot 4\beta r^2.$$

Plugging this into Equation (24) yields the necessary condition. In the above equation, the term $+\frac{1}{2} \cdot 4\beta r^2$ is bounded from Lagrange remainder. The $4\beta$ term appears because of the same reason as before.

$\square$

To compare the robustness of ensemble models and the single model, we show the corresponding conditions for single-model robustness.

**Proposition B.1** (Gradient and Confidence Margin Conditions for Single-Model Robustness). *Given input $\boldsymbol{x}_0 \in \mathbb{R}^d$ with ground-truth label $y_0 \in [C]$. Model $F(\boldsymbol{x}_0) = y_0$, and it is $\beta$-smooth.*

- *(Sufficient Condition) If for any $y_1 \in [C]$ such that $y_1 \neq y_0$,*

$$\|\nabla_{\boldsymbol{x}} f^{y_0/y_1}(\boldsymbol{x}_0)\|_2 \leq \frac{1}{r} f^{y_0/y_1}(\boldsymbol{x}_0) - \beta r, \tag{25}$$

*$F$ is $r$-robust at point $\boldsymbol{x}_0$.*

- *(Necessary Condition) If $F$ is $r$-robust at point $\boldsymbol{x}_0$, for any $y_1 \in [C]$ such that $y_1 \neq y_0$,*

$$\|\nabla_{\boldsymbol{x}} f^{y_0/y_1}(\boldsymbol{x}_0)\|_2 \leq \frac{1}{r} f^{y_0/y_1}(\boldsymbol{x}_0) + \beta r. \tag{26}$$

*Proof of Proposition B.1.* This proposition is apparent given the following inequality from Taylor expansion

$$f^{y_0/y_1}(\boldsymbol{x}_0) - r\|\nabla_{\boldsymbol{x}} f^{y_0/y_1}(\boldsymbol{x}_0)\|_2 - \beta r^2 \leq \min_{\boldsymbol{x}:\|\boldsymbol{x}-\boldsymbol{x}_0\|_2 \leq r} f^{y_0/y_1}(\boldsymbol{x}) \leq f^{y_0/y_1}(\boldsymbol{x}_0) - r\|\nabla_{\boldsymbol{x}} f^{y_0/y_1}(\boldsymbol{x}_0)\|_2 + \beta r^2$$

and the necessary and sufficient robust condition in Fact B.1. $\square$

B.4 COMPARE CERTIFIED ROBUSTNESS OF ENSEMBLE MODEL AND SINGLE MODELS

**Corollary 1** (Comparison of Ensemble and Single-Model Robustness). *Given an input $\boldsymbol{x}_0 \in \mathbb{R}^d$ with ground-truth label $y_0 \in [C]$. Suppose we have two $\beta$-smooth base models $\{f_1, f_2\}$, which are both $r$-robust at point $\boldsymbol{x}_0$. For any $\Delta \in [0, 1)$:*

- *(Weighted Ensemble) Define Weighted Ensemble $\mathcal{M}_{\mathrm{WE}}$ with base models $\{f_1, f_2\}$. Suppose $\mathcal{M}_{\mathrm{WE}}(\boldsymbol{x}_0) = y_0$. If for any label $y_i \neq y_0$, the base models' smoothness $\beta \leq \Delta \cdot \min\{f_1^{y_0/y_i}(\boldsymbol{x}_0), f_2^{y_0/y_i}(\boldsymbol{x}_0)\}/(c^2 r^2)$, and the gradient cosine similarity $\cos\langle \nabla_{\boldsymbol{x}} f_1^{y_0/y_i}(\boldsymbol{x}_0), \nabla_{\boldsymbol{x}} f_2^{y_0/y_i}(\boldsymbol{x}_0) \rangle \leq \cos\theta$, then the $\mathcal{M}_{\mathrm{WE}}$ with weights $\{w_1, w_2\}$ is at least $R$-robust at point $\boldsymbol{x}_0$ with*

$$R = r \cdot \frac{1-\Delta}{1+\Delta}\left(1 - C_{\mathrm{WE}}(1-\cos\theta)\right)^{-1/2}, \text{where} \tag{8}$$

$$C_{\mathrm{WE}} = \min_{y_i:y_i \neq y_0} \frac{2w_1 w_2 f_1^{y_0/y_i}(\boldsymbol{x}_0) f_2^{y_0/y_i}(\boldsymbol{x}_0)}{(w_1 f_1^{y_0/y_i}(\boldsymbol{x}_0) + w_2 f_2^{y_0/y_i}(\boldsymbol{x}_0))^2}, c = \max\{\tfrac{1-\Delta}{1+\Delta}\left(1 - C_{\mathrm{WE}}(1-\cos\theta)\right)^{-1/2}, 1\}.$$

- *(Max-Margin Ensemble) Define Max-Margin Ensemble $\mathcal{M}_{\mathrm{MME}}$ with the base models $\{f_1, f_2\}$. Suppose $\mathcal{M}_{\mathrm{MME}}(\boldsymbol{x}_0) = y_0$. If for any label $y_1 \neq y_0$ and $y_2 \neq y_0$, the base models' smoothness $\beta \leq \Delta \cdot \min\{f_1^{y_0/y_1}(\boldsymbol{x}_0), f_2^{y_0/y_2}(\boldsymbol{x}_0)\}/(c^2 r^2)$, and the gradient cosine similarity $\cos\langle \nabla_{\boldsymbol{x}} f_1^{y_0/y_1}(\boldsymbol{x}_0), \nabla_{\boldsymbol{x}} f_2^{y_0/y_2}(\boldsymbol{x}_0) \rangle \leq \cos\theta$, then the $\mathcal{M}_{\mathrm{MME}}$ is at least $R$-robust at point $\boldsymbol{x}_0$ with*

$$R = r \cdot \frac{1-\Delta}{1+\Delta}\left(1 - C_{\mathrm{MME}}(1-\cos\theta)\right)^{-1/2}, \text{where} \tag{9}$$

$$C_{\mathrm{MME}} = \min_{\substack{y_1, y_2: \\ y_1, y_2 \neq y_0}} \frac{2 f_1^{y_0/y_1}(\boldsymbol{x}_0) f_2^{y_0/y_2}(\boldsymbol{x}_0)}{(f_1^{y_0/y_1}(\boldsymbol{x}_0) + f_2^{y_0/y_2}(\boldsymbol{x}_0))^2}, c = \max\{\tfrac{1-\Delta}{1+\Delta}\left(1 - C_{\mathrm{MME}}(1-\cos\theta)\right)^{-1/2}, 1\}.$$

*Proof of Corollary 1.* We first prove the theorem for Weighted Ensemble. For arbitrary $y_i \neq y_0$, we have

$$\begin{aligned}
&\|w_1 \nabla_{\boldsymbol{x}} f_1^{y_0/y_i}(\boldsymbol{x}_0) + w_2 \nabla_{\boldsymbol{x}} f_2^{y_0/y_i}(\boldsymbol{x}_0)\|_2 \\
=&\sqrt{w_1^2 \|\nabla_{\boldsymbol{x}} f_1^{y_0/y_i}(\boldsymbol{x}_0)\|_2^2 + w_2^2 \|\nabla_{\boldsymbol{x}} f_2^{y_0/y_i}(\boldsymbol{x}_0)\|_2^2 + 2w_1 w_2 \langle \nabla_{\boldsymbol{x}} f_1^{y_0/y_i}(\boldsymbol{x}_0),\ f_2^{y_0/y_i}(\boldsymbol{x}_0) \rangle} \\
\leq&\sqrt{w_1^2 \|\nabla_{\boldsymbol{x}} f_1^{y_0/y_i}(\boldsymbol{x}_0)\|_2^2 + w_2^2 \|\nabla_{\boldsymbol{x}} f_2^{y_0/y_i}(\boldsymbol{x}_0)\|_2^2 + 2w_1 w_2 \|\nabla_{\boldsymbol{x}} f_1^{y_0/y_i}(\boldsymbol{x}_0)\|_2 \|\nabla_{\boldsymbol{x}} f_2^{y_0/y_i}(\boldsymbol{x}_0)\|_2 \cos\theta} \\
\overset{(i.)}{\leq}&\sqrt{w_1^2 \left(\tfrac{1}{r} f_1^{y_0/y_i}(\boldsymbol{x}_0) + \beta r\right)^2 + w_2^2 \left(\tfrac{1}{r} f_2^{y_0/y_i}(\boldsymbol{x}_0) + \beta r\right)^2 + 2w_1 w_2 \left(\tfrac{1}{r} f_1^{y_0/y_i}(\boldsymbol{x}_0) + \beta r\right)\left(\tfrac{1}{r} f_2^{y_0/y_i}(\boldsymbol{x}_0) + \beta r\right)\cos\theta} \\
=&\frac{1}{r}\sqrt{w_1^2 \left(f_1^{y_0/y_i}(\boldsymbol{x}_0) + \beta r^2\right)^2 + w_2^2 \left(f_2^{y_0/y_i}(\boldsymbol{x}_0) + \beta r^2\right)^2 + 2w_1 w_2 \left(f_1^{y_0/y_i}(\boldsymbol{x}_0) + \beta r^2\right)\left(f_2^{y_0/y_i}(\boldsymbol{x}_0) + \beta r^2\right)\cos\theta} \\
\overset{(ii.)}{\leq}&\frac{1}{r} \cdot \left(1 + \frac{\Delta}{c^2}\right)\sqrt{w_1^2 f_1^{y_0/y_i}(\boldsymbol{x}_0)^2 + w_2^2 f_2^{y_0/y_i}(\boldsymbol{x}_0)^2 + 2w_1 w_2 f_1^{y_0/y_i}(\boldsymbol{x}_0) f_2^{y_0/y_i}(\boldsymbol{x}_0)\cos\theta} \\
=&\frac{1}{r} \cdot \left(1 + \frac{\Delta}{c^2}\right)\sqrt{\left(w_1 f_1^{y_0/y_i}(\boldsymbol{x}_0) + w_2 f_2^{y_0/y_i}(\boldsymbol{x}_0)\right)^2 - 2(1-\cos\theta) w_1 f_1^{y_0/y_i}(\boldsymbol{x}_0) w_2 f_2^{y_0/y_i}(\boldsymbol{x}_0)} \\
\overset{(iii.)}{\leq}&\frac{1}{r} \cdot \left(1 + \frac{\Delta}{c^2}\right)\sqrt{1 - (1-\cos\theta)C_{\mathrm{WE}}}\left(w_1 f_1^{y_0/y_i}(\boldsymbol{x}_0) + w_2 f_2^{y_0/y_i}(\boldsymbol{x}_0)\right)
\end{aligned}$$

where $(i.)$ follows from the necessary condition in Proposition B.1; $(ii.)$ uses the condition on $\beta$; and $(iii.)$ replaces $2w_1 w_2 f_1^{y_0/y_i}(\boldsymbol{x}_0) f_2^{y_0/y_i}(\boldsymbol{x}_0)$ leveraging $C_{\mathrm{WE}}$. Now, we define

$$K := \frac{1-\Delta}{1+\Delta}\left(1 - C_{\mathrm{WE}}(1-\cos\theta)\right)^{-1/2}.$$

All we need to do is to prove that $\mathcal{M}_{\mathrm{WE}}$ is robust within radius $Kr$. To do so, from Equation (4), we upper bound $\|w_1 \nabla_{\boldsymbol{x}} f_1^{y_0/y_i}(\boldsymbol{x}_0) + w_2 \nabla_{\boldsymbol{x}} f_2^{y_0/y_i}(\boldsymbol{x}_0)\|_2$ by $\frac{1}{Kr}\left(w_1 f_1^{y_0/y_i}(\boldsymbol{x}_0) + w_2 f_2^{y_0/y_i}(\boldsymbol{x}_0)\right) -$

$\beta K r(w_1 + w_2)$:

$$\|w_1 \nabla_{\boldsymbol{x}} f_1^{y_0/y_i}(\boldsymbol{x}_0) + w_2 \nabla_{\boldsymbol{x}} f_2^{y_0/y_i}(\boldsymbol{x}_0)\|_2$$

$$\leq \frac{1}{r} \cdot \left(1 + \frac{\Delta}{c^2}\right) \sqrt{1 - (1 - \cos\theta)C_{\text{WE}}} \left(w_1 f_1^{y_0/y_i}(\boldsymbol{x}_0) + w_2 f_2^{y_0/y_i}(\boldsymbol{x}_0)\right)$$

$$\leq \frac{1}{r}(1 + \Delta)\sqrt{1 - (1 - \cos\theta)C_{\text{WE}}} \left(w_1 f_1^{y_0/y_i}(\boldsymbol{x}_0) + w_2 f_2^{y_0/y_i}(\boldsymbol{x}_0)\right)$$

$$= \frac{1}{r} \cdot \frac{1 - \Delta}{\frac{1-\Delta}{1+\Delta}\left(1 - (1 - \cos\theta)C_{\text{WE}}\right)^{-1/2}} \left(w_1 f_1^{y_0/y_i}(\boldsymbol{x}_0) + w_2 f_2^{y_0/y_i}(\boldsymbol{x}_0)\right)$$

$$= \frac{1}{Kr}(1 - \Delta)\left(w_1 f_1^{y_0/y_i}(\boldsymbol{x}_0) + w_2 f_2^{y_0/y_i}(\boldsymbol{x}_0)\right)$$

$$\leq \frac{1}{Kr}\left(w_1 f_1^{y_0/y_i}(\boldsymbol{x}_0) + w_2 f_2^{y_0/y_i}(\boldsymbol{x}_0) - \Delta \min\{f_1^{y_0/y_i}(\boldsymbol{x}_0), f_2^{y_0/y_i}(\boldsymbol{x}_0)\}(w_1 + w_2)\right).$$

Notice that $\Delta \min\{f_1^{y_0/y_i}(\boldsymbol{x}_0), f_2^{y_0/y_i}(\boldsymbol{x}_0)\} \geq \beta c^2 r^2$ from $\beta$'s condition, so

$$\|w_1 \nabla_{\boldsymbol{x}} f_1^{y_0/y_i}(\boldsymbol{x}_0) + w_2 \nabla_{\boldsymbol{x}} f_2^{y_0/y_i}(\boldsymbol{x}_0)\|_2$$

$$\leq \frac{1}{Kr}\left(w_1 f_1^{y_0/y_i}(\boldsymbol{x}_0) + w_2 f_2^{y_0/y_i}(\boldsymbol{x}_0) - \beta c^2 r^2(w_1 + w_2)\right)$$

$$= \frac{1}{Kr}\left(w_1 f_1^{y_0/y_i}(\boldsymbol{x}_0) + w_2 f_2^{y_0/y_i}(\boldsymbol{x}_0)\right) - \beta K r(w_1 + w_2) \cdot \frac{c^2}{K^2}$$

$$\leq \frac{1}{Kr}\left(w_1 f_1^{y_0/y_i}(\boldsymbol{x}_0) + w_2 f_2^{y_0/y_i}(\boldsymbol{x}_0)\right) - \beta K r(w_1 + w_2).$$

From Equation (4), the theorem for Weighted Ensemble is proved.

Now we prove the theorem for Max-Margin Ensemble. Similarly, for any arbitrary $y_1, y_2$ such that $y_1 \neq y_0, y_2 \neq y_0$, we have

$$\|\nabla_{\boldsymbol{x}} f_1^{y_0/y_1}(\boldsymbol{x}_0) + \nabla_{\boldsymbol{x}} f_2^{y_0/y_2}(\boldsymbol{x}_0)\|_2$$

$$\leq \frac{1}{r} \cdot \left(1 + \frac{\Delta}{c^2}\right) \sqrt{1 - (1 - \cos\theta)C_{\text{MME}}} \left(f_1^{y_0/y_1}(\boldsymbol{x}_0) + f_2^{y_0/y_2}(\boldsymbol{x}_0)\right).$$

Now we define

$$K' := \frac{1 - \Delta}{1 + \Delta}\left(1 - C_{\text{MME}}(1 - \cos\theta)\right)^{-1/2}.$$

Again, from $\beta$'s condition we have $\Delta \min\{f_1^{y_0/y_1}(\boldsymbol{x}_0), f_2^{y_0/y_2}(\boldsymbol{x}_0)\} \geq \beta c^2 r^2$ and

$$\|\nabla_{\boldsymbol{x}} f_1^{y_0/y_1}(\boldsymbol{x}_0) + \nabla_{\boldsymbol{x}} f_2^{y_0/y_2}(\boldsymbol{x}_0)\|_2$$

$$\leq \frac{1}{K'r}\left(f_1^{y_0/y_i}(\boldsymbol{x}_0) + f_2^{y_0/y_i}(\boldsymbol{x}_0)\right) - 2\beta K' r.$$

From Equation (6), the ensemble is $(K'r)$-robust at point $\boldsymbol{x}_0$, i.e., the theorem for Max-Margin Ensemble is proved. $\qquad\square$

DISCUSSION

**Optimizing Weighted Ensemble.** As we can observe from Corollary 1, we can adjust the weights $\{w_1, w_2\}$ for Weighted Ensemble to change $C_{\text{WE}}$ and the certified robust radius (Equation (8)). Then comes the problem of which set of weights can achieve the highest certified robust radius. Since larger $C_{\text{WE}}$ results in higher radius, we need to choose

$$(w_1^{OPT}, w_2^{OPT}) = \arg\max_{w_1, w_2} \min_{y_i : y_i \neq y_0} \frac{2 w_1 w_2 f_1^{y_0/y_i}(\boldsymbol{x}_0) f_2^{y_0/y_i}(\boldsymbol{x}_0)}{(w_1 f_1^{y_0/y_i}(\boldsymbol{x}_0) + w_2 f_2^{y_0/y_i}(\boldsymbol{x}_0))^2}.$$

Since this quantity is scale-invariant, we can fix $w_1$ and optimize over $w_2$ to get the optimal weights. In particular, if there are only two classes, we have a closed-form solution

$$(w_1^{OPT}, w_2^{OPT}) = \arg\max_{w_1, w_2} \frac{2w_1 w_2 f_1^{y_0/y_1}(\boldsymbol{x}_0) f_2^{y_0/y_1}(\boldsymbol{x}_0)}{(w_1 f_1^{y_0/y_1}(\boldsymbol{x}_0) + w_2 f_2^{y_0/y_1}(\boldsymbol{x}_0))^2}$$
$$= \{k \cdot f_2^{y_0/y_1}(\boldsymbol{x}_0), k \cdot f_1^{y_0/y_1}(\boldsymbol{x}_0) : k \in \mathbb{R}_+\},$$

and corresponding $C_{\text{WE}}$ achieves the maximum $1/2$.

For a special case—average weighted ensemble, we get the corresponding certified robust radius by setting $w_1 = w_2$ and plug the yielded

$$C_{\text{WE}} = \min_{y_i: y_i \neq y_0} \frac{2 f_1^{y_0/y_i}(\boldsymbol{x}_0) f_2^{y_0/y_i}(\boldsymbol{x}_0)}{(f_1^{y_0/y_i}(\boldsymbol{x}_0) + f_2^{y_0/y_i}(\boldsymbol{x}_0))^2} \in (0, 1/2].$$

into Equation (8).

**Comparison between ensemble and single-model robustness.** We expand the discussion in Section 2.1. The similar forms of $R$ in the corollary allow us to discuss the Weighted Ensemble and Max-Margin Ensemble together. Specifically, we let $C$ be either $C_{\text{WE}}$ or $C_{\text{MME}}$, then

$$R = r \cdot \frac{1 - \Delta}{1 + \Delta} \left(1 - C(1 - \cos\theta)\right)^{-1/2}.$$

Since when $R > r$, both ensembles have higher certified robustness than the base models, we solve this condition for $\cos\theta$:

$$R > r$$
$$\iff \left(\frac{1 - \Delta}{1 + \Delta}\right)^2 > 1 - C(1 - \cos\theta)$$
$$\iff \cos\theta \leq 1 - \frac{4\Delta}{C(1 + \Delta)^2}.$$

Notice that $C \in (0, 1/2]$. From this condition, we can easily observe that when the gradient cosine similarity is smaller, it is more likely that the ensemble has higher certified robustness than the base models. When the model is smooth enough, according to the condition on $\beta$, we can notice that $\Delta$ would be close to zero. As a result, $1 - \frac{4\Delta}{C(1+\Delta)^2}$ is close to 1. Thus, unless the gradient of base models is (or close to) colinear, it always holds that the ensemble (either WE or MME) has higher certified robustness than the base models.

## C  ANALYSIS OF ENSEMBLE SMOOTHING STRATEGIES

In Section 3 we mainly use the adapted randomized model smoothing strategy which is named Ensemble Before Smoothing (EBS). We also consider Ensemble After Smoothing (Ensemble After Smoothing). Through the following analysis, we will show Ensemble Before Smoothing is generally better than Ensemble After Smoothing which justifies our choice of the strategy.

We formally define Ensemble Before Smoothing strategy as below:

**Definition C.1** (Strategy: **Ensemble Before Smoothing (EBS)**). Let $\mathcal{M}$ be an ensemble model over base models $\{f_i\}_{i=1}^N$. Let $\epsilon$ be a random variable. The EBS ensemble $G_{\mathcal{M}}^{\epsilon} : \mathbb{R}^d \mapsto [C]$ at input $\boldsymbol{x}_0 \in \mathbb{R}^d$ is defined by:

$$G_{\mathcal{M}}^{\epsilon}(\boldsymbol{x}_0) = \arg\max_{j \in [C]} \Pr_{\epsilon}(\mathcal{M}(\boldsymbol{x}_0 + \epsilon) = j). \tag{27}$$

We define the Ensemble After Smoothing strategy accordingly:

**Definition C.2** (Strategy: **Ensemble After Smoothing (EAS)**). Let $\mathcal{M}$ be an ensemble model over base models $\{f_i\}_{i=1}^N$. Let $\epsilon$ be a random variable. The EAS ensemble $H_{\mathcal{M}}^{\epsilon} : \mathbb{R}^d \mapsto [C]$ at input $\boldsymbol{x}_0 \in \mathbb{R}^d$ is defined as:

$$H_{\mathcal{M}}^{\epsilon}(\boldsymbol{x}_0) := G_{F_c}^{\epsilon}(\boldsymbol{x}_0) \quad \text{where} \quad c = \arg\max_{i \in [N]} g_{F_i}^{\epsilon}(\boldsymbol{x}_0)_{G_{F_i}^{\epsilon}(\boldsymbol{x}_0)}. \tag{28}$$

Here, $c$ is the index of the smoothed base model selected.

*Remark.* In EBS, we first construct a model ensemble $\mathcal{M}$ based on base models using WE or MME protocol, then apply randomized smoothing on top of the classifier. The classifier predicts the most frequent class of $\mathcal{M}$ when the input follows distribution $\boldsymbol{x}_0 + \epsilon$.

In EAS, we use $\epsilon$ to construct smoothed classifiers for base models respectively. Then, for given input $\boldsymbol{x}_0$, the ensemble agrees on the base model which has the highest probability for its predicted class.

### C.1  CERTIFIED ROBUSTNESS

In this subsection, we characterize the certified robustness when using both strategies.

#### C.1.1  ENSEMBLE BEFORE SMOOTHING

**Proposition C.1** (Certified Robustness for Ensemble Before Smoothing). *Let $G_{\mathcal{M}}^{\epsilon}$ be an ensemble constructed by EBS strategy. The random variable $\epsilon \sim \mathcal{N}(0, \sigma^2 \boldsymbol{I}_d)$. Then the ensemble $G_{\mathcal{M}}^{\epsilon}$ is $r$-robust at point $\boldsymbol{x}_0$ where*

$$r := \sigma \Phi^{-1} \left( g_{\mathcal{M}}^{\epsilon}(\boldsymbol{x}_0)_{G_{\mathcal{M}}^{\epsilon}(\boldsymbol{x}_0)} \right). \tag{29}$$

*Here, $g_{\mathcal{M}}^{\epsilon}(\boldsymbol{x}_0)_j = \Pr_{\epsilon}(\mathcal{M}(\boldsymbol{x}_0 + \epsilon) = j)$.*

The proposition is a direct application of Theorem B.1.

#### C.1.2  ENSEMBLE AFTER SMOOTHING

**Theorem C.1** (Certified robustness for Ensemble After Smoothing). *Let $H_{\mathcal{M}}^{\epsilon}$ be an ensemble constructed by EAS strategy over base models $\{f_i\}_{i=1}^N$. The random variable $\epsilon \sim \mathcal{N}(0, \sigma^2 \boldsymbol{I}_d)$. Let $y_0 = H_{\mathcal{M}}^{\epsilon}(\boldsymbol{x}_0)$. For each $i \in [N]$, define*

$$r_i := \begin{cases} \sigma \Phi^{-1} \left( g_{F_i}^{\epsilon}(\boldsymbol{x}_0)_{G_{F_i}^{\epsilon}(\boldsymbol{x}_0)} \right), & \text{if } G_{F_i}^{\epsilon}(\boldsymbol{x}_0) = y_0 \\ -\sigma \Phi^{-1} \left( g_{F_i}^{\epsilon}(\boldsymbol{x}_0)_{G_{F_i}^{\epsilon}(\boldsymbol{x}_0)} \right). & \text{if } G_{F_i}^{\epsilon}(\boldsymbol{x}_0) \neq y_0 \end{cases}$$

*Then the ensemble $H_{\mathcal{M}}^{\epsilon}$ is $r$-robust at point $\boldsymbol{x}_0$ where*

$$r := \frac{\max_{i \in [N]} r_i + \min_{i \in [N]} r_i}{2}. \tag{30}$$

*Remark.* The theorem appears to be a bit counter-intuitive — picking the best smoothed model in terms of certified robustness cannot give strong certified robustness for the ensemble. As long as the base models have different certified robust radius (i.e., $r_i$'s are different), the $r$, certified robust radius for the ensemble, is strictly inferior to that of the best base model (i.e., $\max r_i$). Furthermore, if there exists a base model with wrong prediction (i.e., $r_i \leq 0$), the certified robust radius $r$ is strictly smaller than *half* of the best base model.

*Proof of Theorem C.1.* Without loss of generality, we assume $r_1 > r_2 > \cdots > r_N$. Let the perturbation added to $\boldsymbol{x}_0$ has $L_2$ length $\delta$.

When $\delta \leq r_N$, since picking any model always gives the right prediction, the ensemble is robust.

When $r_N < \delta \leq \frac{r_1+r_N}{2}$, the highest robust radius with wrong prediction is $\delta - r_N$, and we can still guarantee that model $f_1$ has robust radius at least $r_1 - \delta$ from the smoothness of function $\boldsymbol{x} \mapsto g^{\boldsymbol{\epsilon}}_{F_1}(\boldsymbol{x})_{G^{\boldsymbol{\epsilon}}_{F_1}(\boldsymbol{x}_0)}$ (Salman et al., 2019). Since $r_1 - \delta \geq \frac{r_1-r_N}{2} \geq \delta - r_N$, the ensemble will agree on $f_1$ or other base model with correct prediction and still gives the right prediction.

When $\delta > \frac{r_1+r_N}{2}$, suppose $f_N$ is a linear model and only predicts two labels (which achieves the tight robust radius bound according to Cohen et al. (2019)), then $f_N$ can have robust radius $\delta - r_N$ for the wrong prediction. At the same time, for any other model $f_i$ which is linear and predicts correctly, the robust radius is at most $r_i - \delta$. Since $r_i - \delta < r_1 - \delta < \frac{r_1-r_N}{2} < \delta - r_N$, the ensemble can probably give wrong prediction.

In summary, as we have shown, the certified robust radius can be at most $r$. For any radius $\delta > r$, there exist base models which lead the ensemble $H^{\boldsymbol{\epsilon}}_{\mathcal{M}}(\boldsymbol{x}_0 + \delta \boldsymbol{e})$ to predict the label other than $y_0$. □

## C.2 Comparison of Two Strategies

In this subsection, we compare the two ensemble strategies when the ensembles are constructed from two base models.

**Corollary C.1** (Smoothing Strategy Comparison). *Given $\mathcal{M}_{\mathrm{MME}}$, a Max-Margin Ensemble constructed from base models $\{f_a, f_b\}$. Let $\boldsymbol{\epsilon} \sim \mathcal{N}(0, \sigma^2 \boldsymbol{I}_d)$. Let $G^{\boldsymbol{\epsilon}}_{\mathcal{M}_{\mathrm{MME}}}$ be the EBS ensemble, and $H^{\boldsymbol{\epsilon}}_{\mathcal{M}_{\mathrm{MME}}}$ be the EAS ensemble. Suppose at point $\boldsymbol{x}_0$ with ground-truth label $y_0$, $G^{\boldsymbol{\epsilon}}_{F_a}(\boldsymbol{x}_0) = G^{\boldsymbol{\epsilon}}_{F_b}(\boldsymbol{x}_0) = y_0$, $g^{\boldsymbol{\epsilon}}_{F_a}(\boldsymbol{x}_0) > 0.5$, $g^{\boldsymbol{\epsilon}}_{F_b}(\boldsymbol{x}_0) > 0.5$.*

*Let $\delta$ be their probability difference for class $y_0$, i.e, $\delta := |g^{\boldsymbol{\epsilon}}_{F_a}(\boldsymbol{x}_0)_{y_0} - g^{\boldsymbol{\epsilon}}_{F_b}(\boldsymbol{x}_0)_{y_0}|$,. Let $p_{\min}$ be the smaller probability for class $y_0$ between them, i.e., $p_{\min} := \min\{g^{\boldsymbol{\epsilon}}_{F_a}(\boldsymbol{x}_0)_{y_0}, g^{\boldsymbol{\epsilon}}_{F_b}(\boldsymbol{x}_0)_{y_0}\}$. We denote $p$ to the probability of choosing the correct class when the base models disagree with each other; denote $p_{ab}$ to the probability of both base models agreeing on the correct class:*

$$p := \Pr_{\boldsymbol{\epsilon}} \left( \mathcal{M}_{\mathrm{MME}}(\boldsymbol{x}_0 + \boldsymbol{\epsilon}) = y_0 \mid F_a(\boldsymbol{x}_0 + \boldsymbol{\epsilon}) \neq F_b(\boldsymbol{x}_0 + \boldsymbol{\epsilon}) \text{ and } (F_a(\boldsymbol{x}_0 + \boldsymbol{\epsilon}) = y_0 \text{ or } F_b(\boldsymbol{x}_0 + \boldsymbol{\epsilon}) = y_0) \right),$$

$$p_{ab} := \Pr_{\boldsymbol{\epsilon}} \left( F_a(\boldsymbol{x}_0 + \boldsymbol{\epsilon}) = F_b(\boldsymbol{x}_0 + \boldsymbol{\epsilon}) = y_0 \right).$$

*We have:*

*1. If $p > 1/2 + (2 + 4(p_{\min} - p_{ab})/\delta)^{-1}$, $r_G > r_H$.*

*2. If $p \leq 1/2$, $r_H \geq r_G$.*

*Here, $r_G$ is the certified robust radius of $G^{\boldsymbol{\epsilon}}_{\mathcal{M}_{\mathrm{MME}}}$ computed from Equation (29); and $r_H$ is the certified robust radius of $H^{\boldsymbol{\epsilon}}_{\mathcal{M}_{\mathrm{MME}}}$ computed from Equation (30).*

*Remark.* Since $p$ is the probability where the ensemble chooses the correct prediction between two base model predictions, with Max-Margin Ensemble, we think $p > 1/2$ with non-trivial margin.

The quantity $p_{\min} - p_{ab}$ and $\delta$ both measure the base model's diversity in terms of predicted label distribution, and generally they should be close. As a result, $1/2 + (2 + 4(p_{\min} - p_{ab})/\delta)^{-1} \approx 1/2 + 1/6 = 2/3$, and case (1) should be much more likely to happen than case (2). Therefore, *EBS usually yields higher robustness guarantee*. We remark that the similar tendency also holds with multiple base models.

*Proof of Corollary C.1.* For convenience, define $p_a := g_{F_a}^\epsilon(\boldsymbol{x}_0)_{y_0}, p_b := g_{F_b}^\epsilon(\boldsymbol{x}_0)_{y_0}$, where $p_a = p_b + \delta$ and $p_{\min} = p_b$.

From Proposition C.1 and Theorem C.1, we have

$$r_G := \frac{\sigma}{2} \cdot 2\Phi^{-1}\left(\Pr_\epsilon(\mathcal{M}_{\mathrm{MME}}(\boldsymbol{x}_0 + \epsilon) = y_0)\right), \quad r_H := \frac{\sigma}{2}\left(\Phi^{-1}(p_a) + \Phi^{-1}(p_b)\right).$$

Notice that $\Pr_\epsilon(\mathcal{M}_{\mathrm{MME}}(\boldsymbol{x}_0 + \epsilon) = y_0) = p_{ab} + p(p_a + p_b - 2p_{ab})$, we can rewrite $r_G$ as

$$r_G = \frac{\sigma}{2} \cdot 2\Phi^{-1}(p_{ab} + p(p_a + p_b - 2p_{ab})).$$

1. When $p > 1/2 + (2 + 4(p_{\min} - p_{ab})/\delta)^{-1}$,
   since

   $$p > \frac{1}{2} + \frac{1}{2 + \frac{4(p_{\min} - p_{ab})}{\delta}} = \frac{1}{2} + \frac{\delta}{2\delta + 4(p_b - p_{ab})} = \frac{p_a + p_b + \delta - 2p_{ab}}{2(p_a + p_b - 2p_{ab})} = \frac{p_a - p_{ab}}{p_a + p_b - 2p_{ab}},$$

   we have $p_{ab} + p(p_a + p_b - 2p_{ab}) > p_a$. Therefore, $r_G > \sigma\Phi^{-1}(p_a)$. Whereas, $r_H \le \sigma/2 \cdot 2\Phi^{-1}(p_a) = \sigma\Phi^{-1}(p_a)$. So $r_G > r_H$.

2. When $p \le 1/2$,

   $$p_{ab} + p(p_a + p_b - 2p_{ab}) \le p_{ab} + 1/2 \cdot (p_a + p_b - 2p_{ab}) = (p_a + p_b)/2.$$

   Therefore, $r_G \le \sigma\Phi^{-1}((p_a + p_b)/2)$. Notice that $\Phi^{-1}$ is convex in $[1/2, +\infty)$, so $\Phi^{-1}(p_a) + \Phi^{-1}(p_b) \ge 2\Phi^{-1}((p_a + p_b)/2)$, i.e., $r_H \ge r_G$.

   $\square$

# D PROOFS AND EXPERIMENTS OF ROBUSTNESS FOR SMOOTHED ML ENSEMBLE

In this appendix, we provide the detailed proofs and discussions for the results in Sections 3.1 and 3.2. Moreover, to present a more intuitive understanding, we show both numerical and realistic experiments for the theorems.

## D.1 CERTIFIED ROBUSTNESS VIA RANDOMIZED SMOOTHING

First, using the notion of $(\epsilon, p)$-Statistical Robust (Definition 5), we prove the certified robustness of the single model and ensembles under the i.i.d. assumption of the confidence scores. As noted in Section 3.1 and Appendix B.1, when $\epsilon$ follows some distribution such as $\mathcal{N}(0, \sigma^2 I_d)$, we can translate the statistical robustness guarantee to $r$-robustness guarantee of the smoothed classifier.

The following lemma is frequently used in our following proofs:

**Lemma D.1.** *Suppose the random variable $X$ satisfies $\mathbb{E}X > 0$, $\mathrm{Var}(X) < \infty$ and for any $x \in \mathbb{R}_+$, $\Pr(X \geq \mathbb{E}X + x) = \Pr(X \leq \mathbb{E}X - x)$, then*

$$\Pr(X \leq 0) \leq \frac{\mathrm{Var}(X)}{2(\mathbb{E}X)^2}.$$

*Proof of Lemma D.1.* Apply Chebyshev's inequality on random variable $X$ and notice that $X$ is symmetric, then we can easily observe this lemma. □

### D.1.1 CERTIFIED ROBUSTNESS FOR SINGLE MODEL

As the start point, we first show a direct proposition stating the certified robustness guarantee of the single model.

**Definition D.1** $((\epsilon, \lambda, p)$-Single Confident). Given a classification model $F$. If at point $x_0$ with ground-truth label $y_0$ and the random variable $\epsilon$, we have

$$\Pr_{\epsilon}\left(\max_{y_j \in [C]: y_j \neq y_0} f(x_0 + \epsilon)_{y_j} \leq \lambda(1 - f(x_0 + \epsilon)_{y_0})\right) = 1 - p,$$

we call $F$ $(\epsilon, \lambda, p)$-single confident at point $x_0$.

**Proposition D.1** (Certified Robustness for Single Model). *Let $\epsilon$ be a random variable. Let $F$ be a classification model, which is $(\epsilon, \lambda_3, p)$-single confident. Let $x_0 \in \mathbb{R}^d$ be the input with ground-truth $y_0 \in [C]$. Suppose $f(x_0 + \epsilon)_{y_0}$ follows symmetric distribution with mean $\mu$ and variance $s^2$, where $\mu > (1 + \lambda_3^{-1})^{-1}$. We have*

$$\Pr_{\epsilon}(F(x_0 + \epsilon) = y_0) \geq 1 - p - \frac{s^2}{2(\mu - (1 + \lambda_3^{-1})^{-1})^2}.$$

*Proof of Proposition D.1.* We consider the distribution of quantity $Y := f(x_0 + \epsilon)_{y_0} - \lambda_3(1 - f(x_0 + \epsilon)_{y_0})$. Since the model $F$ is $(\epsilon, \lambda_3, p)$-single confident, with probability $1 - p$, $Y \leq f(x_0 + \epsilon)_{y_0} - \max_{y_j \in [C]: y_j \neq y_0} f(x_0 + \epsilon)_{y_j}$. We note that since

$$\mathbb{E}Y = (1 + \lambda_3)\mu - \lambda_3, \ \mathrm{Var}(Y) = (1 + \lambda_3)^2 s^2,$$

from Lemma D.1,

$$\Pr(Y \leq 0) \leq \frac{s^2}{2(\mu - (1 + \lambda_3^{-1})^{-1})^2}.$$

Thus,

$$\Pr(F(x_0 + \epsilon) = y_0) = 1 - \Pr(F(x_0 + \epsilon) \neq y_0)$$

$$= 1 - \Pr\left(f(x_0 + \epsilon)_{y_0} - \max_{y_j \in [C]: y_j \neq y_0} f(x_0 + \epsilon)_{y_j} < 0\right)$$

$$\geq 1 - p - \Pr(Y \leq 0)$$

$$\geq 1 - p - \frac{s^2}{2(\mu - (1 + \lambda_3^{-1})^{-1})^2}.$$

□

### D.1.2 CERTIFIED ROBUSTNESS FOR ENSEMBLES

Now we are ready to prove the certified robustness of the Weighted Ensemble and Max-Margin Ensemble (Theorems 4 and 5).

In the following text, we first define statistical margins for both WE and MME, and point out their connections to the notion of $(\epsilon, p)$-Statistical Robust. Then, we reason about the expectation, variance, and tail bounds of the statistical margins. Finally, we derive the certified robustness from the statistical margins.

**Definition D.2** ($\hat{X}_1$; Statistical Margin for WE $\mathcal{M}_{\mathrm{WE}}$). Let $\mathcal{M}_{\mathrm{WE}}$ be Weighted Ensemble defined over base models $\{f_i\}_{i=1}^N$ with weights $\{w_i\}_{i=1}^N$. Suppose $\mathcal{M}_{\mathrm{WE}}$ is $(\epsilon, \lambda_1, p)$-WE-confident. We define random variable $\hat{X}_1$ which is depended by random variable $\epsilon$:

$$\hat{X}_1(\epsilon) := (1 + \lambda_1) \sum_{j=1}^N w_j f_j(\boldsymbol{x}_0 + \boldsymbol{\epsilon})_{y_0} - \lambda_1 \|\boldsymbol{w}\|_1. \tag{31}$$

**Definition D.3** ($\hat{X}_2$; Statistical Margin for MME $\mathcal{M}_{\mathrm{MME}}$). Let $\mathcal{M}_{\mathrm{MME}}$ be Max-Margin Ensemble defined over base models $\{f_i\}_{i=1}^N$. Suppose $\mathcal{M}_{\mathrm{MME}}$ is $(\epsilon, \lambda_2, p)$-MME-confident. We define random variable $\hat{X}_2$ which is depended by random variable $\epsilon$:

$$\hat{X}_2(\epsilon) := (1 + \lambda_2) \left( \max_{i \in [N]} f_i(\boldsymbol{x}_0 + \boldsymbol{\epsilon})_{y_0} + \min_{i \in [N]} f_i(\boldsymbol{x}_0 + \boldsymbol{\epsilon})_{y_0} \right) - 2\lambda_2. \tag{32}$$

We have the following observation:

**Lemma D.2.** *For Weighted Ensemble,*

$$\Pr_{\boldsymbol{\epsilon}} \left( \mathcal{M}_{\mathrm{WE}}(\boldsymbol{x}_0 + \boldsymbol{\epsilon}) = y_0 \right) \geq 1 - p - \Pr_{\boldsymbol{\epsilon}} \left( \hat{X}_1(\epsilon) < 0 \right).$$

*For Max-Margin Ensemble,*

$$\Pr_{\boldsymbol{\epsilon}} \left( \mathcal{M}_{\mathrm{MME}}(\boldsymbol{x}_0 + \boldsymbol{\epsilon}) = y_0 \right) \geq 1 - p - \Pr_{\boldsymbol{\epsilon}} \left( \hat{X}_2(\epsilon) < 0 \right).$$

*Proof of Lemma D.2.* (1) For Weighted Ensemble, we define the random variable $X_1$:

$$X_1(\epsilon) := \min_{y_i \in [C]: y_i \neq y_0} \sum_{j=1}^N w_j f_j^{y_0/y_i}(\boldsymbol{x}_0 + \boldsymbol{\epsilon}).$$

Since $\mathcal{M}_{\mathrm{WE}}$ is $(\epsilon, \lambda_1, p)$-WE-confident, from Definition 6, with probability $1 - p$, we have

$$X_1(\epsilon) \geq \sum_{j=1}^N w_j \left( f_j(\boldsymbol{x}_0 + \boldsymbol{\epsilon})_{y_0} - \lambda_2(1 - f_j(\boldsymbol{x}_0 + \boldsymbol{\epsilon})_{y_0}) \right)$$

$$= (1 + \lambda_2) \sum_{j=1}^N w_j f_j(\boldsymbol{x}_0 + \boldsymbol{\epsilon})_{y_0} - \lambda_1 \|\boldsymbol{w}\|_1 = \hat{X}_1(\epsilon).$$

Therefore,

$$\Pr_{\boldsymbol{\epsilon}}(\mathcal{M}_{\mathrm{WE}}(\boldsymbol{x}_0 + \boldsymbol{\epsilon}) = y_0) = \Pr_{\boldsymbol{\epsilon}}(X_1(\epsilon) \geq 0) \geq 1 - p - \Pr_{\boldsymbol{\epsilon}}(\hat{X}_2(\epsilon) < 0).$$

(2) For Max-Margin Ensemble, we define the random variable $X_2$:

$$X_2(\epsilon) := \max_{i \in [N]} \min_{y_i \in [C]: y_i \neq y_0} f_i^{y_0/y_i}(\boldsymbol{x}_0 + \boldsymbol{\epsilon}) + \min_{i \in [N]} \min_{y_i \in [C]: y_i \neq y_0} f_i^{y_0/y_i}(\boldsymbol{x}_0 + \boldsymbol{\epsilon}).$$

Similarly, since $\mathcal{M}_{\mathrm{MME}}$ is $(\epsilon, \lambda_2, p)$-MME-confident, from Definition 7, with probability $1 - p$, we have

$$X_2(\epsilon) \geq \max_{i \in [N]} \left( f_i(\boldsymbol{x}_0 + \boldsymbol{\epsilon})_{y_0} - \lambda_2(1 - f_i(\boldsymbol{x}_0 + \boldsymbol{\epsilon})_{y_0}) \right) + \min_{i \in [N]} \left( f_i(\boldsymbol{x}_0 + \boldsymbol{\epsilon})_{y_0} - \lambda_2(1 - f_i(\boldsymbol{x}_0 + \boldsymbol{\epsilon})_{y_0}) \right)$$

$$= (1 + \lambda_2) \left( \max_{i \in [N]} f_i(\boldsymbol{x}_0 + \boldsymbol{\epsilon})_{y_0} + \min_{i \in [N]} f_i(\boldsymbol{x}_0 + \boldsymbol{\epsilon})_{y_0} \right) - 2\lambda_2 = \hat{X}_2(\epsilon).$$

Moreover, from Lemma B.1, we know

$$\Pr_{\boldsymbol{\epsilon}}(\mathcal{M}(\boldsymbol{x}_0 + \boldsymbol{\epsilon}) = y_0) \geq \Pr_{\boldsymbol{\epsilon}}(X_2(\boldsymbol{\epsilon}) \geq 0) \geq 1 - p - \Pr_{\boldsymbol{\epsilon}}(\hat{X}_2(\boldsymbol{\epsilon}) < 0).$$

$\square$

As the result, to quantify the statistical robustness of two types of ensembles, we can analyze the distribution of statistical margins $\hat{X}_1$ and $\hat{X}_2$.

**Lemma D.3** (Expectation and variance of $\hat{X}_1$ and $\hat{X}_2$). *Let $\hat{X}_1$ and $\hat{X}_2$ be defined by Definition D.2 and Definition D.3 respectively. Assume $\{f_i(\boldsymbol{x}_0 + \boldsymbol{\epsilon})_{y_0}\}_{i=1}^N$ are i.i.d. and follow symmetric distribution with mean $\mu$ and variance $s^2$. Define $s_f^2 = \mathrm{Var}(\min_{i \in [N]} f_i(\boldsymbol{x}_0 + \boldsymbol{\epsilon})_{y_0})$. We have*

$$\mathbb{E}\,\hat{X}_1(\boldsymbol{\epsilon}) = (1 + \lambda_1)\|\boldsymbol{w}\|_1 \mu - \lambda_1 \|\boldsymbol{w}\|_1, \quad \mathrm{Var}\,\hat{X}_1(\boldsymbol{\epsilon}) = (1 + \lambda_1)^2 s^2 \|\boldsymbol{w}\|_2^2,$$
$$\mathbb{E}\,\hat{X}_2(\boldsymbol{\epsilon}) = 2(1 + \lambda_2)\mu - 2\lambda_2, \qquad \mathrm{Var}\,\hat{X}_2(\boldsymbol{\epsilon}) \leq 4(1 + \lambda_2)^2 s_f^2.$$

*Proof of Lemma D.3.*

$$\mathbb{E}\hat{X}_1(\boldsymbol{\epsilon}) = (1 + \lambda_1) \sum_{j=1}^N \mathbb{E} w_j f_j(\boldsymbol{x}_0 + \boldsymbol{\epsilon})_{y_0} - \lambda_1 \|\boldsymbol{w}\|_1 = (1 + \lambda_1)\|\boldsymbol{w}\|_1 \mu - \lambda_1 \|\boldsymbol{w}\|_1;$$

$$\mathrm{Var}\hat{X}_1(\boldsymbol{\epsilon}) = (1 + \lambda_1)^2 \sum_{j=1}^N w_j^2 \mathrm{Var}(f_j(\boldsymbol{x}_0 + \boldsymbol{\epsilon})_{y_0}) = (1 + \lambda_1)^2 s^2 \|\boldsymbol{w}\|_2^2.$$

According to the symmetric distribution property of $\{f_i(\boldsymbol{x}_0 + \boldsymbol{\epsilon})_{y_0}\}_{i=1}^N$, we have

$$\mathbb{E}\,\hat{X}_2(\boldsymbol{\epsilon}) = \mathbb{E}(1 + \lambda_2)\left(\max_{i \in [N]} f_i(\boldsymbol{x}_0 + \boldsymbol{\epsilon})_{y_0} + \min_{i \in [N]} f_i(\boldsymbol{x}_0 + \boldsymbol{\epsilon})_{y_0}\right) - 2\lambda_2$$
$$= 2(1 + \lambda_2)\mu - 2\lambda_2.$$

Also, due the symmetry, we have

$$\mathrm{Var}\left(\min_{i \in [N]} f_i(\boldsymbol{x}_0 + \boldsymbol{\epsilon})_{y_0}\right) = \mathrm{Var}\left(\max_{i \in [N]} f_i(\boldsymbol{x}_0 + \boldsymbol{\epsilon})_{y_0}\right) = s_f^2.$$

As a result,

$$\mathrm{Var}\,\hat{X}_2(\boldsymbol{\epsilon}) \leq (1 + \lambda_2)^2 \cdot 4s_f^2.$$

$\square$

From Lemma D.3, now with Lemma D.1, we are ready to derive the statistical robustness lower bound for WE and MME.

**Theorem 4** (Certified Robustness for WE). *Let $\boldsymbol{\epsilon}$ be a random variable supported on $\mathbb{R}^d$. Let $\mathcal{M}_{\mathrm{WE}}$ be a Weighted Ensemble defined over $\{f_i\}_{i=1}^N$ with weights $\{w_i\}_{i=1}^N$. The $\mathcal{M}_{\mathrm{WE}}$ is $(\boldsymbol{\epsilon}, \lambda_1, p)$-WE confident. Let $\boldsymbol{x}_0 \in \mathbb{R}^d$ be the input with ground-truth label $y_0 \in [C]$. Assume $\{f_i(\boldsymbol{x}_0 + \boldsymbol{\epsilon})_{y_0}\}_{i=1}^N$, the confidence scores across base models for label $y_0$, are i.i.d. and follow symmetric distribution with mean $\mu$ and variance $s^2$, where $\mu > (1 + \lambda_1^{-1})^{-1}$. We have*

$$\Pr_{\boldsymbol{\epsilon}}(\mathcal{M}_{\mathrm{WE}}(\boldsymbol{x}_0 + \boldsymbol{\epsilon}) = y_0) \geq 1 - p - \frac{\|\boldsymbol{w}\|_2^2}{\|\boldsymbol{w}\|_1^2} \cdot \frac{s^2}{2\left(\mu - \left(1 + \lambda_1^{-1}\right)^{-1}\right)^2}. \tag{14}$$

**Theorem 5** (Certified Robustness for MME). *Let $\boldsymbol{\epsilon}$ be a random variable. Let $\mathcal{M}_{\mathrm{MME}}$ be a Max-Margin Ensemble defined over $\{f_i\}_{i=1}^N$. The $\mathcal{M}_{\mathrm{MME}}$ is $(\boldsymbol{\epsilon}, \lambda_2, p)$-MME confident. Let $\boldsymbol{x}_0 \in \mathbb{R}^d$ be the input with ground-truth label $y_0 \in [C]$. Assume $\{f_i(\boldsymbol{x}_0 + \boldsymbol{\epsilon})_{y_0}\}_{i=1}^N$, the confidence scores across base models for label $y_0$, are i.i.d. and follow symmetric distribution with mean $\mu$ where $\mu > (1 + \lambda_2^{-1})^{-1}$. Define $s_f^2 = \mathrm{Var}(\min_{i \in [N]} f_i(\boldsymbol{x}_0 + \boldsymbol{\epsilon})_{y_0})$. We have*

$$\Pr_{\boldsymbol{\epsilon}}(\mathcal{M}_{\mathrm{MME}}(\boldsymbol{x}_0 + \boldsymbol{\epsilon}) = y_0) \geq 1 - p - \frac{s_f^2}{2\left(\mu - \left(1 + \lambda_2^{-1}\right)^{-1}\right)^2}. \tag{15}$$

*Proof of Theorems 4 and 5.* Combining Lemmas D.1 to D.3, we get the theorem. □

*Remark.* Theorems 4 and 5 provide two statistical robustness lower bounds for both types of ensembles, which is shown to be able to translate to certified robustness.

For the Weighted Ensemble, noticing that $\hat{X}_1$ is the weighted sum of several independent variables, we can further apply McDiarmid's Inequality to get another bound

$$\Pr_{\epsilon}(\mathcal{M}_{\mathrm{WE}}(\boldsymbol{x}_0 + \boldsymbol{\epsilon}) = y_0) \geq 1 - p - \exp\left(-2\frac{\|\boldsymbol{w}\|_1^2}{\|\boldsymbol{w}\|_2^2}\left(\mu - \left(1 + \lambda_1^{-1}\right)^{-1}\right)^2\right),$$

which is tighter than Equation (14) when $\|\boldsymbol{w}\|_1^2/\|\boldsymbol{w}\|_2^2$ is large. For average weighted ensemble, $\|\boldsymbol{w}\|_1^2/\|\boldsymbol{w}\|_2^2 = N$. Thus, when $N$ is large, this bound is tighter.

Both theorems are applicable under the i.i.d. assumption of confidence scores. The another assumption $\mu > \max\{(1 + \lambda_1^{-1})^{-1}, (1 + \lambda_2^{-1})^{-1}\}$ insures that both ensembles have higher probability of predicting the true class rather than other classes, i.e., the ensembles have non-trivial clean accuracy.

## D.2 COMPARISON OF CERTIFIED ROBUSTNESS

We first show and prove an important lemma. Then, based on the lemma and Theorems 4 and 5, we derive the comparison corollary.

**Lemma D.4.** *For $\mu, \lambda_1, \lambda_2, C > 0$, when $\max\{\lambda_1/(1 + \lambda_1), \lambda_2/(1 + \lambda_2)\} < \mu \leq 1$, and $C < 1$, we have*

$$\frac{\mu - (\lambda_2^{-1} + 1)^{-1}}{\mu - (\lambda_1^{-1} + 1)^{-1}} < C \iff \frac{\lambda_1}{\lambda_2} < \lambda_2^{-1}\left(\left(C^{-1}\left(\mu - \frac{\lambda_2}{1 + \lambda_2}\right) + 1 - \mu\right)^{-1} - 1\right). \quad (33)$$

*Proof of Lemma D.4.*

$$\frac{\mu - (\lambda_2^{-1} + 1)^{-1}}{\mu - (\lambda_1^{-1} + 1)^{-1}} < C$$

$$\iff \frac{1}{\lambda_2^{-1} + 1} - \frac{C}{\lambda_1^{-1} + 1} > \mu(1 - C)$$

$$\iff \frac{\lambda_1/\lambda_2}{\lambda_2^{-1} + \lambda_1/\lambda_2} < \frac{C^{-1}}{\lambda_2^{-1} + 1} - \mu(C^{-1} - 1)$$

$$\iff \frac{\lambda_1}{\lambda_2}\left(1 - \mu + C^{-1}\left(\mu - \frac{1}{\lambda_2^{-1} + 1}\right)\right) < \lambda_2^{-1}\left(C^{-1}\left(\frac{1}{\lambda_2^{-1} + 1} - \mu\right) + \mu\right)$$

$$\iff \frac{\lambda_1}{\lambda_2} < \lambda_2^{-1}\frac{C^{-1}\left(\frac{1}{\lambda_2^{-1} + 1} - \mu\right) + \mu}{C^{-1}\left(\mu - \frac{1}{\lambda_2^{-1} + 1}\right) + 1 - \mu}$$

$$\iff \frac{\lambda_1}{\lambda_2} < \lambda_2^{-1}\left(\left(C^{-1}\left(\mu - \frac{\lambda_2}{1 + \lambda_2}\right) + 1 - \mu\right)^{-1} - 1\right).$$

□

Now we can show and prove the comparison corollary.

**Corollary 2** (Comparison of Certified Robustness). *Let $\epsilon$ be a random variable supported on $\mathbb{R}^d$. Over base models $\{f_i\}_{i=1}^N$, let $\mathcal{M}_{\mathrm{MME}}$ be Max-Margin Ensemble, and $\mathcal{M}_{\mathrm{WE}}$ the Weighted Ensemble with weights $\{w_i\}_{i=1}^N$. Let $\boldsymbol{x}_0 \in \mathbb{R}^d$ be the input with ground-truth label $y_0 \in [C]$. Assume $\{f_i(\boldsymbol{x}_0 + \boldsymbol{\epsilon})_{y_0}\}_{i=1}^N$, the confidence scores across base models for label $y_0$, are i.i.d, and follow symmetric distribution with mean $\mu$ and variance $s^2$, where $\mu > \max\{(1 + \lambda_1^{-1})^{-1}, (1 + \lambda_2^{-1})^{-1}\}$. Define $s_f^2 = \mathrm{Var}(\min_{i \in [N]} f_i(\boldsymbol{x}_0 + \boldsymbol{\epsilon})_{y_0})$ and assume $s_f < s$.*

- *When*

$$\frac{\lambda_1}{\lambda_2} < \lambda_2^{-1} \left( \left( \frac{s}{s_f} \left( \mu - \left( 1 + \lambda_2^{-1} \right)^{-1} \right) + 1 - \mu \right)^{-1} - 1 \right), \tag{16}$$

for any weights $\{w_i\}_{i=1}^N$, $\mathcal{M}_{\mathrm{WE}}$ has higher certified robustness than $\mathcal{M}_{\mathrm{MME}}$.

- *When*

$$\frac{\lambda_1}{\lambda_2} > \lambda_2^{-1} \left( \left( \frac{s}{\sqrt{N} s_f} \left( \mu - \left( 1 + \lambda_2^{-1} \right)^{-1} \right) + 1 - \mu \right)^{-1} - 1 \right), \tag{17}$$

for any weights $\{w_i\}_{i=1}^N$, $\mathcal{M}_{\mathrm{MME}}$ has higher certified robustness than $\mathcal{M}_{\mathrm{WE}}$.

*Here, the certified robustness is given by Theorems 4 and 5.*

*Proof of Corollary 2.* (1) According to Lemma D.4, we have

$$\frac{\lambda_1}{\lambda_2} < \lambda_2^{-1} \left( \left( \frac{s}{s_f} \left( \mu - \left( 1 + \lambda_2^{-1} \right)^{-1} \right) + 1 - \mu \right)^{-1} - 1 \right)$$

$$\Longrightarrow \frac{\mu - (\lambda_2^{-1} + 1)^{-1}}{\mu - (\lambda_1^{-1} + 1)^{-1}} < \frac{s_f}{s}$$

$$\Longrightarrow \sqrt{\frac{\|\boldsymbol{w}\|_2^2}{\|\boldsymbol{w}\|_1^2}} \frac{\mu - (\lambda_2^{-1} + 1)^{-1}}{\mu - (\lambda_1^{-1} + 1)^{-1}} < \frac{s_f}{s}$$

$$\Longrightarrow \frac{\|\boldsymbol{w}\|_2^2}{\|\boldsymbol{w}\|_1^2} \cdot \frac{s^2}{2 \left( \mu - \left( 1 + \lambda_1^{-1} \right)^{-1} \right)^2} < \frac{s_f^2}{2 \left( \mu - \left( 1 + \lambda_2^{-1} \right)^{-1} \right)^2}.$$

According to Theorems 4 and 5, we know the RHS in Equation (14) is larger than the RHS in Equation (15), i.e., $\mathcal{M}_{\mathrm{WE}}$ has higher certified robustnesss than $\mathcal{M}_{\mathrm{MME}}$.

(2) According to Lemma D.4, we have

$$\frac{\lambda_1}{\lambda_2} > \lambda_2^{-1} \left( \left( \frac{s}{\sqrt{N} s_f} \left( \mu - \left( 1 + \lambda_2^{-1} \right)^{-1} \right) + 1 - \mu \right)^{-1} - 1 \right)$$

$$\Longrightarrow \frac{\mu - (\lambda_2^{-1} + 1)^{-1}}{\mu - (\lambda_1^{-1} + 1)^{-1}} > \frac{\sqrt{N} s_f}{s}$$

$$\Longrightarrow \sqrt{\frac{\|\boldsymbol{w}\|_2^2}{\|\boldsymbol{w}\|_1^2}} \frac{\mu - (\lambda_2^{-1} + 1)^{-1}}{\mu - (\lambda_1^{-1} + 1)^{-1}} > \frac{s_f}{s}$$

$$\Longrightarrow \frac{\|\boldsymbol{w}\|_2^2}{\|\boldsymbol{w}\|_1^2} \cdot \frac{s^2}{2 \left( \mu - \left( 1 + \lambda_1^{-1} \right)^{-1} \right)^2} > \frac{s_f^2}{2 \left( \mu - \left( 1 + \lambda_2^{-1} \right)^{-1} \right)^2}.$$

According to Theorems 4 and 5, we know the RHS in Equation (15) is larger than the RHS in Equation (14), i.e., $\mathcal{M}_{\mathrm{MME}}$ has higher certified robustnesss than $\mathcal{M}_{\mathrm{WE}}$. $\qquad\square$

*Remark.* As we can observe in the proof, there is a gap between Equation (16) and Equation (17) — when $\lambda_1/\lambda_2$ lies in between RHS of Equation (16) and RHS of Equation (17), it is undetermined which ensemble protocol has higher robustness. Indeed, this uncertainty is caused by the adjustable weights $\{w_i\}_{i=1}^N$ of the Weighted Ensemble. If we only consider the average ensemble, then this gap is closed:

$$\frac{\lambda_1}{\lambda_2} \underset{\mathcal{M}_{\mathrm{WE}} \text{ more robust}}{\overset{\mathcal{M}_{\mathrm{MME}} \text{ more robust}}{\gtrless}} \lambda_2^{-1} \left( \left( \frac{s}{\sqrt{N} s_f} \left( \mu - \left( 1 + \lambda_2^{-1} \right)^{-1} \right) + 1 - \mu \right)^{-1} - 1 \right).$$

In the corollary, we assume that $s_f < s$. Note that $s^2$ is the variance of single variable and $s_f^2$ is the variance of minimum of $N$ i.i.d. variables. For common symmetry distributions, along with the increase of $N$, $s_f$ shrinks in the order of $O(1/N^B)$ where $B \in (0, 2]$. Thus, as long as $N$ is large, the assumption will always hold. An exception is that when these random variables follow the exponential distribution, where $s_f$ does not shrink along with the increase of $N$. However, since these random variables are confidence scores which are in $[0, 1]$, they cannot obey exponential distribution.

### D.3    A Concrete Case: Uniform Distribution

As shown by Saremi & Srivastava (2020) (Remark 2.1), when the input dimension $d$ is large, the Gaussian noise $\epsilon \sim \mathcal{N}(0, \sigma^2 I_d) \approx \text{Unif}(\sigma\sqrt{d}S_{d-1})$, i.e., $x_0 + \epsilon$ is highly *uniformly distributed* on the $(d-1)$-sphere centered at $x_0$. Motivated by this, we study the case where the confidence scores $\{f_i(x_0 + \epsilon)_{y_0}\}_{i=1}^N$ are also uniformly distributed.

Under this additional assumption, we can further make the certified robustness for the single model and both ensembles more concrete.

#### D.3.1    Certified Robustness for Single Model

**Proposition D.2** (Certified Robustness for Single Model under Uniform Distribution). *Let $\epsilon$ be a random variable supported on $\mathbb{R}^d$. Let $F$ be a classification model, which is $(\epsilon, \lambda_3, p)$-single confident. Let $x_0 \in \mathbb{R}^d$ be the input with ground-truth $y_0 \in [C]$. Suppose $f(x_0 + \epsilon)_{y_0}$ is uniformly distributed in $[a, b]$. We have*

$$\Pr_{\epsilon}(F(x_0 + \epsilon) = y_0) \geq 1 - p - \text{clip}\left(\frac{1/(1 + \lambda_3^{-1}) - a}{b - a}\right),$$

*where*    $\text{clip}(x) = \max(\min(x, 1), 0).$

*Proof of Proposition D.2.* We consider the distribution of quantity $Y := f(x_0 + \epsilon)_{y_0} - \lambda_3(1 - f(x_0 + \epsilon)_{y_0})$. Since the model $F$ is $(\epsilon, \lambda_3, p)$-single confident, with probability $1 - p$, $Y \leq f(x_0 + \epsilon)_{y_0} - \max_{y_j \in [C]: y_j \neq y_0} f(x_0 + \epsilon)_{y_j}$. At the same time, because $f(x_0 + \epsilon)_{y_0}$ follows the distribution $\mathcal{U}([a, b])$,

$$Y = (1 + \lambda_3)f(x_0 + \epsilon)_{y_0} - \lambda_3$$

follows the distribution $\mathcal{U}([(1 + \lambda_3)a - \lambda_3, (1 + \lambda_3)b - \lambda_3])$. Therefore,

$$\Pr(Y \leq 0) = \text{clip}\left(\frac{\lambda_3 - (1 + \lambda_3)a}{(1 + \lambda_3)(b - a)}\right).$$

As the result,

$$\Pr\left(f(x_0 + \epsilon)_{y_0} - \max_{y_j \in [C]: y_j \neq y_0} f(x_0 + \epsilon)_{y_j} \leq 0\right) \leq p + \text{clip}\left(\frac{\lambda_3 - (1 + \lambda_3)a}{(1 + \lambda_3)(b - a)}\right),$$

which is exactly

$$\Pr\left(F(x_0 + \epsilon) = y_0\right) \geq 1 - p - \text{clip}\left(\frac{\lambda_3 - (1 + \lambda_3)a}{(1 + \lambda_3)(b - a)}\right) = 1 - p - \text{clip}\left(\frac{1/(1 + \lambda_3^{-1}) - a}{b - a}\right).$$

$\square$

#### D.3.2    Certified Robustness for Ensembles

Still, we define $\hat{X}_1(\epsilon)$ and $\hat{X}_2(\epsilon)$ according to Definitions D.2 and D.3. Under the uniform distribution assumption, we have the following lemma.

**Lemma D.5** (Expectation and Variance of $\hat{X}_1$ and $\hat{X}_2$ under Uniform Distribution). *Let $\hat{X}_1$ and $\hat{X}_2$ be defined by Definition D.2 and Definition D.3 respectively. Assume that under the distribution of $\epsilon$,*

*the base models' confidence scores for true class $\{f_i(\boldsymbol{x}_0 + \boldsymbol{\epsilon})_{y_0}\}_{i=1}^N$ are pairwise i.i.d and uniformly distributed in range $[a, b]$. We have*

$$\mathbb{E}\,\hat{X}_1(\boldsymbol{\epsilon}) = \frac{1}{2}(1+\lambda_1)\|\boldsymbol{w}\|_1(a+b) - \lambda_1\|\boldsymbol{w}\|_1, \quad \mathrm{Var}\,\hat{X}_1(\boldsymbol{\epsilon}) = \frac{1}{12}(1+\lambda_1)^2\|\boldsymbol{w}\|_2^2(b-a)^2,$$

$$\mathbb{E}\,\hat{X}_2(\boldsymbol{\epsilon}) = (1+\lambda_2)(a+b) - 2\lambda_2, \qquad \mathrm{Var}\,\hat{X}_2(\boldsymbol{\epsilon}) \le (1+\lambda_2)^2\frac{4}{N+1}\left(\frac{2}{N+2} - \frac{1}{N+1}\right)(b-a)^2.$$

*Proof of Lemma D.5.* We start from analyzing $\hat{X}_1$. From the definition

$$\hat{X}_1(\boldsymbol{\epsilon}) := (1+\lambda_1)\sum_{j=1}^N w_j f_j(\boldsymbol{x}_0 + \boldsymbol{\epsilon})_{y_0} - \lambda_1\|\boldsymbol{w}\|_1 \tag{31}$$

where $\{f_i(\boldsymbol{x}_0 + \boldsymbol{\epsilon})_{y_0}\}_{i=1}^N$ are i.i.d. variables obeying uniform distribution $\mathcal{U}([a, b])$,

$$\mathbb{E}\,\hat{X}_1(\boldsymbol{\epsilon}) = (1+\lambda_1)\|\boldsymbol{w}\|_1\frac{a+b}{2} - \lambda_1\|\boldsymbol{w}\|_1 = \frac{1}{2}(1+\lambda_1)\|\boldsymbol{w}\|_1(a+b) - \lambda_1\|\boldsymbol{w}\|_1,$$

$$\mathrm{Var}\,\hat{X}_1(\boldsymbol{\epsilon}) = (1+\lambda_1)^2\sum_{j=1}^N w_j^2\frac{1}{12}(b-a)^2 = \frac{1}{12}(1+\lambda_1)^2\|\boldsymbol{w}\|_2^2(b-a)^2.$$

Now analyze the expectation of $\hat{X}_2$. By the symmetry of uniform distribution, we know

$$\mathbb{E}\,\hat{X}_2(\boldsymbol{\epsilon}) = (1+\lambda_2)\cdot 2\mathbb{E}\,f_i(\boldsymbol{x}_0 + \boldsymbol{\epsilon})_{y_0} - 2\lambda_2 = (1+\lambda_2)(a+b) - 2\lambda_2.$$

To reason about the variance, we need the following fact:

**Fact D.1.** *Let $x_1, x_2, \ldots, x_n$ be uniformly distributed and independent random variables. Specifically, for each $1 \le i \le n$, $\boldsymbol{x}_i \sim \mathcal{U}([a, b])$. Then we have*

$$\mathrm{Var}\left(\min_{1\le i\le n} x_i\right) = \mathrm{Var}\left(\max_{1\le i\le n} x_i\right) = \frac{1}{n+1}\left(\frac{2}{n+2} - \frac{1}{n+1}\right)(b-a)^2.$$

Observing that each i.i.d. $f_i(\boldsymbol{x}_0 + \boldsymbol{\epsilon})_{y_0}$ is exactly identical to $x_i$ in Fact D.1, we have

$$\mathrm{Var}\left(\max_{i\in[N]} f_i(\boldsymbol{x}_0 + \boldsymbol{\epsilon})_{y_0} + \min_{i\in[N]} f_i(\boldsymbol{x}_0 + \boldsymbol{\epsilon})_{y_0}\right) \le \frac{4}{N+1}\left(\frac{2}{N+2} - \frac{1}{N+1}\right)(b-a)^2.$$

Therefore,

$$\mathrm{Var}\,\hat{X}_2(\boldsymbol{\epsilon}) \le (1+\lambda_2)^2\frac{4}{N+1}\left(\frac{2}{N+2} - \frac{1}{N+1}\right)(b-a)^2.$$

$\square$

*Proof of Fact D.1.* From symmetry of uniform distribution, we know $\mathrm{Var}\left(\min_{1\le i\le n} x_i\right) = \mathrm{Var}\left(\max_{1\le i\le n} x_i\right)$. So here we only consider $Y := \min_{1\le i\le n} x_i$. Its CDF $F$ and PDF $f$ can be easily computed:

$$F(y) = 1 - \Pr\left(\min_i x_i \ge y\right) = 1 - \left(\frac{b-y}{b-a}\right)^n, \quad f(y) = F'(y) = n\frac{(b-y)^{n-1}}{(b-a)^n}, \text{ where } y \in [a, b].$$

Hence,

$$\mathbb{E}\,Y = \int_a^b yf(y)\mathrm{d}y = \frac{y(b-y)^n + (n+1)^{-1}(b-y)^{n+1}}{(b-a)^n}\bigg|_b^a = a + \frac{b-a}{n+1},$$

$$\mathbb{E}\,Y^2 = \int_a^b y^2 f(y)\mathrm{d}y = \int_a^b ny^2 \frac{(b-y)^{n-1}}{(b-a)^n}\mathrm{d}y$$

$$= -\left(\frac{b-y}{b-a}\right)^n y^2 \Big|_a^b + 2\int_a^b \left(\frac{b-y}{b-a}\right)^n y\mathrm{d}y$$

$$= -\left(\frac{b-y}{b-a}\right)^n y^2 \Big|_a^b + \frac{2}{n+1}\left(-\frac{(b-y)^{n+1}}{(b-a)^n}y + \int \frac{(b-y)^{n+1}}{(b-a)^n}\mathrm{d}y\right)\Big|_a^b$$

$$= -\left(\frac{b-y}{b-a}\right)^n y^2 \Big|_a^b + \frac{2}{n+1}\left(-\frac{(b-y)^{n+1}}{(b-a)^n}y - \frac{1}{n+2}\frac{(b-y)^{n+2}}{(b-a)^n}\right)\Big|_a^b$$

$$= a^2 + \frac{2}{n+1}(b-a)a + \frac{2}{(n+1)(n+2)}(b-a)^2.$$

As the result, $\mathrm{Var}\,Y = \mathbb{E}Y^2 - (\mathbb{E}Y)^2 = \frac{1}{n+1}\left(\frac{2}{n+2} - \frac{1}{n+1}\right)(b-a)^2.$ $\qquad\square$

Now, similarly, we use Lemma D.1 to derive the statistical robustness lower bound for WE and MME. We omit the proofs since they are direct applications of Lemma D.5, Lemma D.1, and Lemma D.2.

**Theorem D.1** (Certified Robustness for WE under Uniform Distribution). *Let $\mathcal{M}_{\mathrm{WE}}$ be a Weighted Ensemble defined over $\{f_i\}_{i=1}^N$ with weights $\{w_i\}_{i=1}^N$. Let $\boldsymbol{x}_0 \in \mathbb{R}^d$ be the input with ground-truth label $y_0 \in [C]$. Let $\boldsymbol{\epsilon}$ be a random variable supported on $\mathbb{R}^d$. Under the distribution of $\boldsymbol{\epsilon}$, suppose $\{f_i(\boldsymbol{x}_0 + \boldsymbol{\epsilon})_{y_0}\}_{i=1}^N$ are i.i.d. and uniformly distributed in $[a, b]$. The $\mathcal{M}_{\mathrm{WE}}$ is $(\boldsymbol{\epsilon}, \lambda_1, p)$-WE confident. Assume $\frac{a+b}{2} > \frac{1}{1+\lambda_1^{-1}}$. We have*

$$\Pr_{\boldsymbol{\epsilon}}(\mathcal{M}_{\mathrm{WE}}(\boldsymbol{x}_0 + \boldsymbol{\epsilon}) = y_0) \geq 1 - p - \frac{d_{\boldsymbol{w}}K_1^2}{12},$$

$$\text{where} \quad d_{\boldsymbol{w}} = \frac{\|\boldsymbol{w}\|_2^2}{\|\boldsymbol{w}\|_1^2}, \ K_1 = \frac{b-a}{\frac{a+b}{2} - \frac{1}{1+\lambda_1^{-1}}}. \tag{34}$$

**Theorem D.2** (Certified Robustness for MME under Uniform Distribution). *Let $\mathcal{M}_{\mathrm{MME}}$ be a Max-Margin Ensemble over $\{f_i\}_{i=1}^N$. Let $\boldsymbol{x}_0 \in \mathbb{R}^d$ be the input with ground-truth label $y_0 \in [C]$. Let $\boldsymbol{\epsilon}$ be a random variable supported on $\mathbb{R}^d$. Under the distribution of $\boldsymbol{\epsilon}$, suppose $\{f_i(\boldsymbol{x}_0 + \boldsymbol{\epsilon})_{y_0}\}_{i=1}^N$ are i.i.d. and uniformly distributed in $[a, b]$. $\mathcal{M}_{\mathrm{MME}}$ is $(\boldsymbol{\epsilon}, \lambda_2, p)$-MME confident. Assume $\frac{a+b}{2} > \frac{1}{1+\lambda_2^{-1}}$. We have*

$$\Pr_{\boldsymbol{\epsilon}}(\mathcal{M}_{\mathrm{MME}}(\boldsymbol{x}_0 + \boldsymbol{\epsilon}) = y_0) \geq 1 - p - \frac{c_N K_2^2}{4},$$

$$\text{where} \quad c_N = \frac{2}{N+1}\left(\frac{2}{N+2} - \frac{1}{N+1}\right), \ K_2 = \frac{b-a}{\frac{a+b}{2} - \frac{1}{1+\lambda_2^{-1}}}. \tag{35}$$

### D.3.3 COMPARISON

Now under the uniform distribution, we can also have the certified robustness comparison.

**Corollary D.1** (Comparison of Certified Robustness under Uniform Distribution). *Over base models $\{f_i\}_{i=1}^N$, let $\mathcal{M}_{\mathrm{MME}}$ be Max-Margin Ensemble, and $\mathcal{M}_{\mathrm{WE}}$ the Weighted Ensemble with weights $\{w_i\}_{i=1}^N$. Let $\boldsymbol{x}_0 \in \mathbb{R}^d$ be the input with ground-truth label $y_0 \in [C]$. Let $\boldsymbol{\epsilon}$ be a random variable supported on $\mathbb{R}^d$. Under the distribution of $\boldsymbol{\epsilon}$, suppose $\{f_i(\boldsymbol{x}_0 + \boldsymbol{\epsilon})_{y_0}\}_{i=1}^N$ are i.i.d. and uniformly distributed with mean $\mu$. Suppose $\mathcal{M}_{\mathrm{WE}}$ is $(\boldsymbol{\epsilon}, \lambda_1, p)$-WE confident, and $\mathcal{M}_{\mathrm{MME}}$ is $(\boldsymbol{\epsilon}, \lambda_2, p)$-MME confident. Assume $\mu > \max\left\{\frac{1}{1+\lambda_1^{-1}}, \frac{1}{1+\lambda_2^{-1}}\right\}$.*

- *When*

$$\frac{\lambda_1}{\lambda_2} < \lambda_2^{-1}\left(\left((N+1)\sqrt{\frac{N+2}{6N}}\left(\mu - \frac{1}{1+\lambda_2^{-1}}\right) + 1 - \mu\right)^{-1} - 1\right), \tag{36}$$

*$\mathcal{M}_{\mathrm{WE}}$ has higher certified robustness than $\mathcal{M}_{\mathrm{MME}}$.*

- *When*

$$\frac{\lambda_1}{\lambda_2} > \lambda_2^{-1} \left( \left( \frac{N+1}{N} \sqrt{\frac{N+2}{6}} \left( \mu - \frac{1}{1+\lambda_2^{-1}} \right) + 1 - \mu \right)^{-1} - 1 \right), \qquad (37)$$

$\mathcal{M}_{\mathrm{MME}}$ *has higher certified robustness than* $\mathcal{M}_{\mathrm{WE}}$.

- *When*

$$N > 6 \left( 1 - \frac{1}{\mu(1+\lambda_2^{-1})} \right)^{-2} - 2, \qquad (38)$$

*for any* $\lambda_1$, $\mathcal{M}_{\mathrm{MME}}$ *has higher or equal certified robustness than* $\mathcal{M}_{\mathrm{WE}}$.

*Here, the certified robustness is given by Theorems D.1 and D.2.*

*Proof of Corollary D.1.* First, we notice that a uniform distribution with mean $\mu$ can be any distribution $\mathcal{U}([a, b])$ where $(a+b)/2 = \mu$. We replace $\mu$ by $(a+b)/2$.

Then (1) and (2) follow from Lemma D.4 similar to the proof of Corollary 2.

(3) Since

$$N > 6 \left( 1 - \frac{1}{\mu(1+\lambda_2^{-1})} \right)^{-2} - 2 \implies \left( \sqrt{\frac{N+2}{6}} \left( \mu - \frac{1}{1+\lambda_2^{-1}} \right) + 1 - \mu \right)^{-1} < 1$$

$$\implies \left( \frac{N+1}{N} \sqrt{\frac{N+2}{6}} \left( \mu - \frac{1}{1+\lambda_2^{-1}} \right) + 1 - \mu \right)^{-1} < 1,$$

the RHS of Equation (37) is smaller than 0. Thus, for any $\lambda_1$, since $\lambda_1/\lambda_2 > 0$, the Equation (17) is satisfied. According to (2), $\mathcal{M}_{\mathrm{MME}}$ has higher certified robustness than $\mathcal{M}_{\mathrm{WE}}$. □

*Remark.* Comparing to the general corollary (Corollary 2), under the uniform distribution, we have an additional finding that when $N$ is sufficiently large, we will always have higher certified robustness for Max-Margin Ensemble than Weighted Ensemble. This is due to the more efficient variance reduction of Max-Margin Ensemble than Weighted Ensemble. As shown in Lemma D.5, the quantity $\mathrm{Var}\hat{X}(\epsilon)/(\mathbb{E}\hat{X}(\epsilon))^2$ for Weighted Ensemble is $\Omega(1/N)$, while for Max-Margin Ensemble is $O(1/N^2)$. As the result, when $N$ becomes larger, Max-Margin Ensemble has higher certified robustness.

We use uniform assumption here to give an illustration in a specific regime. Since the assumption may not hold exactly in practice, we think it would be an interesting future direction to generalize the analysis to other distributions such as the Gaussian distribution that corresponds to locally linear classifiers. The result from these distribution may be derived from their specific concentration bound for maximum/minimum i.i.d. random variables as discussed at the end of Appendix D.2.

## D.4 NUMERICAL EXPERIMENTS

To validate and give more intuitive explanations for our theorems, we present some numerical experiments.

### D.4.1 ENSEMBLE COMPARISON FROM NUMERICAL SAMPLING

As discussed in Section 3.2, $\lambda_1/\lambda_2$ reflects the transferability across base models. It is challenging to get enough amount of different ensembles of various transferability levels while keeping all other variables controlled. Therefore, we simulate the transferability of ensembles numerically by varying $\lambda_1/\lambda_2$ (see the definitions of $\lambda_1$ and $\lambda_2$ in Definitions 6 and 7), and sampling the confidence scores $\{f_i(\boldsymbol{x}_0 + \boldsymbol{\epsilon})_{y_0}\}$ and $\{\max_{j\in[C]:j\neq y_0} f_i(\boldsymbol{x}_0 + \boldsymbol{\epsilon})_j\}$ under determined $\lambda_1$ and $\lambda_2$. For each level of $\lambda_1/\lambda_2$, with the samples, we compute the certified robust radius $r$ using randomized smoothing (Theorem B.1) and compare the radius difference of Weighted Ensemble and Max-Margin Ensemble. According to Corollary 2 in Section 3.2, we should observe the tendency that along with

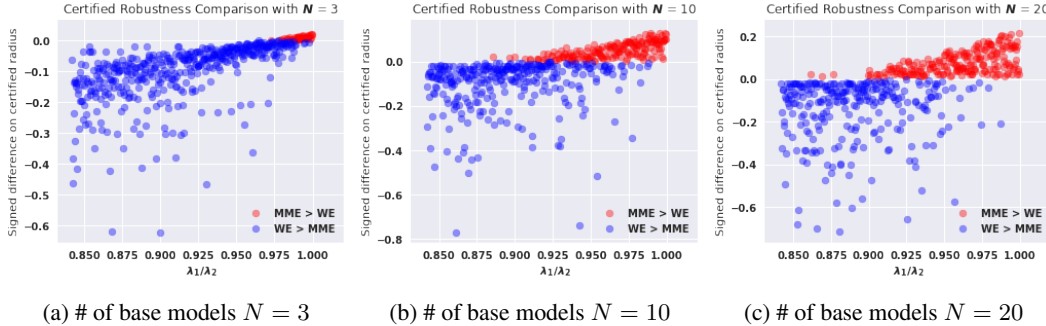

(a) # of base models $N = 3$      (b) # of base models $N = 10$      (c) # of base models $N = 20$

Figure 2: Signed *certified robust radius* difference between MME and WE by $\lambda_1/\lambda_2$ under different numbers of base models $N$. Here we fix $\lambda_2$ to be $0.95$ and uniformly sample $\lambda_1 \in [0.8, 0.95)$. The confidence score for the true class on each base model is uniformly sampled from $[a, b]$, where $a$ is sampled from $[0.3, 1.0)$ and $b$ is sampled from $[a, 1.0)$ uniformly for each instance. **Blue** points correspond to the negative radius difference (i.e., WE has larger radius than MME) and **Red** points correspond to the positive radius difference (i.e., MME has larger radius than WE).

the increase of transferability $\lambda_1/\lambda_2$, Max-Margin Ensemble would gradually become better than Weighted Ensemble.

Figure 2 verifies the trends: with the increase of $\lambda_1/\lambda_2$, MME model tends to achieve higher certified radius than WE model. Moreover, we notice that under the same $\lambda_1/\lambda_2$, with the larger number of base models $N$, the MME tends to be relatively better compared with WE. This is because we sample the confidence score uniformly and under the uniform distribution, MME tends to be better than WE when the number of base models $N$ becomes large, according to Corollary D.1.

The concrete number settings of $\lambda_1, \lambda_2$, and the sampling interval of confidence scores are entailed in the caption of Figure 2.

### D.4.2 ENSEMBLE COMPARISON FROM CERTIFIED ROBUSTNESS PLOTTING

In Corollary D.1, we derive the concrete certified robustness for both ensembles and the single model under i.i.d. and uniform distribution assumption. In fact, from the corollary, we can directly compute the certified robust radius without sampling, as long as we assume the added noise $\epsilon$ is Gaussian. In Figure 3, we plot out such certified robust radius for the single model, the WE, and the MME.

Concretely, in the figure, we assume that the true class confidence score for each base model is i.i.d. and *uniformly distributed* in $[a, b]$. The Weighted Ensemble is $(\epsilon, \lambda_1, 0.01)$-WE confident; the Max-Margin Ensemble is $(\epsilon, \lambda_2, 0.01)$-MME confident; and the single model is $(\epsilon, \lambda_3, 0.01)$-MME confident. We guarantee that $\lambda_1 \leq \lambda_3 \leq \lambda_2$ to simulate the scenario that ensembles are based on the same set of base models to make a fair comparison. We directly apply the results from our analysis (Theorem D.1, Theorem D.2, Proposition D.2) to get the statistical robustness for single model and both ensembles. Then, we leverage Theorem B.1 to get the certified robust radius (with $\sigma = 1.0, N = 100000$ and failing probability $\alpha = 0.001$ which are aligned with realistic setting). The $x$-axis is the number of base models $N$ and the $y$-axis is the certified robustness. We note that $N$ is not applicable to the single model, so we plot the single model's curve by a horizontal red dashed line.

From the figure, we observe that when the number of base models $N$ becomes larger, both ensembles perform much better than the single model. We remark that when $N$ is small, the ensembles have $0$ certified robustness mainly because our theoretical bounds for ensembles are not tight enough with the small $N$. Furthermore, we observe that the Max-Margin Ensemble gradually surpasses Weighted Ensemble when $N$ is large, which conforms to our Corollary D.1. Note that the left sub-figure has smaller transferability $\lambda_1/\lambda_2$ and the right subfigure has larger transferability $\lambda_1/\lambda_2$, it again conforms to our Corollary 2 and discussion in Section 3.2 that in the left subfigure the Weighted Ensemble is relatively more robust than the Max-Margin Ensemble.

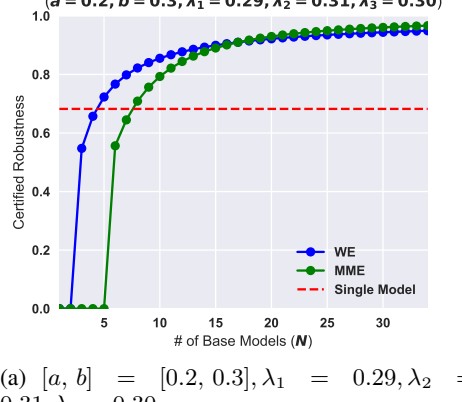
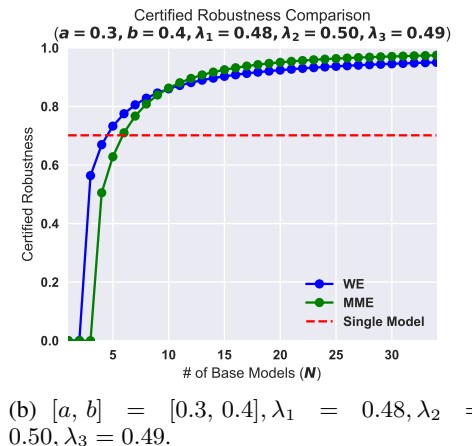

(a) $[a, b] = [0.2, 0.3], \lambda_1 = 0.29, \lambda_2 = 0.31, \lambda_3 = 0.30$.

(b) $[a, b] = [0.3, 0.4], \lambda_1 = 0.48, \lambda_2 = 0.50, \lambda_3 = 0.49$.

Figure 3: Comparison of certified robustness (in terms of certified robust radius) of Max-Margin Ensemble, Weighted Ensemble, and single model under concrete numerical settings. The $y$-axis is the certified robustness and the $x$-axis is the number of base models. The confidence score for the true class is uniformly distributed in $[a, b]$. The Weighted Ensemble (shown by **blue line**) is $(\epsilon, \lambda_1, 0.01)$-WE confident; the Max-Margin Ensemble (shown by **green line**) is $(\epsilon, \lambda_2, 0.01)$-MME confident; and the single model (shown by **red line**) is $(\epsilon, \lambda_3, 0.01)$-MME confident.

### D.4.3 ENSEMBLE COMPARISON FROM REALISTIC DATA

We study the correlation between transferability $\lambda_1/\lambda_2$ and whether Weighted Ensemble or Max-Margin Ensemble is more certifiably robust using realistic data.

By varying the hyper-parameters of DRT, we find out a setting where over the same set of base models, Weighted Ensemble and Max-Margin Ensemble have similar certified robustness, i.e., for about half of the test set samples, WE is more robust; for another half, MME is more robust. We collect $1,000$ test set samples in total. Then, for each test set sample, we compute the transferability $\lambda_1/\lambda_2$ and whether WE or MME has the higher certified robust radius. We remark that $\lambda_1$ and $\lambda_2$ are difficult to be practically estimated so we use the average confidence portion as the proxy:

- For WE,

$$\lambda_1 = \mathbb{E}_{\epsilon} \frac{\max_{y_j \in [C]: y_j \neq y_0} \sum_{i=1}^{N} w_i f_i(\boldsymbol{x}_0 + \boldsymbol{\epsilon})_{y_j}}{\sum_{i=1}^{N} w_i (1 - f_i(\boldsymbol{x}_0 + \boldsymbol{\epsilon})_{y_0})}.$$

- For MME,

$$\lambda_2 = \mathbb{E}_{\epsilon} \max_{i \in [N]} \frac{\max_{y_j \in [C]: y_j \neq y_0} f_i(\boldsymbol{x}_0 + \boldsymbol{\epsilon})_{y_j}}{(1 - f_i(\boldsymbol{x}_0 + \boldsymbol{\epsilon})_{y_0})}.$$

Now we study the correlation between

$$X := \lambda_1/\lambda_2 - \text{RHS of Equation (17) and } Y := \mathbf{1}[\text{MME has higher certified robustness}].$$

To do so, we draw the ROC curve where the threshold on $X$ does binary classification on $Y$. The curve and the AUC score is shown in Figure 4. From the ROC curve, we find that $X$ and $Y$ are apparently positively correlated since $\text{AUC} = 0.66 > 0.5$, which again verifies Corollary 2. We remark that besides $X$, other factors such as non-symmetric or non-i.i.d. confidence score distribution may also play a role.

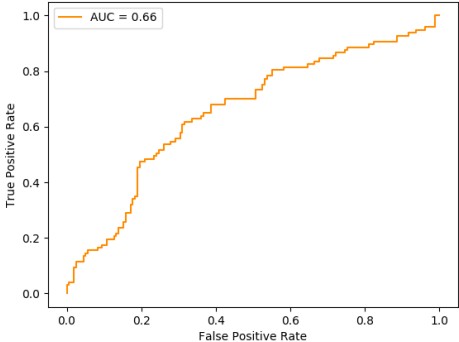

Figure 4: ROC curve of the $\mathbf{1}[\text{MME has higher certified robustness}]$ classification task with the threshold variable $X$.

## E    EXPERIMENT DETAILS

**Evaluation metric**: We use the *certified test set accuracy* at each radius $r$ as our evaluation metric, which is defined as the fraction of the test set samples that the smoothed classifier can certify the robustness within the $L_2$ ball of radius $r$. Since the computation of the accurate value of this metric is intractable, we report the *approximate certified test accuracy* (Cohen et al., 2019) sampled through the Monte Carlo procedure. For each sample, the robustness certification holds with probability at least $1 - \alpha$. Following the literature, we choose $\alpha = 0.001$, $n_0 = 100$ for Monte Carlo sampling during prediction phase, and $n = 100,000$ for Monte Carlo sampling during certification phase.

### E.1    MNIST

**Baseline models' hyper-parameter configuration**: We choose the number of noise samples per instance $m = 2$ and Gaussian smoothing parameter $\sigma \in \{0.25, 0.5, 1.0\}$ for all the training methods. For SmoothAdv, we consider the attack to be 10-step $L_2$ PGD attack with perturbation scale $\delta = 1.0$ without pretraining and unlabelled data augmentation. We reproduced similar results to their paper by using their open-sourced code[1].

**Training details**: We use LeNet architecture and trained each base model for 90 epochs. For the training optimizer, we use the SGD-momentum with the initial learning rate $\alpha = 0.01$. The learning rate is decayed for every 30-epochs with decay ratio $\gamma = 0.1$ and the batch size equals to 256. For DRT experiments, we start our training with the small learning rate $\alpha = 5 \times 10^{-4}$ and finetune the base models for another 90 epochs. During the training, we find too large regularization weights may cause the model's training collapse on MNIST. We turn to use small DRT hyper-parameters $\rho_1, \rho_2$ and report the one with the best certified accuracy on each radius $r$.

**Certified accuracy curve**: Figure 6 and Figure 7 show the certified accuracy curve with different base model type and smoothing parameter $\sigma$ among the range of the radius $r$. We can notice that while simply applying MME or WE can improve the certified accuracy slightly, DRT could help boost the certified accuracy on each radius $r$ with a significant scale.

**DRT hyper-parameters**: We investigate the DRT hyper-parameters $\rho_1 \in \{0.1, 0.2, 0.3, 0.5, 1.0\}$ and $\rho_2 \in \{0.5, 1.0, 2.0, 5.0\}$ corresponding to different smoothing parameter $\sigma \in \{0.25, 0.5, 1.0\}$. Here we put the detailed results for every hyper-parameter setting in Tables 4 to 6 and bold the numbers with the highest certified accuracy on each radius $r$ among all the tables with different $\sigma$'s. From the experiments, we found that the GD loss's weight $\rho_1$ can have the major influence on the ensemble model's functionality: if we choose larger $\rho_1$, the model will achieve slightly worse certified accuracy on small radius $r$, but better certified accuracy on large $r$. We cannot choose too large $\rho_1$ on small $\sigma$ cases (e.g., $\sigma = 0.25$). Otherwise, the training procedure will collapse. Here we show the DRT-based model's *approximate certified accuracy* with different $\rho_1$ in Figure 5.

---

[1] https://github.com/Hadisalman/smoothing-adversarial/

Table 4: DRT-$(\rho_1, \rho_2)$ model's certified accuracy under different radius $r$ on MNIST dataset. Smoothing parameter $\sigma = 0.25$.

| Radius $r$ | $\rho_1$ | $\rho_2$ | 0.00 | 0.25 | 0.50 | 0.75 |
|---|---|---|---|---|---|---|
| Gaussian (Cohen et al., 2019) | - | - | 99.1 | 97.9 | 96.6 | 93.0 |
| SmoothAdv (Salman et al., 2019) | - | - | 99.1 | 98.4 | 97.0 | 96.3 |
| MME (Gaussian) | - | - | 99.2 | 98.4 | 96.8 | 93.6 |
| DRT + MME (Gaussian) | 0.1 | 0.2 | 99.4 | 98.3 | 97.5 | 95.1 |
| | 0.1 | 0.5 | **99.5** | **98.6** | 97.1 | 94.8 |
| | 0.2 | 0.5 | **99.5** | 98.5 | 97.4 | 95.1 |
| MME (SmoothAdv) | - | - | 99.2 | 98.2 | 97.3 | 96.4 |
| DRT + MME (SmoothAdv) | 0.1 | 0.2 | 99.1 | 98.4 | 97.5 | 96.4 |
| | 0.1 | 0.5 | 99.1 | 98.3 | **97.6** | **96.7** |
| | 0.2 | 0.5 | 99.1 | 98.4 | 97.5 | 96.6 |
| WE (Gaussian) | - | - | 99.2 | 98.4 | 96.9 | 93.7 |
| DRT + WE (Gaussian) | 0.1 | 0.2 | **99.5** | 98.4 | 97.3 | 95.1 |
| | 0.1 | 0.5 | **99.5** | **98.6** | 97.1 | 94.9 |
| | 0.2 | 0.5 | **99.5** | 98.5 | 97.3 | 95.3 |
| WE (SmoothAdv) | - | - | 99.2 | 98.2 | 97.4 | 96.4 |
| DRT + WE (SmoothAdv) | 0.1 | 0.2 | 99.1 | 98.4 | 97.5 | 96.5 |
| | 0.1 | 0.5 | 99.1 | 98.2 | **97.6** | 96.6 |
| | 0.2 | 0.5 | 99.0 | 98.4 | 97.5 | **96.7** |

Alternatively, we found that the CM loss's weight $\rho_2$ can have positive influence on model's performance: the larger $\rho_2$ we choose, the better certified accuracy we can get. Choosing large $\rho_2$ does not harm model's functionality too much, but the improvement we received will become marginal. For MNIST, $(\sigma, \rho_1, \rho_2) \in \{(0.25, 0.1, 0.2), (0.5, 0.5, 5.0), (1.0, 1.0, 5.0)\}$ are good combinations.

**Efficiency Analysis**: We regard the *execution time per mini-batch* as our efficiency criterion. For MNIST with batch size equals to 256, DRT with the Gaussian smoothing base model only requires 1.04s to finish one mini-batch training to achieve the comparable results to the SmoothAdv method which requires 1.86s. Moreover, DRT with the SmoothAdv base model requires 2.52s per training batch but achieves much better results. The evaluation is on single NVIDIA GeForce GTX 1080 Ti GPU.

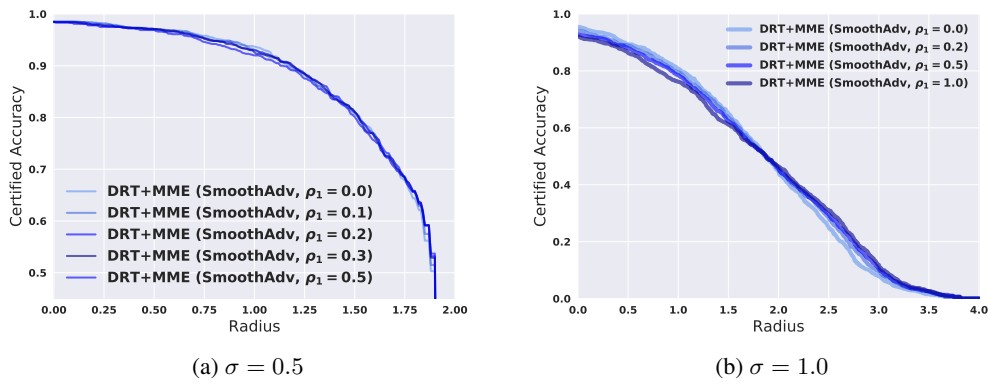

(a) $\sigma = 0.5$                    (b) $\sigma = 1.0$

Figure 5: **Effect of** $\rho_1$: Comparison of *approximate certified accuracy* of DRT models on MNIST with different GD Loss's weight $\rho_1$.

### E.2    CIFAR-10

**Baseline models' hyper-parameter configuration**: We choose the number of noise samples per instance $m = 2$ and Gaussian smoothing parameter $\sigma \in \{0.25, 0.5, 1.0\}$ for all the training methods.

Table 5: DRT-$(\rho_1, \rho_2)$ model's certified accuracy under different radius $r$ on MNIST dataset. Smoothing parameter $\sigma = 0.50$.

| Radius $r$ | $\rho_1$ | $\rho_2$ | 0.00 | 0.25 | 0.50 | 0.75 | 1.00 | 1.25 | 1.50 | 1.75 |
|---|---|---|---|---|---|---|---|---|---|---|
| Gaussain (Cohen et al., 2019) | - | - | 99.0 | 97.7 | 96.4 | 94.7 | 90.0 | 83.0 | 68.2 | 43.5 |
| SmoothAdv (Salman et al., 2019) | - | - | 98.6 | 98.0 | 97.0 | 95.4 | 93.0 | 87.7 | 80.2 | 66.3 |
| MME (Gaussian) | - | - | 99.0 | 97.7 | 96.8 | 94.9 | 90.5 | 84.3 | 69.8 | 48.5 |
| DRT + MME (Gaussian) | 0.2 | 2.0 | 99.1 | 98.4 | 97.2 | 95.2 | 92.6 | 86.5 | 74.3 | 54.1 |
| | 0.2 | 5.0 | 99.1 | **98.6** | 97.1 | 95.3 | 92.6 | 86.2 | 74.0 | 54.3 |
| | 0.5 | 2.0 | 99.2 | 98.3 | 97.4 | 95.5 | 92.1 | 86.4 | 74.7 | 55.6 |
| | 0.5 | 5.0 | 99.0 | 98.2 | 97.3 | 95.1 | 91.6 | 84.8 | 73.7 | 52.4 |
| | 1.0 | 5.0 | 99.1 | 98.2 | 97.2 | 95.2 | 92.2 | 85.8 | 74.4 | 54.4 |
| MME (SmoothAdv) | - | - | 98.6 | 98.0 | 97.0 | 95.5 | 93.2 | 88.1 | 80.6 | 67.8 |
| DRT + MME (SmoothAdv) | 0.1 | 0.5 | 98.4 | 97.8 | 97.0 | 95.5 | 92.7 | 87.7 | 80.9 | 67.9 |
| | 0.1 | 1.0 | 98.4 | 97.9 | 97.0 | 95.5 | 92.9 | 88.1 | 80.8 | 67.2 |
| | 0.1 | 5.0 | 98.5 | 98.2 | 97.0 | 95.4 | 93.1 | 88.4 | 81.2 | 68.3 |
| | 0.2 | 0.5 | 98.4 | 97.7 | 97.2 | 95.3 | 92.3 | 87.7 | 79.3 | 68.4 |
| | 0.2 | 2.0 | 98.4 | 97.6 | 97.1 | 95.3 | 92.3 | 87.8 | 80.2 | 67.7 |
| | 0.2 | 5.0 | 98.4 | 97.8 | 97.1 | 95.2 | 93.0 | 87.9 | 80.3 | 68.3 |
| | 0.3 | 5.0 | 98.4 | 97.5 | 97.1 | 95.0 | 92.4 | 87.7 | 79.7 | 68.3 |
| | 0.5 | 2.0 | 98.5 | 97.3 | 96.6 | 94.3 | 91.6 | 86.7 | 79.5 | 68.6 |
| | 0.5 | 5.0 | 98.4 | 97.5 | 96.9 | 94.6 | 92.0 | 87.5 | 80.1 | 67.8 |
| | 1.0 | 5.0 | 98.2 | 97.3 | 96.8 | 94.5 | 91.9 | 87.4 | 80.0 | 67.6 |
| WE (Gaussian) | - | - | 99.0 | 97.8 | 96.8 | 94.9 | 90.6 | 84.5 | 70.4 | 48.2 |
| DRT + WE (Gaussian) | 0.2 | 2.0 | 99.2 | 98.4 | 97.2 | 95.2 | 92.5 | 86.2 | 74.3 | 53.5 |
| | 0.2 | 5.0 | 99.1 | **98.6** | 97.1 | 95.3 | 92.6 | 86.4 | 74.2 | 54.4 |
| | 0.5 | 2.0 | 99.2 | 98.3 | 97.4 | 95.6 | 92.1 | 86.5 | 74.7 | 55.3 |
| | 0.5 | 5.0 | 99.1 | 98.2 | 97.1 | 95.1 | 91.7 | 85.4 | 73.5 | 51.0 |
| | 1.0 | 5.0 | 99.1 | 98.2 | 97.2 | 95.2 | 92.2 | 85.9 | 75.1 | 55.3 |
| WE (SmoothAdv) | - | - | 98.7 | 98.0 | 97.0 | 95.5 | **93.4** | 88.2 | 81.1 | 67.9 |
| DRT + WE (SmoothAdv) | 0.1 | 0.5 | 98.4 | 97.8 | 97.0 | 95.5 | 92.7 | 87.8 | 80.6 | 68.1 |
| | 0.1 | 1.0 | 98.5 | 97.9 | 97.0 | 95.5 | 93.1 | 88.0 | 81.2 | 67.7 |
| | 0.1 | 5.0 | 98.5 | 98.2 | 97.0 | 95.4 | 93.3 | **88.5** | **81.4** | 68.6 |
| | 0.2 | 0.5 | 98.4 | 97.7 | 97.2 | 95.4 | 92.3 | 87.6 | 79.7 | 68.0 |
| | 0.2 | 2.0 | 98.4 | 97.6 | 97.1 | 95.3 | 92.3 | 87.8 | 80.6 | 68.1 |
| | 0.2 | 5.0 | 98.4 | 97.9 | 97.1 | 95.1 | **93.4** | 88.2 | 80.4 | 69.1 |
| | 0.3 | 5.0 | 98.4 | 97.5 | 97.1 | 95.0 | 92.4 | 87.9 | 79.9 | **69.3** |
| | 0.5 | 2.0 | 98.4 | 97.3 | 96.6 | 94.3 | 91.8 | 86.7 | 79.6 | 68.1 |
| | 0.5 | 5.0 | 98.4 | 97.5 | 96.9 | 94.7 | 92.0 | 87.7 | 79.7 | 67.7 |
| | 1.0 | 5.0 | 98.4 | 97.7 | 97.0 | 95.2 | 92.6 | 88.4 | 81.1 | 68.2 |

For SmoothAdv, we consider the attack to be 10-step $L_2$ PGD attack with perturbation scale $\delta = 1.0$ without pretraining and unlabelled data augmentation. We also reproduced the similar results mentioned in baseline's papers.

**Training details**: We use ResNet-110 architecture for each base model and train them for 150 epochs. During the training, We utilize the SGD-momentum with the initial learning rate $\alpha = 0.1$, which is decayed for every 50-epochs with ratio $\gamma = 0.1$. For DRT experiments, we start the training with the learning rate $\alpha = 5 \times 10^{-3}$ and finetune our base models for another 150 epochs.

**Certified accuracy curve**: Figure 8 and Figure 9 show the certified accuracy curve with different base model type and smoothing parameter $\sigma$ among the range of the radius $r$. We can see the same trends: Applying the MME/WE mechanism will give a slight improvement and DRT can help make this improvement to be significant.

**DRT hyper-parameter**: We studied the DRT hyper-parameter $\rho \in \{0.1, 0.2, 0.5, 1.0, 1.5\}$ and $\rho_2 \in \{0.5, 2.0, 5.0\}$ corresponding to different $\sigma \in \{0.25, 0.5, 1.0\}$ and put the detailed results in Tables 7 to 9. We bold the numbers with the highest certified accuracy on each radius $r$ among all the tables with different $\sigma$'s. The results show the similar conclusion about the choosing of $\rho_1$ and $\rho_2$. $(\sigma, \rho_1, \rho_2) \in \{(0.25, 0.1, 0.5), (0.5, 1.0, 5.0), (1.0, 1.5, 5.0)\}$ could be the good choices on CIFAR-10 dataset.

Table 6: DRT-$(\rho_1, \rho_2)$ model's certified accuracy under different radius $r$ on MNIST dataset. Smoothing parameter $\sigma = 1.00$.

| Radius $r$ | $\rho_1$ | $\rho_2$ | 0.00 | 0.25 | 0.50 | 0.75 | 1.00 | 1.25 | 1.50 | 1.75 | 2.00 | 2.25 | 2.50 |
|---|---|---|---|---|---|---|---|---|---|---|---|---|---|
| Gaussain Smoothing | - | - | 96.5 | 94.3 | 91.1 | 87.0 | 80.2 | 71.8 | 60.1 | 46.6 | 33.0 | 20.5 | 11.5 |
| SmoothAdv | - | - | 95.3 | 93.5 | 89.3 | 85.6 | 80.4 | 72.8 | 63.9 | 54.6 | 43.2 | 34.3 | 24.0 |
| MME (Gaussian) | - | - | 96.4 | 94.8 | 91.3 | 87.7 | 80.8 | 73.5 | 61.0 | 48.8 | 34.7 | 23.4 | 12.7 |
| | 0.5 | 2.0 | 96.0 | 93.9 | 90.1 | 86.3 | 80.7 | 73.2 | 63.0 | 52.0 | 38.9 | 26.9 | 15.6 |
| DRT + MME (Gaussian) | 0.5 | 5.0 | 95.8 | 94.1 | 90.0 | 86.6 | 80.4 | 72.9 | 62.4 | 51.3 | 40.0 | 27.8 | 16.5 |
| | 1.0 | 5.0 | 95.3 | 93.1 | 89.7 | 85.8 | 80.0 | 72.7 | 62.9 | 52.0 | 39.8 | 28.5 | 17.6 |
| MME (SmoothAdv) | - | - | 95.4 | 93.4 | 89.3 | 86.1 | 80.7 | 73.1 | 65.0 | 55.0 | 44.8 | 35.0 | 25.2 |
| | 0.2 | 2.0 | 94.1 | 91.9 | 88.6 | 84.5 | 79.4 | 72.4 | 63.4 | 54.0 | 45.0 | 36.6 | 27.3 |
| | 0.2 | 5.0 | 94.1 | 91.6 | 88.9 | 84.4 | 79.3 | 72.3 | 63.2 | 54.2 | 46.1 | 36.9 | 28.5 |
| DRT + MME (SmoothAdv) | 0.5 | 2.0 | 92.8 | 91.3 | 87.7 | 83.2 | 77.3 | 71.2 | 62.2 | 53.3 | 45.5 | 37.0 | 29.7 |
| | 0.5 | 5.0 | 92.5 | 91.2 | 88.0 | 83.5 | 78.5 | 71.2 | 62.2 | 53.8 | 45.2 | 37.7 | 29.2 |
| | 1.0 | 5.0 | 92.1 | 90.0 | 86.3 | 81.3 | 76.2 | 69.4 | 61.1 | 54.0 | **46.4** | **38.6** | **31.1** |
| WE (Gaussian) | - | - | 96.3 | 94.9 | 91.3 | 87.7 | 80.7 | 73.5 | 61.1 | 49.0 | 35.2 | 23.7 | 12.9 |
| | 0.5 | 2.0 | 95.9 | 93.9 | 90.2 | 86.3 | 80.7 | 73.2 | 63.2 | 51.9 | 38.6 | 27.0 | 15.5 |
| WE + MME (Gaussian) | 0.5 | 5.0 | 95.9 | 94.1 | 90.0 | 86.4 | 80.4 | 73.1 | 62.3 | 51.7 | 39.8 | 27.5 | 16.4 |
| | 1.0 | 5.0 | 95.4 | 93.1 | 89.7 | 85.8 | 80.0 | 72.7 | 62.9 | 52.1 | 39.9 | 28.5 | 17.8 |
| WE (SmoothAdv) | - | - | 95.2 | 93.4 | 89.4 | 86.2 | 80.8 | 73.3 | 64.8 | 55.1 | 44.7 | 35.2 | 24.9 |
| | 0.2 | 2.0 | 94.2 | 91.9 | 88.6 | 84.5 | 79.6 | 72.5 | 63.7 | 53.9 | 44.9 | 36.4 | 27.3 |
| | 0.2 | 5.0 | 94.2 | 91.6 | 88.9 | 84.4 | 79.3 | 72.5 | 63.3 | 54.3 | 45.9 | 36.9 | 28.7 |
| DRT + MME (SmoothAdv) | 0.5 | 2.0 | 92.6 | 91.3 | 87.7 | 83.1 | 77.5 | 71.1 | 62.4 | 53.3 | 45.3 | 36.7 | 29.3 |
| | 0.5 | 5.0 | 92.5 | 91.2 | 88.0 | 83.4 | 78.5 | 71.1 | 62.3 | 53.7 | 45.3 | 37.8 | 29.5 |
| | 1.0 | 5.0 | 92.1 | 90.0 | 86.4 | 81.4 | 76.3 | 69.7 | 61.1 | 54.0 | **46.4** | 38.4 | 31.0 |

Table 7: DRT-$(\rho_1, \rho_2)$ model's certified accuracy under different radius $r$ on CIFAR-10 dataset. Smoothing parameter $\sigma = 0.25$.

| Radius $r$ | $\rho_1$ | $\rho_2$ | 0.00 | 0.25 | 0.50 | 0.75 |
|---|---|---|---|---|---|---|
| Gaussain Smoothing | - | - | 78.9 | 64.4 | 47.4 | 30.6 |
| SmoothAdv | - | - | 68.9 | 61.0 | 54.4 | 45.7 |
| MME (Gaussian) | - | - | 80.8 | 68.2 | 53.4 | 37.4 |
| | 0.1 | 0.5 | 81.4 | **70.4** | 57.6 | 43.4 |
| DRT + MME (Gaussian) | 0.2 | 0.5 | 78.8 | 69.2 | 57.8 | 43.8 |
| | 0.5 | 2.0 | 73.3 | 61.7 | 51.0 | 39.3 |
| | 0.5 | 5.0 | 66.2 | 57.1 | 46.2 | 34.4 |
| MME (SmoothAdv) | - | - | 71.4 | 64.5 | 57.6 | 48.4 |
| | 0.1 | 0.5 | 72.6 | 67.2 | **60.2** | 50.3 |
| DRT + MME (SmoothAdv) | 0.2 | 0.5 | 71.8 | 66.5 | 59.3 | 50.4 |
| | 0.5 | 0.5 | 68.2 | 64.3 | 58.2 | 48.9 |
| WE (Gaussian) | - | - | 80.7 | 68.3 | 53.6 | 37.5 |
| | 0.1 | 0.5 | **81.5** | **70.4** | 57.7 | 43.4 |
| DRT + WE (Gaussian) | 0.2 | 0.5 | 78.8 | 69.3 | 57.9 | 44.0 |
| | 0.5 | 2.0 | 73.4 | 61.7 | 51.0 | 39.2 |
| | 0.5 | 5.0 | 66.2 | 57.1 | 46.1 | 34.5 |
| WE (SmoothAdv) | - | - | 71.8 | 64.6 | 57.8 | 48.5 |
| | 0.1 | 0.5 | 72.6 | 67.0 | **60.2** | 50.3 |
| DRT + WE (SmoothAdv) | 0.2 | 0.5 | 71.9 | 66.5 | 59.4 | **50.5** |
| | 0.5 | 0.5 | 68.2 | 64.3 | 58.4 | 49.1 |

**Efficiency Analysis**: We also use the *execution time per mini-batch* as our efficiency criterion. For CIFAR-10 with batch size equals to 256, DRT with the Gaussian smoothing base model requires 3.82s to finish one mini-batch training to achieve the competitive results to 10-step PGD attack based SmoothAdv method which requires 6.39s. All the models are trained in parallel on 4 NVIDIA GeForce GTX 1080 Ti GPUs.

Table 8: DRT-$(\rho_1, \rho_2)$ model's certified accuracy under different radius $r$ on CIFAR-10 dataset. Smoothing parameter $\sigma = 0.50$.

| Radius $r$ | $\rho_1$ | $\rho_2$ | 0.00 | 0.25 | 0.50 | 0.75 | 1.00 | 1.25 | 1.50 | 1.75 |
|---|---|---|---|---|---|---|---|---|---|---|
| Gaussain Smoothing | - | - | 68.2 | 57.1 | 44.9 | 33.7 | 23.1 | 16.3 | 10.0 | 5.4 |
| SmoothAdv | - | - | 60.6 | 54.2 | 47.9 | 41.2 | 34.8 | 28.5 | 21.9 | 17.1 |
| MME (Gaussian) | - | - | 69.5 | 59.6 | 47.3 | 38.4 | 29.0 | 19.6 | 13.3 | 7.6 |
| | 0.2 | 2.0 | 69.7 | 61.0 | 50.9 | 40.3 | 30.8 | 22.5 | 15.8 | 10.0 |
| | 0.2 | 5.0 | 68.0 | 59.9 | 50.0 | 40.8 | 30.1 | 22.1 | 15.2 | 9.6 |
| DRT + MME (Gaussian) | 0.5 | 2.0 | 67.8 | 58.5 | 49.0 | 39.9 | 31.6 | 23.4 | 16.1 | 10.2 |
| | 0.5 | 5.0 | 65.5 | 58.4 | 49.0 | 40.1 | 31.2 | 23.6 | 16.5 | 10.2 |
| | 1.0 | 2.0 | 64.5 | 55.8 | 47.5 | 39.4 | 31.1 | 23.6 | 14.8 | 9.3 |
| | 1.0 | 5.0 | 62.2 | 54.1 | 46.5 | 38.8 | 29.7 | 22.8 | 16.6 | 11.0 |
| MME (SmoothAdv) | - | - | 61.0 | 54.8 | 48.7 | 42.2 | 36.2 | 29.8 | 23.9 | 19.1 |
| | 0.2 | 5.0 | 62.2 | 56.4 | 50.3 | 43.4 | 37.5 | 26.7 | 24.6 | 19.4 |
| DRT + MME (SmoothAdv) | 0.5 | 5.0 | 61.9 | 56.2 | 50.3 | 43.5 | 37.6 | 31.8 | 24.8 | 19.6 |
| | 1.0 | 5.0 | 61.4 | 55.9 | 50.0 | 43.2 | 37.4 | 32.0 | 25.4 | 19.9 |
| WE (Gaussian) | - | - | 69.4 | 59.7 | 47.5 | 38.4 | 29.2 | 19.7 | 13.3 | 7.5 |
| | 0.2 | 2.0 | 69.7 | 61.2 | 50.8 | 40.2 | 30.8 | 22.4 | 15.9 | 10.0 |
| | 0.2 | 5.0 | 68.0 | 59.9 | 50.1 | 40.8 | 30.1 | 22.1 | 15.4 | 9.7 |
| DRT + WE (Gaussian) | 0.5 | 2.0 | 67.8 | 58.5 | 49.2 | 39.8 | 31.7 | 23.5 | 16.2 | 10.4 |
| | 0.5 | 5.0 | 65.5 | 58.4 | 49.1 | 40.3 | 31.3 | 24.2 | 16.4 | 10.3 |
| | 1.0 | 2.0 | 64.6 | 55.9 | 47.5 | 39.6 | 31.0 | 24.0 | 14.8 | 9.4 |
| | 1.0 | 5.0 | 62.3 | 54.2 | 46.6 | 38.8 | 29.8 | 22.9 | 16.6 | 10.9 |
| WE (SmoothAdv) | - | - | 61.1 | 54.8 | 48.8 | 42.3 | 36.2 | 29.6 | 24.2 | 19.0 |
| | 0.2 | 5.0 | 62.2 | 56.3 | 50.3 | 43.4 | 37.5 | 26.9 | 24.7 | 19.3 |
| DRT + WE (SmoothAdv) | 0.5 | 5.0 | 61.9 | 56.2 | 50.2 | 43.4 | **37.9** | 31.8 | 25.0 | 19.6 |
| | 1.0 | 5.0 | 61.5 | 56.0 | 50.1 | 43.3 | 37.5 | **32.2** | **25.6** | 19.9 |

Table 9: DRT-$(\rho_1, \rho_2)$ model's certified accuracy under different radius $r$ on CIFAR-10 dataset. Smoothing parameter $\sigma = 1.00$.

| Radius $r$ | $\rho_1$ | $\rho_2$ | 0.00 | 0.25 | 0.50 | 0.75 | 1.00 | 1.25 | 1.50 | 1.75 | 2.00 |
|---|---|---|---|---|---|---|---|---|---|---|---|
| Gaussain Smoothing | - | - | 48.9 | 42.7 | 35.4 | 28.7 | 22.8 | 18.3 | 13.6 | 10.5 | 7.3 |
| SmoothAdv | - | - | 47.8 | 43.3 | 39.5 | 34.6 | 30.3 | 25.0 | 21.2 | 18.2 | 15.7 |
| MME (Gaussian) | - | - | 50.2 | 44.0 | 37.5 | 30.9 | 24.1 | 19.3 | 15.6 | 11.6 | 8.8 |
| | 0.5 | 5.0 | 49.4 | 44.2 | 37.8 | 31.6 | 25.4 | 22.6 | 18.2 | 14.4 | 12.4 |
| DRT + MME (Gaussian) | 1.0 | 5.0 | 49.8 | 44.4 | 39.0 | 31.6 | 25.6 | 22.6 | 18.2 | 15.0 | 12.0 |
| | 1.5 | 5.0 | 48.0 | 42.4 | 36.4 | 30.4 | 26.2 | 22.0 | 18.4 | 15.4 | 12.8 |
| MME (SmoothAdv) | - | - | 48.2 | 43.7 | 40.1 | 35.4 | 31.3 | 26.2 | 22.6 | 19.5 | 16.2 |
| | 0.2 | 5.0 | 48.2 | 43.9 | 40.1 | 35.4 | 31.5 | 26.7 | 22.9 | 19.8 | 16.8 |
| DRT + MME (SmoothAdv) | 0.5 | 5.0 | 48.1 | 43.8 | 40.3 | 35.7 | 31.8 | 26.9 | 23.1 | 20.1 | 17.5 |
| | 1.0 | 5.0 | 47.9 | 43.6 | 39.8 | 35.6 | 31.7 | 26.9 | 23.2 | 20.2 | 17.6 |
| | 1.5 | 5.0 | 47.8 | 43.4 | 39.5 | 35.4 | 31.6 | 26.7 | 23.1 | **20.4** | 18.1 |
| WE (Gaussian) | - | - | 50.4 | 44.1 | 37.5 | 30.9 | 24.2 | 19.2 | 15.9 | 11.8 | 8.9 |
| | 0.5 | 5.0 | 49.5 | 44.3 | 37.8 | 31.8 | 25.6 | 22.5 | 18.2 | 14.4 | 12.3 |
| DRT + WE (Gaussian) | 1.0 | 5.0 | 49.8 | 44.4 | 39.1 | 31.7 | 25.6 | 22.8 | 18.4 | 15.1 | 12.1 |
| | 1.5 | 5.0 | 48.2 | 42.5 | 36.6 | 30.4 | 26.1 | 22.1 | 18.2 | 15.7 | 12.6 |
| WE (SmoothAdv) | - | - | 48.2 | 43.7 | 40.2 | 35.4 | 31.5 | 26.2 | 22.7 | 19.6 | 16.0 |
| | 0.2 | 5.0 | 48.2 | 43.8 | 40.2 | 35.3 | 31.6 | 26.8 | 23.0 | 19.9 | 16.7 |
| DRT + WE (SmoothAdv) | 0.5 | 5.0 | 48.2 | 43.8 | 40.5 | 35.7 | 31.9 | 26.8 | 23.3 | 20.2 | 17.5 |
| | 1.0 | 5.0 | 47.8 | 43.6 | 39.9 | 35.5 | 31.7 | 26.9 | 23.3 | 20.1 | 17.5 |
| | 1.5 | 5.0 | 47.8 | 43.4 | 39.6 | 35.4 | 31.4 | 26.7 | 23.0 | **20.4** | **18.0** |

## E.3 IMAGENET

For ImageNet, we utilize ResNet-50 architecture and train each base model for 90 epochs using the SGD-momentum optimizer. The initial learning rate $\alpha = 0.1$. The learning rate is decayed for every 30-epochs with decay ratio $\gamma = 0.1$. We tried different Gaussian smoothing parameter $\sigma \in \{0.50, 1.00\}$, and consider the best hyper-parameter configuration for different $\sigma$ in the baseline models. We explored the DRT hyper-parameter $\rho_1 \in \{0.5, 1.0, 1.5\}, \rho_2 \in \{1.0, 2.0, 5.0\}$ in our experiments and started with the learning rate $\alpha = 5 \times 10^{-3}$ during the DRT. Table 10 shows the

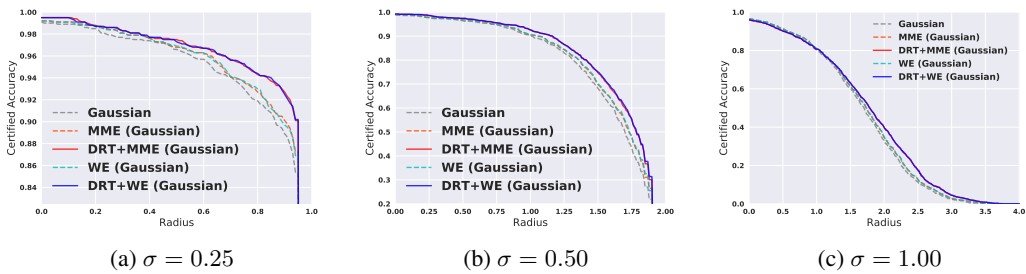

Figure 6: Comparison of *approximate certified accuracy* among different Gaussian-smoothing based methods with various smoothing parameter $\sigma \in \{0.25, 0.50, 1.00\}$ on MNIST.

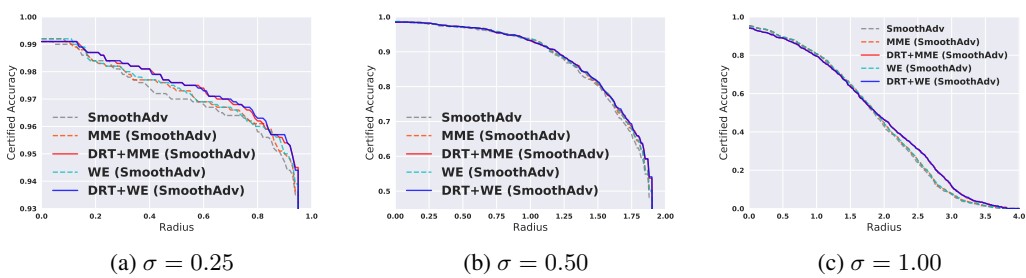

Figure 7: Comparison of *approximate certified accuracy* among different SmoothAdv based methods with various smoothing parameter $\sigma \in \{0.25, 0.50, 1.00\}$ on MNIST.

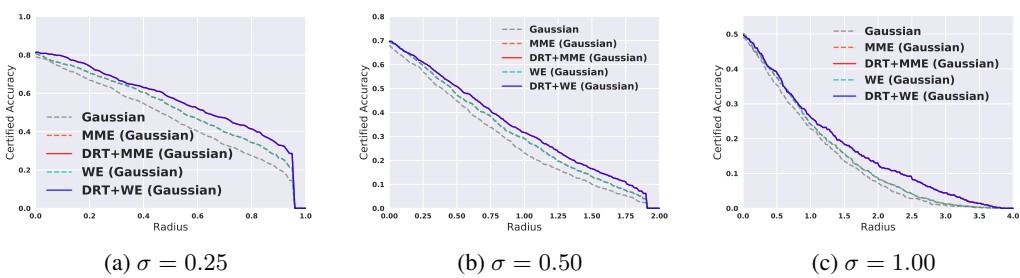

Figure 8: Comparison of *approximate certified accuracy* among different Gaussian-smoothing based methods with various smoothing parameter $\sigma \in \{0.25, 0.50, 1.00\}$ on CIFAR-10.

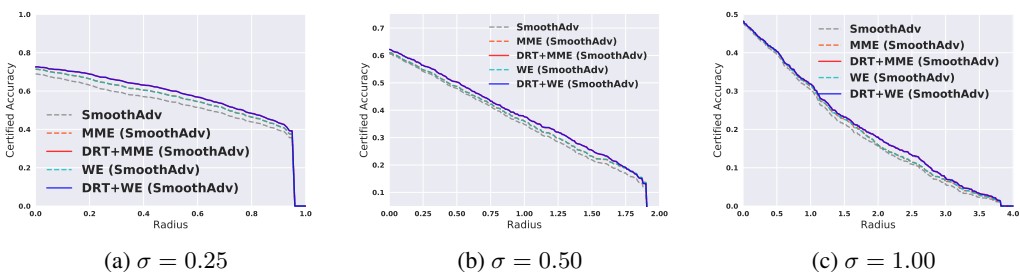

Figure 9: Comparison of *approximate certified accuracy* among different SmoothAdv based methods with various smoothing parameter $\sigma \in \{0.25, 0.50, 1.00\}$ on CIFAR-10.

results. We observe the same trends that MME or WE can improve the certified robustness on each radius $r$. Due to the computation cost of training the DRT ensemble model on ImageNet, we can

Table 10: The certified accuracy under different radius $r$ for ImageNet dataset.

| Radius $r$ | 0.00 | 0.50 | 1.00 | 1.50 | 2.00 | 2.50 | 3.00 |
|---|---|---|---|---|---|---|---|
| Gaussian (Cohen et al., 2019) | 57 | 46 | 37 | 29 | 19 | 15 | 12 |
| SmoothAdv (Salman et al., 2019) | 54 | 49 | 43 | 37 | 27 | 25 | 20 |
| MME (Gaussian) | 58 | 47 | 38 | 31 | 21 | 16 | 14 |
| DRT + MME (Gaussian) | 52 | 46 | 42 | 34 | 24 | 19 | 18 |
| MME (SmoothAdv) | 55 | 50 | 44 | 38 | 27 | 26 | 21 |
| DRT + MME (SmoothAdv) | 49 | 44 | 42 | 37 | 29 | 27 | 22 |
| WE (Gaussian) | 58 | 47 | 38 | 31 | 21 | 17 | 14 |
| DRT + WE (Gaussian) | 52 | 46 | 41 | 33 | 24 | 19 | 18 |
| WE (SmoothAdv) | 55 | 50 | 44 | 38 | 28 | 26 | 22 |
| DRT + WE (SmoothAdv) | 49 | 44 | 42 | 36 | 29 | 27 | 22 |

Table 11: Certified accuracy comparison with other baseline methods for MNIST dataset.

| Radius $r$ | 0.00 | 0.25 | 0.50 | 0.75 | 1.00 | 1.25 | 1.50 | 1.75 | 2.00 | 2.25 | 2.50 |
|---|---|---|---|---|---|---|---|---|---|---|---|
| Gaussian (Cohen et al., 2019) | 99.1 | 97.9 | 96.6 | 94.7 | 90.0 | 83.0 | 68.2 | 46.6 | 33.0 | 20.5 | 11.5 |
| SmoothAdv (Salman et al., 2019) | 99.1 | 98.4 | 97.0 | 96.3 | 93.0 | 87.7 | 80.2 | 66.3 | 43.2 | 34.3 | 24.0 |
| MACER (Zhai et al., 2019) | 99.2 | 98.5 | 97.4 | 94.6 | 90.2 | 83.5 | 72.4 | 54.4 | 36.6 | 26.4 | 16.5 |
| Stability (Li et al., 2019) | 99.3 | **98.6** | 97.1 | 93.8 | 90.7 | 83.2 | 69.2 | 46.8 | 33.1 | 20.0 | 11.2 |
| **D**iversity-**R**egularized **T**raining | **99.5** | **98.6** | **97.6** | **96.7** | **93.4** | **88.5** | **83.3** | **69.6** | **48.3** | **39.4** | **33.5** |

Table 12: Certified accuracy comparison with other baseline methods for CIFAR-10 dataset.

| Radius $r$ | 0.00 | 0.25 | 0.50 | 0.75 | 1.00 | 1.25 | 1.50 | 1.75 | 2.00 |
|---|---|---|---|---|---|---|---|---|---|
| Gaussian (Cohen et al., 2019) | 78.9 | 64.4 | 47.4 | 33.7 | 23.1 | 18.3 | 13.6 | 10.5 | 7.3 |
| SmoothAdv (Salman et al., 2019) | 68.9 | 61.0 | 54.4 | 45.7 | 34.8 | 28.5 | 21.9 | 18.2 | 15.7 |
| MACER (Zhai et al., 2019) | 79.5 | 68.8 | 55.6 | 42.3 | 35.0 | 27.5 | 23.4 | 20.4 | 17.5 |
| Stability (Li et al., 2019) | 72.4 | 58.2 | 43.4 | 27.5 | 23.9 | 16.0 | 15.6 | 11.4 | 7.8 |
| SWEEN (Liu et al., 2020) | **84.2** | **72.0** | **60.3** | 46.6 | 37.2 | 29.2 | 24.6 | 22.0 | 19.1 |
| **D**iversity-**R**egularized **T**raining | 81.5 | 70.4 | 60.2 | **50.5** | **39.5** | **36.0** | **30.4** | **24.1** | **20.3** |

only try one sub-optimal hyper-parameter setting for DRT and the result shows that DRT cannot bring more improvements on MME/WE results. We plan to select more suitable hyper-parameters for DRT to obtain better results on the ImageNet task.

### E.4 COMPARISON WITH OTHER BASELINES

We compare our **D**iversity-**R**egularized **T**raining with other baselines on MNIST and CIFAR-10 dataset. Baselines are: (1) **Gaussian smoothing** (Cohen et al., 2019): Training a smoothed classifier by applying Gaussian augmentation. (2) **SmoothAdv** (Salman et al., 2019): Integrating adversarial training on the soft approximation. (3) **MACER** (Zhai et al., 2019): Adding the regularization term to maximize the certified radius $R = \frac{\sigma}{2}(p_A - p_B)$ on training instances. (4) **Stability** (Li et al., 2019): Adding the adversarial perturbation by maintaining the Stability on classification results. (5) **SWEEN** (Liu et al., 2020): Building the weighted ensemble model by composing the base models of different architectures.

For all baselines, we choose the best certified accuracy for each radius $r$ as reported in their papers. We do not consider the model trained by additional (unlabeled) data. Results are shown in Table 11 and Table 12. We can see that we achieve the best certified accuracy on every $r$ on MNIST dataset and achieve comparable or better results compared to the best certified accuracy on CIFAR-10 dataset.

Table 13: Certified accuracy achieved by training with GD Loss (GDL) or Confidence Margin Loss (CML) for CIFAR-10 dataset.

| Radius $r$ | 0.00 | 0.25 | 0.50 | 0.75 | 1.00 | 1.25 | 1.50 | 1.75 | 2.00 |
|---|---|---|---|---|---|---|---|---|---|
| MME (Gaussian) | 80.8 | 68.2 | 53.4 | 38.4 | 29.0 | 19.6 | 15.6 | 11.6 | 8.8 |
| GDL + MME (Gaussian) | 81.0 | 69.0 | 55.6 | 41.9 | 30.4 | 24.8 | 20.1 | 16.9 | 14.7 |
| CML + MME (Gaussian) | 81.2 | 69.4 | 54.4 | 39.6 | 29.2 | 21.6 | 17.0 | 13.1 | 12.8 |
| DRT + MME (Gaussian) | 81.4 | 70.4 | 57.8 | 43.8 | 31.6 | 26.2 | 22.4 | 18.8 | 16.6 |

## E.5 THE EFFECTS OF GD LOSS AND CONFIDENCE MARGIN LOSS

We studied the effects of GD Loss and Confidence Margin Loss separately by setting $\rho_1 = 0$ or $\rho_2 = 0$ but tuning another parameter only. We did this ablation study on CIFAR-10 dataset with ensemble of Gaussian-smoothed base models and the results are shown in Table 13.

We observed that both GD Loss (GDL) and Confidence Margin Loss (CML) could have positive effects on improving the certified accuracy while GDL plays a major role in the larger radius. While combining these two regularization loss together as our DRT loss, the ensemble model could achieve the best certified accuracy among all the radii.

