# OpenReview forum: "On the Certified Robustness for Ensemble Models and Beyond"
_ICLR.cc/2021/Conference — Reject_

### Official Review · AnonReviewer4 · 2020-10-28
**Difficult to Follow**

**Rating:** 5
**Confidence:** 2

**Review:**


This manuscript provides proofs for certified robustness for ensembles and proposes a new approach called Diversity Regularized Training (DRT) based on the theoretical findings. In addition to the standard loss term, DRT contains two regularization terms: (i) gradient diversity (GD) loss, and (ii) confidence margin loss (CM). These loss terms encourage the joint gradient difference for each model pair and large margin between the true and runner-up classes for base models. The authors discuss some theoretical findings in detail and demonstrate the performance of DRT on several datasets. I find the work valuable in the sense that it nicely combines ideas of ensembling and certified robustness. However, it was difficult to follow all the theoretical results and how they motivated the main finding (DRT) was not clear. Below are my comments/questions:

- I want to thank the authors for nice summary in Appendix B1 and B2 on certified robustness and their map to ensembles.

- Theorem 1, which serves as the foundation for the paper, assumes that either best prediction or the runner-up prediction is true class for any base model. I wonder how realistic this assumption is.

- Also in theorem 1, I am confused that both f and y have index i. The number of base models does not need to match the number of classes.

- In proof for Theorem 2, it could be helpful to mention each step. For instance, it was not clear to me where Lagrangian reminder was applied. Also, I recommend proving necessary and sufficient conditions separately.

- In theorem 3, N is assumed to be 2. Is it possible to extend this to N > 2? So, discussion on such this could be helpful.

- In the first paragraph on page 5, I do not follow which equation was meant in RHS.

- These statements on page 5 are not clear to me: "... and leads to higher certified ensemble robustness." and "Thus, increasing confidence margins can lead to higher ensemble robustness."

- For GD loss, why only pairwise summations were considered? Isn't it easier to look at overall diversity of gradients?

- All pair-wise computation of quantities in regularizer terms should come with some computational complexity. It would be nice to include a discussion on this.

- Results for Salman et al., 2019 on Table 2 do not match the paper of Salman et al., 2019 and imagenet results are not convincing. I understand ImageNet data can take too long but should we worry that DRT requires more hyper-parameter tuning? Some discussion on this would be helpful.

---

> ### Author Response · Authors · 2020-11-14
> **Response to Reviewer #4 (1/2)**
>
> Thanks for your inspiring comments and appreciation of our work. We answer the questions below.
>
> \> It was difficult to follow all the theoretical results and how they motivated the main finding (DRT) was not clear.
> Thanks for the suggestions! Below we make a brief summary and provide intuition for the theoretical results, and we will also add them to our revision.
>
> For the theoretical results in our paper, we start by analyzing the general robustness condition of two types of ensembles: WE and MME (Section 2, Proposition 1, and Theorem 1). Based on Taylor expansion, for either ensemble, we can see that to achieve high certified robustness, we need to reduce the L2 magnitude of the joint gradient vector (i.e., increase gradient diversity) and increase the confidence margin between the true label and any other label (Section 2, Theorem 2 and Theorem 3). Furthermore, we compare the certified robust radius between both ensemble models and the base models (Section 2, Corollary 1). Based on these two factors: increase gradient diversity & increase the confidence margin, we propose the two regularization terms: GD loss and CM loss respectively. The DRT is composed of these two terms.
>
> Then in Section 3, we further analyze the certified robustness under the randomized smoothing setting. Here, when randomized smoothing is applied, we can bound the smoothness of the smoothed classifier and thus derive a computable certified robust radius based on the statistical robustness bound (Section 3, Definition 3). Note that this certified robust radius is not tight, i.e., it is usually smaller than the actual robust radius but it is computable. Since the radius is computed from the statistical robustness bound, we analyze the statistical robustness bound for both ensembles (Section 3, Theorem 4 & 5). The results show that whether WE or MME is better depends on the transferability among base models (Corollary 2). These theorems are mainly derived from the concentration bounds of the average of independent random variables. Then the results (Theorem 4 & 5) justify the use of DRT, especially the CM loss, as shown at the end of Section 3.1.
>
> In Section 4, we empirically evaluate the DRT and show it achieves state-of-the-art certified robustness compared with other single-model training approaches and ensemble training approaches.
>
> \> Top-class and runner-up prediction assumption.
>
> We observe that when the ensemble gives the right prediction for a clean sample $x$, there must exist a radius $r$ around $x$, such that the true label is still the runner-up or top class in $r$-ball as long as the confidence change is continuous. Therefore, the assumption just impedes us to reason about arbitrarily large $r$, but we can still reason about moderately large $r$.
>
> Moreover, for WE we do not have this assumption, and for the theorems for the smoothed version (in Section 3) we do not need this assumption either.
>
> \> The number of base models does not need to match the number of classes.
>
> Thanks for pointing this out. Yes, you are right and we will make this clear in our revision. Currently, $C$ represents #class, and $N$ the # base models. In Theorem 1, the superscript $y_i$ indexes the classes other than $y_0$ for each base model $i$ individually.
>
> \> In the proof for Theorem 2, it could be helpful to mention each step.
>
> Thanks for the comment, and we will make the proof in Theorem 2 clear in the revision.
>
> \> In theorem 3, $N$ is assumed to be 2. Is it possible to extend this to $N > 2$?
>
> In the smoothed version in Section 3, we extend the results beyond $N = 2$ in Theorems 4, 5, and Corollary 2.
>
> In this vanilla version, WE can be extended to $N = 2$, and for MME, the challenge is the “maximum” operator in MME. When $N = 2$, the sign of the sum of confidence score differences can indicate which base model the MME will choose. However, it does not hold for $N > 2$, where the “maximum” operator cannot be eliminated easily. We will add this discussion to the revision.
>
> \> In the first paragraph on page 5, I do not follow which equation was meant in RHS.
>
> The RHS means $r \cdot \frac{1-\Delta}{1+\Delta} (1-C_{\text{MME or ME}}(1-cos\theta))^{-1/2}$.

---

> ### Author Response · Authors · 2020-11-14
> **Response to Reviewer #4 (2/2)**
>
> \> These statements on page 5 are not clear to me: "... and leads to higher certified ensemble robustness." and "Thus, increasing confidence margins can lead to higher ensemble robustness."
>
> “smaller gradients sum indicates smaller LHS in Equations (2) to (5) and leads to higher certified ensemble robustness.”: the Equations (2) to (5) all have the form “LHS <= RHS”. Therefore, for any $r$, if we reduce the LHS, the inequality becomes satisfiable for more $r$, i.e., the certified robust radius $r$ becomes larger.
>
> “increasing confidence margins can lead to higher ensemble robustness.”: the notation $f_j^{y_0/y_i}(x) := f_j(x)_{y0} -f_j(x)_{y_i}$ -- the confidence margin between class $y_0$ and $y_i$. Terms in this form appear in the RHS of Equations (2) to (5). Therefore, by increasing the confidence margin, the inequality becomes satisfiable for more $r$, i.e., the certified robust radius $r$ becomes larger. We will make the discussion clear in our revision.
>
> \> For GD loss, why only pairwise summations were considered? Isn't it easier to look at the overall diversity of gradients?
>
> As the theorems show, the robustness of ensembles has a direct connection with the cosine similarity between base models’ gradients. Thus, to consider the diversity within an ensemble, we use the pairwise summations, and it would be very computationally and memory expensive to consider the overall diversity of gradients.
>
> \> All pairwise computation of quantities in regularizer terms should come with some computational complexity. It would be nice to include a discussion on this.
>
> Thanks for the suggestion. During our regularization loss computation, we need to first back prop N terms for gradient computation, then there are $O(N^2)$ terms for pair-wise GD loss and $O(N)$ terms for CM loss. We will add the discussion in our revision.
>
> \> Results for Salman et al., 2019 on Table 2 do not match the paper of Salman et al., 2019 and imagenet results are not convincing. I understand ImageNet data can take too long but should we worry that DRT requires more hyper-parameter tuning? Some discussion on this would be helpful.
>
> For SmoothAdv (Salman et al., 2019) results, we consider the standard 10-step PGD with perturbation scale equals to 1.0 without any data augmentation or additional unlabeled data as shown in some of SmoothAdv’s results, which matches the original results in https://github.com/Hadisalman/smoothing-adversarial/tree/master/data/certify/cifar10/PGD_10steps/eps_255/cifar10/resnet110 (Some of the results in the SmoothAdv are based on unlabeled data augmentation which is not our goal here). For fair comparison, we utilize the exact same setting in our MME/WE (SmoothAdv) and DRT + MME/WE (SmoothAdv) without data augmentation.
>
> For ImageNet, the computation cost is high, so currently we only put the results for one parameter trial under each setting. We also plan to show that with some parameter tuning, we can achieve better results and here are the updated results.
>
> ImageNet:
>
> | &emsp;&emsp;&emsp;&emsp;Radius r                            | &emsp;0.00 | &emsp;0.50 | &emsp;1.00 | &emsp;1.50 | &emsp;2.00 | &emsp;2.50 | &emsp;3.00 |
> |:---------------------------------:|:------:|:------:|:------:|:-----:|:------:|:------:|:------:|
> | Gaussian                          |   &emsp;57   |  &emsp;46   |   &emsp;37  |  &emsp;29  |   &emsp;19  |   &emsp;15  |   &emsp;12  |
> | MME (Gaussian)              |   &emsp;58   |  &emsp;47   |   &emsp;38  |  &emsp;31  |   &emsp;21  |   &emsp;16  |   &emsp;14  |
> | DRT + MME (Gaussian)   |   &emsp;52   |  &emsp;46   |  &emsp;42   |  &emsp;34  |  &emsp;24   |  &emsp;19   |   &emsp;18  |
> | SmoothAdv                       |   &emsp;54   |  &emsp;49   |  &emsp;43   |  &emsp;37  |  &emsp;27   |  &emsp;25   |   &emsp;20  |
> | MME (SmoothAdv)            |  &emsp;55   |  &emsp;50   |  &emsp;44   |  &emsp;38   |  &emsp;27  |  &emsp;26   |   &emsp;21  |
> | DRT + MME (SmoothAdv) |  &emsp;49   |  &emsp;44   |  &emsp;42   |  &emsp;37   |  &emsp;**29**  |  &emsp;**27**  |   &emsp;**22**  |
>
> We put the detailed hyper-parameter setting discussion on MNIST/CIFAR-10 in Appendix E. We can see that with larger GD loss weight, the DRT could perform better on the large radius, and more careful GD loss weights optimization is required for small radius, which is naturally limited by the model capacity as discussed by Reviewer #1.
>
> Thank you again for your comments and devotion to reviewing this paper! We will add detailed explanations and include more updated experimental results in the revision. Hope we have answered all your concerns, and you may reconsider the rating of this work according to the rebuttal. We are also happy to discuss if there are further questions.

---

### Official Review · AnonReviewer2 · 2020-10-28
**Lacks non-trivial theoretical extension or demonstrated empirical improvement**

**Rating:** 4
**Confidence:** 4

**Review:**

Certified robustness approaches have been studied for single models based on interval propagation as well as randomized smoothing. The use of ensembles for empirical robustness has also been studied in the literature. This paper attempts to theoretically study the certifiable defense achieved by ensembles. The paper analyzes the standard Weighted Ensemble (WE) and MaxMargin Ensemble (MME) protocols, and proves the necessary and sufficient conditions for robustness under smoothness assumptions. The key idea is to show and utilize the diversification of gradients and large confidence margins.

Pros:

+ The paper addresses an important challenge of adversarial robustness via ensembles and attempts to develop theoretical bounds for these defenses.

Cons:

- Some theoretical discussion is rather straight-forward formulation of consequences of the definitions of robustness and ensemble methods. Proposition 1 and Theorem 1 follow directly from the definition of WE and MME, and r-robustness.

- There are several notation lapses which make the paper difficult to read. In definition 2, it appears the intent is to take the partial gradient of the model at two different points x and y and then bound the ration of the difference of the gradients and the distance between two points (which would be the curvature). x,y are being used as values and the variable. The boldface font is not used consistently.  Another example is the missing relation between r and epsilon (which is described in the appendix). These make simple results difficult to parse, and negatively effect the readability of the paper and obscure the identification of novelty.

- For Theorem 2, let us set N = 1 (that is the degenerate case when the ensemble is a single model), we see that the rate of change of difference between the top prediction and other predictions are now bounded by a linear spread of the difference divided by the robustness radius r and then we add or subtract (depending on sufficiency/necessity) the impact of curvature. Rearranging the terms so that first and third are on the same side, the theorem will read:
r ( gradient term for prediction difference ) +/-  r^2 ( curvature term for prediction difference ) <=  prediction difference.
Isn't this just a Taylor series approximation of the prediction difference using bounds on the curvature term (assumed as part of the definition)?
Now, if we bring back any arbitrary N, the same would be applicable by the definition of how decisions are made by WE (the "prediction difference" term is now changed).
Theorem 3 terms can be similarly rearranged to make the statement more obvious. Is the reviewer missing some non-obvious observation or challenge in proving Theorem 2/3?

- Despite several other papers also using the general term of "certified robustness", it is important to note that this robustness is against a rather benign perturbation model. Typical perturbations (in particular adversarial attacks or other natural perturbations such as fog, rain for vision) do not fall into this class. But the reviewer is not concerned with this aspect heavily given the prevalence of the rather generic name of "certified robustness" for such approaches.

- The empirical enforcement of diversity in ensembles has been studied in literature such as Pang et al 2019. https://arxiv.org/abs/1901.09981 uses cosine similarity.  Experimental evaluation with these methods is crucial to understand the value of the proposed approach. The current evaluation is very limited.

Questions and suggestions to the authors:

1. It might be a good idea to bring the relationship between r and epsilon in the main text from the appendix. r-robustness is the fundamental definition and in its current presentation r does not occur anywhere.

2. Could you help understand the concern in Weakness bit 3? This will help the reviewer better appreciate the theoretical novelty of the paper. Currently, the theoretical statements appear to make obvious statements once we get through the rather clunky notation and presentation choices.

3. Why is cosine used for diversity? There are several ways to enforce diversity. Was cosine selected arbitrarily or is it motivated by some theoretical insight?

The paper addresses an important challenge of mathematically well-motivated defenses against perturbations. In its current form, the paper appears to formulate rather simplistic observations as theoretical results with a rather dense presentation. The empirical evaluation is significantly incomplete.

After discussion with authors
----------------------------------------

The reviewer thanks the authors for clarifications. With the confusion from the typos resolved, the paper is easier to follow. The technical correctness of the paper is not in doubt any more. But two major concerns still remain:

1.  The diversity enforcement has been reported before, and more comprehensive discussion of related work would be useful.  A detailed empirical analysis would also make the paper balanced and not heavily reliant on the novelty/depth of theoretical contribution.

2.  The main claimed theoretical contribution summarized in Lemma B.1 is tedious but a derivation of an obvious statement (as sketched out in the original review).

The reviewer is raising the score to encourage this work, but still holding to the recommendation of not accepting the paper in its current form.

---

> ### Author Response · Authors · 2020-11-14
> **Response to Reviewer #2 (1/2)**
>
> Thanks for your constructive comments on this work. Here are the answers to your major concerns.
>
> \> Some theoretical discussion is rather straight-forward formulation.
>
> Though Theorem 1 may look intuitive at the first glance, it is non-trivially followed from the definition of MME. Concretely, MME selects the base model that has the maximum margin between the top and runner-up class. This “maximum” is a discrete operator and poses challenges especially in the multi-class setting. We cannot assert that the model predicts the true label by simply looking at the margins between only the true label and other labels unless carefully filtering out possible violated cases as we do in our Lemma B.1. With the lemma, we derive Theorem 1. And Theorem 1 lays the foundation for the subsequent continuous analysis of the discrete MME. On the other hand, the obvious of the final theoretical conclusion also verifies its correctness.
>
> \> Notation lapses.
>
> Thanks for pointing out! We will fix the typo in Definition 2:
> The $\{x_0 + \delta: \|\delta\|_2 \le \epsilon\}$ is actually $\{x_0 + \delta: \|\delta\|_2 \le r\}$.
> The boldface and gradient variable will be revised to be consistent and correct.
>
> We will also make it clear that the beta-smoothness definition (inherited from optimization literature) is equivalent to the curvature bound (as recently used in certified robustness literature).
>
> \> Theorems 2 and 3 are pure Taylor series approximation.
>
> Theorems 2 and 3 combine preceding theoretical findings with Taylor expansion. For example, the (Sufficient Condition) of Theorem 3 is based on non-trivial Lemma B.1.
> Also, from the theorems, we show that for both ensembles, especially MME, we can directly apply Taylor expansion on the vector sum of confidence scores of base models. This is a nice property that typical NNs do not have due to nonlinearity such as softmax. These theorems enable us to have a non-trivial quantitative robustness comparison between the ensembles and the base model in Corollary 1.
>
> \> The use of the general term “certified robustness”.
>
> Though there is a growing body of certified robustness for threat models beyond Lp, since the certified robustness for Lp is still unsatisfactory, the existing work still mainly focuses on this fundamental setting [1]. Therefore, by default, the literature uses the “certified robustness” to refer to the Lp threat model [2,3]. The generalization to other threat models often uses specific terms [4,5].
>
> [1] L Li et al. "SoK: Certified Robustness for Deep Neural Networks." arXiv:2009.04131.
>
> [2] S Singla and S Feizi. "Second-Order Provable Defenses against Adversarial Attacks." ICML 2020.
>
> [3] H Zhang et al. "Towards Stable and Efficient Training of Verifiably Robust Neural Networks." ICLR 2019.
>
> [4] M Jeet, et al. "Towards Verifying Robustness of Neural Networks Against A Family of Semantic Perturbations." CVPR 2020.
>
> [5] R, Anian, et al. "Efficient Certification of Spatial Robustness." arXiv preprint arXiv:2009.09318.

---

> ### Author Response · Authors · 2020-11-14
> **Response to Reviewer #2 (2/2)**
>
> \> The empirical enforcement of diversity in ensembles has been studied in literature such as Pang et al 2019. https://arxiv.org/abs/1901.09981 uses cosine similarity.
>
> Thanks for pointing out the related work and we will add the discussion in our related work. Our paper mainly focuses on the certified robustness of ensemble models, while Pang et al. focuses on the empirical robustness against existing attacks. According to [1], these trained models have very limited certified robustness unlike ours. Also, since they do not use techniques to enhance the certified robustness (such as randomized smoothing or IBP), it is unclear how to compare with them in terms of the certified robustness given they cannot provide it. In general, neither Pang et al or  https://arxiv.org/abs/1901.09981 have experimented on large-scale ImageNet dataset which also makes it challenging to compare, but we will definitely add the discussion with these empirical ensemble models.
>
> \> Bring the relationship between r and epsilon in the main text from the appendix. r-robustness is the fundamental definition and in its current presentation r does not occur anywhere.
>
> In definition 2, the $\epsilon$ is actually $r$, and we will fix it in the revision. The boldfaced $\epsilon$ throughout the paper is a random variable standing for the noise added to the input, and $r$ stands for the robustness radius. We will also make this clear in the preliminary part of the revised version.
>
> \> Why is cosine used for diversity?
>
> Theoretically, as justified in Theorem 2 and 3, the small joint gradient helps for certified robustness. According to the law of cosines, it is connected to the cosine similarity between base model’s gradient, which is an intuitive indicator for model diversity and conforms to the intuition that diversified base models improve the ensemble robustness.
> Empirically, regularizing to increase diversity in this way indeed improves both the certified robustness (by our experiments) and empirical robustness (by Pang et al and https://arxiv.org/abs/1901.09981) of ensemble models.
>
> Thanks for your constructive comments and suggestions again! We will fix the typos, add explanations, and include more experiments in the revision. Hope we have answered all your concerns, and you may reconsider the rating of this work according to the rebuttal. We are also happy to answer any following questions and suggestions.

---

> > ### Comment · AnonReviewer2 · 2020-11-15
> > **Concerns remain ..**
> >
> > The reviewer thanks the authors for clarifications. With the confusion from the typos resolved, the paper is easier to follow but some major concerns remain preventing the reviewer from raising the score.
> >
> > 1. " Thanks for pointing out the related work and we will add the discussion in our related work. Our paper mainly focuses on the certified robustness of ensemble models, while Pang et al. focuses on the empirical robustness against existing attacks .."
> >
> > Since the diversity enforcement has been reported before, the reviewer hopes that the authors will agree that establishing the novelty and non-trivialness of the theoretical results are currently central to the paper's main contribution.
> >
> > 2. "We cannot assert that the model predicts the true label by simply looking at the margins between only the true label and other labels unless carefully filtering out possible violated cases as we do in our Lemma B.1. ... On the other hand, the obvious of the final theoretical conclusion also verifies its correctness."
> >
> > The reviewer is concerned that the key theoretical contribution summarized in Lemma B.1 is tedious but a derivation of an obvious statement (as sketched out in the original review). While the reviewer acknowledges that not all lemmas or theorems should be surprising, the obvious nature of the statement makes it difficult to be excited about the lemma. The follow-up theorems are obtained via Taylor expansion.
> >
> > This is a very promising line of work and comprehensive empirical results will make it stronger (and no longer reliant heavily on the novelty of the theoretical contribution). The reviewer is quite optimistic about the approach but unable to whole heartedly support acceptance of the paper in its current form (which appears to be a work in progress).
> >
> > The reviewer will raise the score to improve the ranking of the paper and hopefully encourage this work, but still hold to the recommendation of not accepting the paper in its current form.

---

> > > ### Author Response · Authors · 2020-11-25
> > > **Response for Remaining Concerns**
> > >
> > > Thank you very much for your prompt reply and the detailed comments after reading our rebuttal!
> > >
> > > > Since the diversity enforcement has been reported before, the reviewer hopes that the authors will agree that establishing the novelty and non-trivialness of the theoretical results are currently central to the paper's main contribution.
> > >
> > > Thanks for pointing this out! Yes, we definitely agree that the novelty and non-trivialness of the theoretical results are central to the main contribution. Following this comment, we will emphasize the novelty of the proposed gradient diversity analysis with theoretical justification, as well as the proposed two training regularizers. We will also highlight the superiority of the empirical performance following the suggestions. Please let us know if there are further suggestions to help improve our work. We have also added a corresponding discussion on the related work in our revision.
> > >
> > > > The reviewer is concerned that the key theoretical contribution summarized in Lemma B.1 is tedious but a derivation of an obvious statement (as sketched out in the original review).
> > >
> > > We thank the reviewer for pointing out the tediousness of the proof for Lemma B.1. We would like to note that within the proof sketch, there are several detailed non-trivial parts that are necessary for the rigorous guarantees. For instance, when proving Lemmas B.1 and B.2, we need to carefully analyze each possible layout of confidence score distribution and filter out possible violations to guarantee the robustness. As a result, the proof may look long while the details are essential to guarantee the rigorousness of the analysis. We added such discussion in the revision (blue text before Theorem 1) to illustrate these details.
> > >
> > > > Comprehensive empirical results.
> > >
> > > Thank you for the comments and we have updated and added the complete results following all the reviews. In particular, we have updated new empirical results and show that the proposed TRS outperforms the baselines on MNIST, CIFAR-10, and ImageNet (Tables 10, 11, and 12). We also added additional detailed ablation studies for the two proposed regularizers in Appendix E, showing the importance for both.
> > >
> > > We want to thank the reviewer again for your devotion to reading this paper, your constructive comments, and your appreciation of our work. We hope you could re-evaluate this work based on our updated revision. Thank you!

---

### Official Review · AnonReviewer3 · 2020-10-28
**Interesting theory, limited experiments with no benefit for large-scale data**

**Rating:** 5
**Confidence:** 4

**Review:**

The paper offers a novel idea regarding improving robustness of ensemble models with a rigorous mathematical background. A comparison between two types of ensemble models (Weighted Ensembles and Max-Margin Ensembles) and to single models offers very good insight into the theoretical dynamics of certified robustness.
Unfortunately, the presented methods for practical applications are two simple regularization terms (of questionable form, see below), and, more importantly the authors are not able to conclusively show any benefits empirically - only MNIST and CIFAR10 show improvements vs baselines, ImageNet is evaluated as the only large-scale dataset, but with negative results.

Positive:
* a proof is offered why certain regularization is beneficial for robustness, while other works are only based on empirical results
* the work consists of several proofs, greatly extending the theoretical background of ensemble model robustness
* even if you ignore the proposed regularization term, theoretical insights of the robustness of Weighted Ensemble vs Max-Margin Ensemble vs single model are of value for further understanding the topic. (Figure 3 in the appendix is great)
* the regularization term seems to work consistently well for CIFAR10 and MNIST

Negative:
* The paper has too much information in too little space - “related work” and “conclusion” seem artificially short to fit the 8 pages.
* The paper reads like two papers glued together (one would be chapter 2 + chapter 4 + appendix E about the regularization term DRT, the other is chapter 3 + appendix C + D about the theoretical certified robustness of the ensemble types with smoothness). The paper has two main pillars (like mentioned above: chapter 2 and chapter 3), but chapter 3 offers no results/conclusions for chapter 4 (the experiments), it seems strangely misplaced between ch. 2 and ch. 4.
* some notations are a bit sloppy, some assumptions are questionable
* the GD regularization term (Gradient diversity loss) should have a different form: In the previous section the authors state that “..the magnitude of gradient sum could be efficiently reduced as long as their directions are diverse.”, but the GD term is still the L2 norm instead of cosine similarity.
* Experiments are extremely limited and the proposed method does not work on large-scale data

Detailed comments:
* In Sub-Section “Key factors for the certified robustness of an ensemble” (p. 5): The authors write “Though reducing the gradient magnitude [something positive], it may hurt the model accuracy significantly” and continue showing how the L2 norm of the sum of two vectors contains the cosine similarity, which is critical for model diversity. Furthermore they argue that by reducing the L2 norm the cosine similarity is also reduced and continue using L2 as their regularization term called Gradient Diversity Loss. But if reducing the base model gradient norm is potentially so bad, then why is the regularization not exclusively the cosine similarity between the two gradients? This should be evaluated.
* based on the analysis, the authors argue the confidence score margin has also important influence on the robustness, so it is artificially increased through another regularization term called Confidence Margin Loss. This opens two follow-up questions:
    * naturally cross entropy already tries to reach high confidence scores, so the question is if the Confidence Margin Loss achieves anything that cross entropy implicitly does not, or if it maybe only stabilizes training convergence. I would like to see some results on that
    * By increasing the confidence scores further, any improvement in regarding robustness may come at the cost of an increased expected calibration error - this would be interesting to evaluate
* Ablation study: an additional evaluation of how beneficial each loss term is for the robustness is missing
* Theorem 4 and onwards: The random variable “epsilon” cannot by element of R^d, since a random variable is a function.
* “Definition 1” has an important typo: “epsilon” should be “r”, otherwise the term “r-robust” makes no sense, if no r is present in the definition
* section 2.1: The whole paper is about Weighted Ensembles and Max-Margin-Ensembles, but their formal definitions are moved to the appendix, please move to main text
* Proposition 1: In “min”, “y_i =/= y_0” is not required because if y_i == y_0, the term is equal 0 (this is not an error, but less clutter is preferable)
* Definition 2: the second gradient is w.r.t. x, but the argument is y
* Theorem 4 and 5, and Corollary 2 assume the confidence scores across several base models are i.i.d. and symmetric random variables. I am not usure if this assumption holds in any case and would appreciate some discussion here: How can the output be seen as symmetrical? Confidence scores are between 0 and 1, so exact symmetry is almost always impossible as long as they are close to 1 (which they are due to the training and if they are in-domain)...


######Post-rebuttal####
The authors have addressed some of my concerns and I have raised my score accordingly

---

> ### Author Response · Authors · 2020-11-14
> **Response to Reviewer #3 (1/2)**
>
> We thank the reviewer’s appreciation for our work, and we address the questions below.
>
> \> Paper structure.
>
> In this paper, we aim to conduct both theoretical and empirical analysis, that means, in chapter 2 we set up the formulations for different ensemble protocols and analyze the general robustness conditions, in chapter 3 we analyze the concrete certified robustness of different ensembles with randomized smoothing, and in chapter 4 we conduct the extensive experiments. In particular, the analysis in chapter 2 motivates the GD loss and CM loss in our proposed regularizer, and chapter 3 further verifies the CM loss (in discussion at the end of 3.1). Overall, we hope our paper can provide solid theoretical justification for the proposed regularizer DRT (chapter 2,3), as well as comprehensive experimental results and comparison with baselines (chapter 4). With one more page limit for the revision, we will expand the “related work” and “conclusion” with more details as well.
>
> \> The form of Gradient Diversity (GD) regularization term.
>
> Thanks for pointing it out. Given $\|a+b\|_2 = \sqrt{a^2 + b^2 + 2\|a\| \|b\| cos(a,b)}$, we can see that increasing direction diversity is equivalent to minimizing the magnitude of the joint gradient. We finally use the L2 magnitude of joint gradient here since: (1) the direction diversity itself, if measured by cosine similarity, is narrowly bounded in $[-1,1]$, and numerically hard to regularize due to the denominator $\|a\| \|b\|$; (2) the analysis shows that small gradient magnitude also helps, and joint L2 can regularize both the gradient magnitude and direction diversity.
> Actually, minimizing the joint L2 magnitude implies an interplay between gradient magnitude and direction diversity. A small joint L2 magnitude can be achieved by either diversified directions or small L2 magnitude from individual gradients. As we discussed above, solely regularizing gradient direction itself is hard. As discussed in the paper, solely regularizing the individual gradient hurts the model performance. However, the joint L2 combines the two goals together and circumvents these difficulties.
> We believe this simple but effective GD regularization term can benefit general robust ensemble training and interest the conference audience as well as the community. We will make this contribution clear in our revision.
>
> \> Limited Experiments and ImageNet results.
>
> We did several numerical experiments as shown in Appendix D.4 and extensive empirical certified robustness evaluation on MNIST, CIFAR, ImageNet dataset while the DRT shows its strength on MNIST, CIFAR. For ImageNet, the performance was limited due to the computation cost so we only tried one parameter set for each setting. We will follow the suggestion and tune other parameters to report the full set of results on ImageNet in our revision. Here are the updated results on ImageNet:
>
> ImageNet:
>
> |    &emsp;&emsp;&emsp;   Radius $r$      | &emsp;$0.00$ | &emsp;$0.50$ | &emsp;$1.00$ | &emsp;$1.50$ | &emsp;$2.00$ | &emsp;$2.50$ | &emsp;$3.00$ |
> |:---------------------:|:------:|:------:|:------:|:------:|:------:|:------:|:------:|
> | Gaussian                          |   &emsp;57   |  &emsp;46   |   &emsp;37  |  &emsp;29  |   &emsp;19  |   &emsp;15  |   &emsp;12  |
> | MME (Gaussian)              |   &emsp;58   |  &emsp;47   |   &emsp;38  |  &emsp;31  |   &emsp;21  |   &emsp;16  |   &emsp;14  |
> | DRT + MME (Gaussian)   |   &emsp;52   |  &emsp;46   |  &emsp;42   |  &emsp;34  |  &emsp;24   |  &emsp;19   |   &emsp;18  |
> | SmoothAdv                       |   &emsp;54   |  &emsp;49   |  &emsp;43   |  &emsp;37  |  &emsp;27   |  &emsp;25   |   &emsp;20  |
> | MME (SmoothAdv)            |  &emsp;55   |  &emsp;50   |  &emsp;44   |  &emsp;38   |  &emsp;27  |  &emsp;26   |   &emsp;21  |
> | DRT + MME (SmoothAdv) |  &emsp;49   |  &emsp;44   |  &emsp;42   |  &emsp;37   |  &emsp;**29**  | &emsp;**27**  |   &emsp;**22**  |
>
> That shows, DRT can indeed improve the certified accuracy on the large radius even with limited parameter tuning.

---

> ### Author Response · Authors · 2020-11-14
> **Response to Reviewer #3 (2/2)**
>
> \> The Confidence Margin Loss’s effect.
>
> Thanks for your suggestions and we run the experiments by considering the GD Loss or Confidence Margin Loss separately on CIFAR-10 dataset:
>
> | &emsp;&emsp;&emsp;Radius r          	     | &emsp;0.00 | &emsp;0.25 | &emsp;0.50 | &emsp;0.75 | &emsp;1.00 | &emsp;1.25 | &emsp;1.50 | &emsp;1.75 | &emsp;2.00 |
> |:---------------------:|:------:|:------:|:------:|:------:|:------:|:------:|:------:|:------:|:------:|
> | MME (Gaussian)            | &emsp;80.8 | &emsp;68.2 | &emsp;53.4 | &emsp;38.4 | &emsp;29.0 | &emsp;19.6 | &emsp;15.6 | &emsp;11.6  |  &emsp;8.8  |
> | GDL + MME (Gaussian) | &emsp;81.0 | &emsp;69.0 | &emsp;55.6 | &emsp;41.9 | &emsp;30.4 | &emsp;24.8 | &emsp;20.1 | &emsp;16.9 | &emsp;14.7 |
> | CML + MME (Gaussian) | &emsp;81.2 | &emsp;69.4 | &emsp;54.4 | &emsp;39.6 | &emsp;29.2 | &emsp;21.6 | &emsp;17.0 | &emsp;13.1 | &emsp;12.8 |
> | DRT + MME (Gaussian) | &emsp;81.4 | &emsp;70.4 | &emsp;57.8 | &emsp;43.8 | &emsp;31.6 | &emsp;26.2 | &emsp;22.4 | &emsp;18.8 | &emsp;16.6 |
>
> We observed that both GD Loss (GDL) and Confidence Margin Loss (CML) could have positive effects on improving the certified accuracy and GDL plays a major role in the larger radius. Compared to standard cross-entropy training, we also found: 1. CML can further enlarge the top-1 confidence gap compared to standard cross-entropy training (the avg confidence gap moves from 0.4 to 0.7)  2. CML could lead the model to be more uncalibrated thus increasing the Expected Calibration Error (ECE) with the increasing of the CML weight from 0 to 5.0 (ECE increases from 2e-3 to 8e-3 onMNIST and 6e-2 to 1.3e-1 on CIFAR10, $\sigma= 0.5$, bins = 10). This shows that CML could have its actual effects instead of stabilizing the training procedure only.
>
> \> Theorem 4 and onwards: The random variable “$\epsilon$” cannot be an element of $R^d$.
>
> Thanks for pointing it out. We will make it precise as: constant $\epsilon$ on $R^d$.
>
> \> “$\epsilon$” should be “$r$”, otherwise the term “$r$-robust” makes no sense, if no $r$ is present in the definition
>
> Thanks a lot for pointing out. This is indeed our typo, and we will revise this in the revision.
>
> \> The whole paper is about Weighted Ensembles and Max-Margin-Ensembles, but their formal definitions are moved to the appendix, please move to the main text
>
> Thank you! We will update it following the suggestion in the revision given that we will have one more page.
>
> \> In Proposition 1: In “$\min$”, “$y_i \neq y_0$” is not required because if $y_i = y_0$, the term is equal 0 (this is not an error, but less clutter is preferable)
>
> Thank you! We will update it.
>
> \> Definition 2: the second gradient is w.r.t. $x$, but the argument is $y$
>
> Thank you for the careful check and we will update it.
>
> \> Symmetrical distribution assumption for confidence score distribution.
>
> Thanks for pointing it out and sorry for the confusion. Here the symmetrical distribution means that the distribution is symmetrical around the mean. Our assumption is with respect to the noise added to the input. For fixed input $x$ and specific label $y_0$, around the mean confidence, we assume that when adding the input noise, the confidence distribution is symmetrical. Our empirical experiments could verify this assumption (see https://i.imgur.com/2gpilpx.png), and we will add these figures in the revision.
>
> Thanks for your insightful comments. We will add the new results, fix the typos, and add the definitions back in the main paper as you mentioned in our revised version. Hope we have answered all the questions, and hope you can reconsider the rating of this solid work.

---

### Official Review · AnonReviewer1 · 2020-10-30

**Rating:** 6
**Confidence:** 4

**Review:**

This paper studies the following problem: How to train a certifiably robust classifier with ensemble methods? The authors considered two types of ensembles: the weighted-average ensemble and large-margin ensemble. They first derived theoretically sufficient and necessary conditions for robustness under two types of ensembles, with the conclusion that large confidence margin and diversified gradients are two factors which contributes to the robustness of ensemble models. Diversity-regularized training, a method of designing loss functions for training ensemble models, is proposed motivated by their theoretical findings. They applied this methodology to randomized smoothing, performed extensive experiments and showed non-trivial improvement over single model methods.

The paper is very well-written and provided extremely detailed discussion about many different aspects of the problem, both theoretically and empirically. From the reviewer's point of view, this is the strongest part of this work. I am very impressed by the level of detail in the appendix, which covers many interesting questions like under which scenario is WE better than MME, why is ensemble before smoothing better than ensemble after smoothing, to name a few. I wish more papers in the community are written in this way.

However, my current evaluation to this paper is a weak accept - it is a bit conservative, but I think it's based on some valid concerns, detailed below.

The experimental results, while showed non-trivial improvements over single model baselines, may not be very strong. In most cases, the improvements are like 3~4%, sometimes a little over 5% in smaller radius settings comparing to smoothadv. This improvement is much smaller than some of the earlier works like smoothadv vs gaussian. Also in certain settings, the improvement over other single model baselines becomes very small (e.g. <2% over MACER in Table 12). The performance is also overall very similar to recent ensemble baselines like Liu et al. 2020. My understanding is that this shows a limitation of ensemble-based methods in certifiable robustness.

To summarize, despite of the limitations mentioned above, I think this work is overall good enough for recommending acceptance and thank the authors for their effort.

Minor comments:
I don't quite get the point of appendix D.3, in particular, the reasoning that trying to justify the uniform distribution assumption of confidence scores. The concentration of measure in high dimensional Gaussian does not imply the uniformity of confidence scores. Although I do understand this assumption as a concrete example  trying to get more interpretable results.

---

> ### Author Response · Authors · 2020-11-14
> **Response to Reviewer #1**
>
> Thanks for your thoughtful suggestions. We checked your questions carefully and answered them as follows.
>
> \> Limited improvement of the experiment results.
>
> First, in this paper we mainly want to emphasize our analysis framework and the methodology, and we use the experimental results as a demonstration without tuning the parameters very carefully. Following your suggestion, we actually tune our parameters a bit and we find that we can indeed get better performance as follows:
>
> MNIST:
>
> |       &emsp;&emsp;&emsp;&nbsp;&nbsp;&nbsp;Radius $r$      | &emsp;`$1.25$ | &emsp;$1.50$ | &emsp;$1.75$ | &emsp;$2.00$ | &emsp;$2.25$ | &emsp;$2.50$ |
> |:---------------------:|:------:|:------:|:------:|:------:|:------:|:------:|
> |        Gaussian       |  &emsp;`83.0  |  &emsp;`68.2  |  &emsp;`46.6  |  &emsp;`33.0  |  &emsp;`20.5  |  &emsp;`11.5  |
> |     MME (Gaussian)    |  &emsp;`84.3  |  &emsp;`69.8  |  &emsp;`48.8  |  &emsp;`34.7  |  &emsp;`23.4  |  &emsp;`12.7  |
> |  DRT + MME (Gaussian) |  &emsp;`86.8  |  &emsp;`75.2  |  &emsp;`55.8  |  &emsp;`44.8  |  &emsp;`38.0  |  &emsp;`27.0  |
> |       SmoothAdv       |  &emsp;`87.7  |  &emsp;`80.2  |  &emsp;`66.3  |  &emsp;`43.2  |  &emsp;`34.3  |  &emsp;`24.0  |
> |    MME (SmoothAdv)    |  &emsp;`88.1  |  &emsp;`80.7  |  &emsp;`67.9  |  &emsp;`44.8  |  &emsp;`35.0  |  &emsp;`25.2  |
> | DRT + MME (SmoothAdv) |  &emsp;`**88.5**  |  &emsp;**`83.2**  |  &emsp;**`68.9**  |  &emsp;**`48.2**  |  &emsp;**`39.2**  |  &emsp;**`33.5**  |
>
> CIFAR:
>
> |       &emsp;&emsp;&emsp;&nbsp;&nbsp;&nbsp;Radius $r$      |  &emsp;$1.00$ | &emsp;`$1.25$ | &emsp;$1.50$ | &emsp;$1.75$ | &emsp;$2.00$ |
> |:---------------------:|:------:|:------:|:------:|:------:|:------:|
> | Gaussian         	        | &emsp;23.1 | &emsp;18.3 | &emsp;13.6 | &emsp;10.5 |  &emsp;7.3  |
> | MME (Gaussian)               | &emsp;29.0 | &emsp;19.6 | &emsp;15.6 | &emsp;11.6 |  &emsp;8.8  |
> | DRT + MME (Gaussian)    | &emsp;31.6 | &emsp;26.2 |&emsp;22.4 | &emsp;18.8 | &emsp;16.6 |
> | SmoothAdv       	         | &emsp;34.8 | &emsp;28.5 | &emsp;21.9 | &emsp;18.2 | &emsp;15.7 |
> | MME (SmoothAdv)            | &emsp;36.2 | &emsp;29.8 |&emsp;23.9 | &emsp;19.5 |&emsp;16.2 |
> | DRT + MME (SmoothAdv) |&emsp;**39.4** | &emsp;**35.8** |&emsp;**30.4** | &emsp;**24.0** | &emsp;**20.1** |
> | SWEEN (Liu et al., 2020)  | &emsp;37.2 | &emsp;29.2 | &emsp;24.6 | &emsp;22.0 |&emsp;19.1 |
>
> As you can see, DRT could improve the model’s certified accuracy a lot (over 17% on {MNIST, Gaussian, $r$ = 2.25}) and better than Liu et al. 2020 with 1% to 6% on the large radius. Indeed, for the small radius, the improvement is limited, and we hypothesize that’s due to the sub-models’ capacity limit within the ensemble, which shows the limitation of ensemble-based methods as you mentioned and we will add related discussion in our revision. However, the significant improvement of the DRT ensemble model compared to the ensemble without DRT indicates that DRT could be a very effective and general way to improve the certified robustness of the ensemble.
>
> \> Questions about the uniform distribution assumption.
>
> Thanks for the question and this assumption is actually for a case study and we will make it clear in our revision. Without the uniform distribution assumption, we can prove a general case that the transferability across base models decides which ensemble is better as shown in Appendix D.2. Here we just want to give an illustration in a specific regim and therefore we make the uniform assumption which may not hold exactly in practice. We think it would be an interesting future direction to generalize the analysis to other distributions such as the Gaussian distribution which corresponds to locally linear classifiers. We will make this discussion clear in Appendix D.3’s remark.
>
> Thanks for your thoughtful suggestions and appreciation. We will update these new results in our revision and make the whole paper to be clearer. Hope we have answered all your questions, so that you can increase your score. We are also happy to discuss further for other concerns.

---

### Author Response · Authors · 2020-11-24
**Revision Summary according to the Feedbacks from Reviewers**

We thank all the reviewers for the valuable comments and we have revised our paper accordingly. Specifically, we made the following updates and highlighted the changes in blue in our revision:
1. We updated the performance of DRT on MINST in Tables 1 and 11, on CIFAR-10 in Tables 2 and 12, showing that DRT models could achieve 3.0 to 10.0 certified accuracy improvement on large radii. Notice that our results are also much better than recent certified robust ensemble work (Liu et al., 2020) on CIFAR-10 when r >= 0.75.
2. We updated experimental results of DRT on ImageNet in Table 10 showing that DRT could improve about 2.0 certified accuracy on large radii. This indicates that DRT has great potential on large-scale dataset.
3. We added more discussion about related work and our contribution compared with these work in “Related work” on page 2, following **R3**’s suggestion.
4. We formally defined the scope of “certified robustness” in “Related work” on page 2, following **R2**’s suggestion.
5. We fixed the typo in Definition 1 (epsilon should be r), and fixed the inconsistency of boldface font and typo in Definition 4, following **R2**’s suggestion.
6. We moved the definition of Weighted Ensemble and Maximum Margin Ensemble to Section 2.1, following **R3**’s suggestion.
7. We deleted "$y_i \neq y_0$" in Proposition 1, following **R3**’s suggestion.
8. We explained the technical details and challenges in proving Theorem 1, and provided an explanation on how Theorem 2 is proved, following **R3**'s suggestion.
9. We noted and discussed that the number of classes is not necessarily equal to the number of base models in the remark after Theorem 1, following **R4**’s suggestion.
10. We connected the "$\beta$-smoothness" with the curvature bound after Definition 4, following **R2**’s suggestion.
11. We discussed the challenges of extending the condition to $N>2$ for Maximum Margin Ensemble after Theorem 3, following **R4**’s suggestion.
12. We explained what "RHS" means in the discussion after Corollary 1, and how reducing LHS and increasing RHS induces larger certified robustness in "Key factors" subsection on page 5, following **R4**’s suggestion.
13. We explained the mechanism of Gradient Diversity (GD) Loss in "Key Factors" subsection on page 5 and Section 2.2 on page 6, showing that it jointly regularizes both the base models' gradient magnitude and diversity, following **R2** and **R3**’s suggestion.
14. We updated the "$\epsilon \in \mathbb{R}^d$" to "$\epsilon$ is a random variable supported on $\mathbb{R}^d$" in Theorems 4, 5 and Corollary 2, following **R3**’s suggestion.
15. We extended the Conclusion section with key factor discussion following **R3**’s suggestion.
16. We added the discussion about the uniform distribution assumption into Appendix D.3, following **R1**’s suggestion.
17. We expanded the proof of Theorems 2 and 3 with more details and illustrations about Lagrange remainder in Appendix B.3, following **R4**’s suggestion.
18. We clarified the specific setting we used for SmoothAdv baseline (without pretraining and unlabelled data augmentation) in Appendix E, following **R4**’s suggestion.
19. We analyzed the effects of GD Loss and Confidence Margin Loss separately and added the discussion in Appendix E.5, following **R3**’s suggestion. It shows that both loss terms could have positive effects on improving DRT model’s certified robustness instead of stabilizing the training procedure only.

---

### Decision · Program_Chairs · 2021-01-07
**Final Decision**

**Decision:**

Reject

**Comment:**

I thank the authors and reviewers for the lively discussions. Although reviewers agreed the work is interesting, there are some concerns about the significance of the results and experiments. None of the reviewers were strongly supportive of the paper while majority of them suggest that the paper needs a bit more work before being accepted. Also, reviewers suggest that the paper is not easy to follow and its writing should be improved. Given all, I think the paper , at the current stage, is below the accept threshold. I encourage authors to edit the paper according to the suggestions by the reviewers.